# Rethinking Inverse Reinforcement Learning: from Data Alignment to Task Alignment

**Weichao Zhou**
Boston University
Boston, MA 02215
zwc662@bu.edu

**Wenchao Li**
Boston University
Boston, MA 02215
wenchao@bu.edu

## Abstract

Many imitation learning (IL) algorithms use inverse reinforcement learning (IRL) to infer a reward function that aligns with the demonstrations. However, the inferred reward function often fails to capture the underlying task objective. In this paper, we propose a novel framework for IRL-based IL that prioritizes task alignment over conventional data alignment. Our framework is a semi-supervised approach that leverages expert demonstrations as weak supervision signals to derive a set of candidate reward functions that align with the task rather than only with the data. It adopts an adversarial mechanism to train a policy with this set of reward functions to gain a collective validation of the policy's ability to accomplish the task. We provide theoretical insights into this framework's ability to mitigate task-reward misalignment and present a practical implementation. Our experimental results show that our framework outperforms conventional IL baselines in complex and transfer learning scenarios. The complete code are available at https://github.com/zwc662/PAGAR.

## 1 Introduction

Inverse reinforcement learning (IRL) Ng and Russell [2000], Finn et al. [2017] has become a popular method for imitation learning (IL), allowing policies to be trained by learning reward functions from expert demonstrations Abbeel and Ng [2004], Ho and Ermon [2016]. Despite its widespread use, IRL-based IL faces significant challenges that often stem from overemphasizing data alignment rather than task alignment. For instance, reward ambiguity, where multiple reward functions can be consistent with the expert demonstrations, makes it difficult to identify the correct reward function. This problem persists even when there are infinite data Ng and Russell [2000], Cao et al. [2021], Skalse et al. [2022a,b]. Additionally, limited availability of demonstrations can further exacerbate this problem, as the data may not fully capture the nuances of the task. Misaligned reward functions can lead to policies that optimize the wrong objectives, resulting in poor performance and even reward hacking Hadfield-Menell et al. [2017], Amodei et al. [2016], Pan et al. [2022], a phenomenon where the policy exploits loopholes in the inferred reward function. These challenges highlight the limitation of exclusively pursuing data alignment in solving real-world tasks.

In light of these considerations, this paper advocates for a paradigm shift from a narrow focus on data alignment to a broader emphasis on task alignment. Grounded in a general formalism of task objectives, we propose identifying the task-aligned reward functions that more accurately reflect the underlying task objectives in their policy utility spaces. Expanding on this concept, we explore the intrinsic relationship between the task objective, reward, and expert demonstrations. This relationship leads us to a novel perspective where expert demonstrations can serve as weak supervision signals for identifying a set of candidate task-aligned reward functions. Under these reward functions, the expert

achieves high -— but not necessarily optimal —- performance. The rationale is that achieving high performance under a task-aligned reward function is often adequate for real-world applications.

Building on this premise, we leverage IRL to derive the set of candidate task-aligned reward functions and propose Protagonist Antagonist Guided Adversarial Reward (PAGAR), a semi-supervised framework designed to mitigate task-reward misalignment by training a policy with this candidate reward set. PAGAR adopts an adversarial training mechanism between a protagonist policy and an adversarial reward searcher, iteratively improving the policy learner to attain high performance across the candidate reward set. This method moves beyond relying on deriving a single reward function from data, enabling a collective validation of the policy's similarity to expert demonstrations in terms of effectiveness in accomplishing tasks. Experimental results show that our algorithm outperforms baselines on complex IL tasks with limited demonstrations and in challenging transfer environments. We summarize our contributions below.

- Introduction of Task Alignment in IRL-based IL: We present a novel perspective that shifts the focus from data alignment to task alignment, addressing the root causes of reward misalignment in IRL-based IL.
- Protagonist Antagonist Guided Adversarial Reward (PAGAR): We propose a new semi-supervised framework that leverages adversarial training to improve the robustness of the learned policy.
- Practical Implementation: We present a practical implementation of PAGAR, including the adversarial reward searching mechanism and the iterative policy-improving process. Experimental results demonstrate superior performances in complex and transfer learning environments.

## 2   Related Works

IRL-based IL circumvents many challenges of traditional IL such as compounding error Ross and Bagnell [2010], Ross et al. [2011], Zhou et al. [2020] by learning a reward function to interpret the expert behaviors Ng et al. [1999], Ng and Russell [2000] and then learning a policy from the reward function via reinforcement learning (RL)Sutton and Barto [2018]. However, the learned reward function may not always align with the underlying task, leading to reward misspecification Pan et al. [2022], Skalse and Abate [2022], reward hacking Skalse et al. [2022b], and reward ambiguity Ng and Russell [2000], Cao et al. [2021]. The efforts on alleviating reward ambiguity include Max-Entropy IRL Ziebart et al. [2008], Max-Margin IRL Abbeel and Ng [2004], Ratliff et al. [2006], and Bayesian IRL Ramachandran and Amir [2007]. GAN-based methods Ho and Ermon [2016], Jeon et al. [2018], Finn et al. [2016], Peng et al. [2019], Fu et al. [2018] use neural networks to learn reward functions from limited demonstrations. However, these efforts that aim to address reward ambiguity fall short of mitigating the general impact of reward misalignment which can be caused by various reasons such as IRL making false assumptions about the relationship between expert policy and expert reward function Skalse et al. [2022a], Hong et al. [2023]. Other attempts to mitigate reward misalignment involve external information other than expert demonstrations Hejna and Sadigh [2023], Zhou and Li [2018, 2022a,b]. Our work adopts the generic setting of IRL-based IL without needing additional information. The idea of considering a reward set instead of focusing on a single reward function is supported by Metelli et al. [2021] and Lindner et al. [2022]. However, these works target reward ambiguity instead of reward misalignment. Our protagonist and antagonist setup is inspired by the concept of unsupervised environment design (UED) Dennis et al. [2020]. In this paper, we develop novel theories in the context of reward learning.

## 3   Preliminaries

**Reinforcement Learning (RL)** models the environment as a Markov Decision Process $\mathcal{M} = \langle \mathbb{S}, \mathbb{A}, \mathcal{P}, d_0 \rangle$ where $\mathbb{S}$ is the state space, $\mathbb{A}$ is the action space, $\mathcal{P}$ is the transition probability, $d_0$ is the initial state distribution. A *policy* $\pi(a|s)$ determines the probability of an RL agent performing an action $a$ at state $s$. By successively performing actions for $T$ steps from an initial state $s^{(0)} \sim d_0$, a *trajectory* $\tau = s^{(0)} a^{(0)} s^{(1)} a^{(1)} \ldots s^{(T)}$ is produced. A state-action based *reward function* is a mapping $r : \mathbb{S} \times \mathbb{A} \rightarrow \mathbb{R}$. The soft Q-value function of $\pi$ is $\mathcal{Q}_\pi(s, a) = r(s, a) + \gamma \cdot \mathbb{E}_{s' \sim \mathcal{P}(\cdot|s,a)} [\mathcal{V}_\pi(s')]$ where $\gamma \in (0, 1]$ is a discount factor, $\mathcal{V}_\pi$ is the soft state-value function of $\pi$ defined as $\mathcal{V}_\pi(s) := \mathbb{E}_{a \sim \pi(\cdot|s)} [\mathcal{Q}_\pi(s, a)] + \mathcal{H}(\pi(\cdot|s))$, and $\mathcal{H}(\pi(\cdot|s))$ is the entropy of $\pi$ at state $s$.

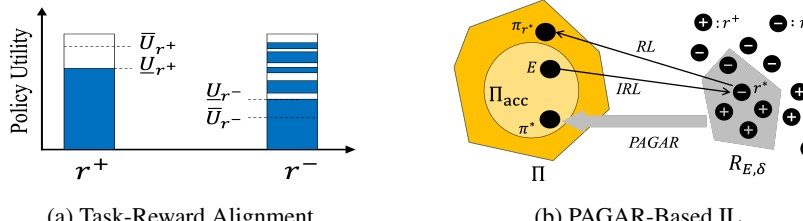

(a) Task-Reward Alignment  (b) PAGAR-Based IL

Figure 1: (a) The two bars respectively represent the policy utility spaces of a task-aligned reward function $r^+$ and a task-misaligned reward function $r^-$. The white color indicates the utilities of acceptable policies, and the blue color indicates the unacceptable ones. Within the utility space of $r^+$, the utilities of all acceptable policies are higher ($\geq \underline{U}_{r^+}$) than those of the unacceptable ones, and the policies with utilities higher than $\overline{U}_{r^+}$ have higher orders than those of utilities lower than $\overline{U}_{r^+}$. Within the utility space of $r^-$, acceptable and unacceptable policies' utilities are mixed together, leading to a low $\underline{U}_{r^-}$ and an even lower $\overline{U}_{r^-}$. (b) IRL-based IL relies solely on IRL's optimal reward function $r^*$ which can be task-misaligned and lead to an unacceptable policy $\pi_{r^*} \in \Pi \backslash \Pi_{acc}$ while PAGAR-based IL learns an acceptable policy $\pi^* \in \Pi_{acc}$ from a set $R_{E,\delta}$ of reward functions.

The soft advantage of performing action $a$ at state $s$ and then following a policy $\pi$ afterwards is $\mathcal{A}_\pi(s,a) = \mathcal{Q}_\pi(s,a) - \mathcal{V}_\pi(s)$. The expected return of $\pi$ under a reward function $r$ is given as $U_r(\pi) = \mathbb{E}_{\tau \sim \pi}[\sum_{t=0}^\infty \gamma^t \cdot r(s^{(t)}, a^{(t)})]$. With a slight abuse of notations, we denote the entropy of a policy as $\mathcal{H}(\pi) := \mathbb{E}_{\tau \sim \pi}[\sum_{t=0}^\infty \gamma^t \cdot \mathcal{H}(\pi(\cdot|s^{(t)}))]$. The standard RL learns an *optimal policy* by maximizing $U_r(\pi)$. The entropy regularized RL learns a *soft-optimal* policy by maximizing the objective function $\mathcal{J}_{RL}(\pi; r) := U_r(\pi) + \mathcal{H}(\pi)$.

**Inverse Reinforcement Learning (IRL)** assumes that a set $E = \{\tau_1, \dots, \tau_N\}$ of expert demonstrations are sampled from the roll-outs of the expert's policy $\pi_E$ and aims to learn the expert reward function $r_E$. IRL Ng and Russell [2000] assumes that $\pi_E$ is *optimal* under $r_E$ and learns $r_E$ by maximizing the margin $U_r(E) - \max_\pi U_r(\pi)$ while Maximum Entropy IRL (MaxEnt IRL) Ziebart et al. [2008] maximizes an entropy regularized objective function $\mathcal{J}_{IRL}(r) = U_r(E) - (\max_\pi U_r(\pi) + \mathcal{H}(\pi))$.

**Generative Adversarial Imitation Learning (GAIL)** Ho and Ermon [2016] draws a connection between IRL and Generative Adversarial Nets (GANs) as shown in Eq.1, where a discriminator $D : \mathbb{S} \times \mathbb{A} \to [0,1]$ is trained by minimizing Eq.1 so that $D$ can accurately identify any $(s,a)$ generated by the agent. Meanwhile, an agent policy $\pi$ is trained as a generator to maximize Eq.1 so that $D$ cannot discriminate $\tau \sim \pi$ from $\tau_E$. Adversarial inverse reinforcement learning (AIRL) Fu et al. [2018] uses a neural-network reward function $r$ to represent $D(s,a) := \frac{\pi(a|s)}{\exp(r(s,a)) + \pi(a|s)}$, rewrites $\mathcal{J}_{IRL}$ as minimizing Eq.1, and proves that the optimal reward satisfies $r^* \equiv \log \pi_E \equiv \mathcal{A}_{\pi_E}$. By training $\pi$ with $r^*$ until optimality, $\pi$ will behave just like $\pi_E$.

$$\mathbb{E}_{(s,a) \sim \pi}[\log D(s,a)] + \mathbb{E}_{(s,a) \sim \pi_E}[\log(1 - D(s,a))] \tag{1}$$

## 4 Task-Reward Alignment

In this section, we formalize the concept of task-reward misalignment in IRL-based IL. We start by defining a notion of task based on the framework from Abel et al. [2021].

**Definition 1 (Task).** Given the policy hypothesis set $\Pi$, a **task** $(\Pi, \preceq_{task}, \Pi_{acc})$ is specified by a partial order $\preceq_{task}$ over $\Pi$ and a non-empty set of acceptable policies $\Pi_{acc} \subseteq \Pi$ such that $\forall \pi_1 \in \Pi_{acc}$ and $\forall \pi_2 \notin \Pi_{acc}$, $\pi_2 \preceq_{task} \pi_1$ always hold.

**Remark:** The notions of policy acceptance and order allow the definition of **task** to accommodate a broad range of real-world tasks[1] including the standard RL tasks (learning the optimal policy from a reward function $r$): given a reward function $r$ and a policy hypothesis set $\Pi$, the standard RL **task** can be written as a tuple $(\Pi, \preceq_{task}, \Pi_{acc})$ where $\preceq_{task}$ satisfies $\forall \pi_1, \pi_2 \in \Pi, \pi_1 \preceq_{task} \pi_2 \Leftrightarrow U_r(\pi_1) \leq U_r(\pi_2)$, and $\Pi_{acc} = \{\pi \mid \forall \pi' \in \Pi. \pi' \preceq_{task} \pi\}$ contains all the optimal policies.

---

[1]See examples in Abel et al. [2021].

Designing reward function(s) that align with the underlying task is essential in RL. Whether a designed reward aligns with the task hinges on how policies are ordered by the task and the utilities of the policies under the reward function. Therefore, we define the task-reward alignment by examining the utility spaces of the reward functions. If the acceptable policy set $\Pi_{acc}$ of the task is given, we let $\underline{U}_r := \min_{\pi \in \Pi_{acc}} U_r(\pi)$ be the minimal utility achieved by any acceptable policy under $r$.

**Definition 2** (**Task-Aligned Reward Functions**). A reward function is a *task-aligned reward function* (denoted as $r^+$) if and only if $\forall \pi \in \Pi \backslash \Pi_{acc}, U_{r^+}(\pi) < \underline{U}_{r^+}(\pi)$. Conversely, if this condition is not met, it is a *task-misaligned reward function* (denoted as $r^-$).

The definition suggests that under a task-aligned reward function $r^+$, all acceptable policies for the task yield higher utilities than unacceptable ones. It also suggests that a policy is deemed acceptable as long as its utility is greater than $\underline{U}_{r^+}$ for some task-aligned reward function $r^+$, even if this policy is not optimal. We also examine whether high utility under a reward function $r$ suggests a higher order under $\preceq_{task}$. We define $\overline{U}_r := \max_{\pi \in \Pi} U_r(\pi) \ s.t. \ \forall \pi_1, \pi_2 \in \Pi, U_r(\pi_1) < U_r(\pi) \leq U_r(\pi_2) \Rightarrow$ $(\pi_1 \preceq_{task} \pi) \wedge (\pi_1 \preceq_{task} \pi_2)$, which is the highest utility threshold such that any policy achieving a higher utility than $\overline{U}_r$ has a higher order than those achieving lower utilities than $\overline{U}_r$. In Figure 1(a) we illustrate how $\overline{U}_r$ and $\underline{U}_r$ vary between task-aligned and misaligned reward functions.

**Proposition 1.** *Given the policy order $\preceq_{task}$ of a task, for any two reward functions $r_1, r_2$, if $\{\pi \mid U_{r_1}(\pi) \geq \overline{U}_{r_1}\} \subseteq \{\pi \mid U_{r_2}(\pi) \geq \overline{U}_{r_2}\}$, then there must exist policies $\pi_1 \in \{\pi \mid U_{r_1}(\pi) \geq \overline{U}_{r_1}\}, \pi_2 \in \{\pi \mid U_{r_2}(\pi) \geq \overline{U}_{r_2}\}$ such that $U_{r_1}(\pi_2) \leq U_{r_1}(\pi_1)$ and $\pi_2 \preceq_{task} \pi_1$ while $U_{r_2}(\pi_2) \geq U_{r_2}(\pi_1)$.*

This proposition implies that a high threshold $\overline{U}_r$ indicates that a high utility corresponds to a high order in terms of $\preceq_{task}$. In particular, for any task-aligned reward function $r^+$, $\{\pi \mid U_{r^+}(\pi) \geq \overline{U}_{r^+}\} \subseteq \Pi_{acc} \equiv \{\pi \mid U_{r^+}(\pi) \geq \underline{U}_{r^+}\}$ (see proof in Appendix A.2). Thus, a small $\{\pi \mid U_{r^+}(\pi) \geq \overline{U}_{r^+}\}$ leads to a large $\{\pi \mid U_{r^+}(\pi) \in [\underline{U}_{r^+}, \overline{U}_{r^+}]\}$. Hence, a task-aligned reward function $r^+$ is more likely to be aligned with the task if it has a wide $[\underline{U}_{r^+}, \overline{U}_{r^+}]$ and a narrow $[\overline{U}_{r^+}, \max_{\pi \in \Pi} U_{r^+}(\pi)]$.

## 4.1 Mitigate Task-Reward Misalignment in IRL-Based IL

In IRL-based IL, a key challenge is that *the underlying task is unknown*, making it difficult to assert if a learned policy is acceptable. We denote the optimal reward function learned from the demonstration set $E$ as $r^*$, and the optimal policy under $r^*$ as $\pi_{r^*}$. When $\pi_{r^*}$ has a poor performance under $r_E$, it is considered to have a high $Regret(\pi_{r^*}, r_E)$ which is defined in Eq.2. If $Regret(\pi_{r^*}, r_E) > \max_{\pi' \in \Pi} U_{r_E}(\pi') - \underline{U}_{r_E}$, then $\pi_{r^*}$ is unacceptable and $r^*$ is task-misaligned.

$$Regret(\pi, r) := \max_{\pi' \in \Pi} U_r(\pi') - U_r(\pi) \tag{2}$$

Several factors can lead to a high $Regret(\pi_{r^*}, r_E)$. For instance, Viano et al. [2021] shows that when expert demonstrations are collected in an environment whose dynamical function differs from that of the learning environment, $|Regret(\pi_{r^*}, r_E)|$ can be positively related to the discrepancy between those dynamical functions. Additionally, we prove in Appendix A.1 that learning from only a few representative expert trajectories can also result in a large $|Regret(\pi_{r^*}, r_E)|$ with a high probability.

Our insight for mitigating such potential task-reward misalignment in IRL-based IL is to *shift our focus from learning an optimal policy that maximizes the intrinsic $r_E$ to learning an acceptable policy $\pi^*$ that achieves a utility higher than $\underline{U}_{r^+}$ under any task-aligned reward function $r^+$*. Our approach is to treat the expert demonstrations as weak supervision signals based on the following.

**Theorem 1.** *Let $\mathcal{I}$ be an indicator function. For any $k \geq \left\{\min_{r^+} \sum_{\pi \in \Pi} \mathcal{I}\{U_{r^+}(\pi) \geq U_{r^+}(\pi_E)\}\right\}$, if $\pi^*$ satisfies $\left\{\sum_{\pi \in \Pi} \mathcal{I}\{U_r(\pi) \geq U_r(\pi^*)\}\right\} < |\Pi_{acc}|$ for all $r \in R_{E,k} := \left\{r \mid \sum_{\pi \in \Pi} \mathcal{I}\{U_r(\pi) \geq U_r(\pi_E)\} \leq k\right\}$, then $\pi^*$ is an acceptable policy, i.e., $\pi^* \in \Pi_{acc}$. Additionally, if $k < |\Pi_{acc}|$, such an acceptable policy $\pi^*$ is guaranteed to exist.*

The statement suggests that we can obtain an acceptable policy by training it to attain high performance across a reward function set $R_{E,k}$ that includes all the reward functions where, for each reward function at most $k$ policies outperform the expert policy $\pi_E$. The minimal value of $k$ is determined by all the task-aligned reward functions in the reward hypothesis set. Appendix A.2 provides the proof.

**How to build** $R_{E,k}$**?** Building $R_{E,k}$ involves setting the parameter $k$. If $r_E$ is a task-aligned reward function and $\pi_E$ is optimal solely under $r_E$, then the minimal $k = 0$, and $R_{E,0}$ only contains $r_E$. However, relying on a singleton $R_{E,0}$ equates to applying vanilla IRL, which is susceptible to misalignment issues, as noted earlier. It is crucial to recognize that $r_E$ might not meet the task-aligned reward function criteria specified in Definition 2, even though its optimal policy $\pi_E$ is acceptable. This situation necessitates a positive $k$, thereby expanding $R_{E,k}$ beyond a single function and changing the role of expert demonstrations from strong supervision to weak supervision. Note that we suggest letting $k \leq |\Pi_{acc}|$ instead of allowing $k \to \infty$ because $R_{E,\infty}$ would then encompass all possible reward functions, and it is impractical to identify a policy capable of achieving high performance across all reward functions. Letting $k \leq |\Pi_{acc}|$ guarantees there exists a feasible policy $\pi^*$, e.g., $\pi_E$ itself. As the task alignment of each reward function typically remains unknown in IRL settings, this paper proposes treating $k$ as an adjustable parameter – starting with a small $k$ and adjusting based on empirical learning outcome, allowing for iterative refinement for alignment with task requirements.

In practice, $\Pi$ can be uncountable, e.g., a Gaussian policy. Hence, we adapt the concept of $k$ in $R_{E,k}$ to a hyperparameter $\delta \leq \delta^* := \max_r \mathcal{J}_{IRL}(r)$, leading us to redefine $R_{E,k}$ as a $\delta$-*optimal reward function set* $R_{E,\delta} := \{r \mid \mathcal{J}_{IRL}(r) \geq \delta\}$. This superlevel set includes all the reward functions under which the optimal policies outperform the expert by at most $-\delta$. If $\delta$ is appropriately selected such that $R_{E,\delta}$ includes task-aligned reward functions, we can mitigate reward misalignment by satisfying the conditions outlined in Definition 3, which are closely related to Definition 2 and Proposition 1.

**Definition 3** (**Mitigation of Task-Reward Misalignment**). Assuming that the reward function set $R_{E,\delta}$ contains task-aligned reward function $r^+$'s, the mitigation of task-reward misalignment in IRL-based IL is to learn a policy $\pi^*$ such that (i) (Weak Acceptance) $\forall r^+ \in R_{E,\delta}, U_{r^+}(\pi^*) \geq \underline{U}_{r^+}$, or (ii) (Strong Acceptance) $\forall r^+ \in R_{E,\delta}, U_{r^+}(\pi^*) \geq \overline{U}_{r^+}$.

While condition (i) states that $\pi^*$ is acceptable for the task, i.e., $\pi^* \in \Pi_{acc}$, condition (ii) further states that $\pi^*$ have a high order in terms of $\preceq_{task}$. Hence, condition (i) is weaker than (ii) because a policy $\pi^*$ satisfying (ii) automatically satisfies (i) according to Definition 2. Given the uncertainty in identifying which reward function is aligned, our solution is to **train a policy to achieve high utilities under all reward functions** in $R_{E,\delta}$ to satisfy the conditions in Definition 3. We explain this approach in the following semi-supervised paradigm, PAGAR.

# 5 Protagonist Antagonist Guided Adversarial Reward (PAGAR)

PAGAR is an adversarial reward searching paradigm which iteratively searches for a reward function to challenge a policy learner by incurring a high regret as defined in Eq.2. We refer to the policy to be learned as the *protagonist policy* and re-write it as $\pi_P$. We then introduce a second policy, dubbed *antagonist policy* $\pi_A$, as a proxy of the $\arg\max_{\pi' \in \Pi} U_r(\pi')$ for Eq.2. For each reward function $r$, we call the regret of $\pi_P$ under $r$, i.e., $Regret(\pi_P, r) = \max_{\pi_A \in \Pi} U_r(\pi_A) - U_r(\pi_P)$, the *Protagonist Antagonist Induced Regret*. We then formally define PAGAR in Definition 4.

**Definition 4** (Protagonist Antagonist Guided Adversarial Reward (**PAGAR**)). Given a candidate reward function set $R$ and a protagonist policy $\pi_P$, PAGAR searches for a reward function $r$ within $R$ to maximize the *Protagonist Antagonist Induced Regret*, i.e., $\max_{r \in R} Regret(\pi_P, r)$.

**PAGAR-based IL** *learns a policy from* $R_{E,\delta}$ *by minimizing the worst-case Protagonist Antagonist Induced Regret* via $MinimaxRegret(R_{E,\delta})$ *as defined in Eq.3 where* $R$ *can be any input reward function set and is set as* $R = R_{E,\delta}$ *in PAGAR-based IL.*

$$MinimaxRegret(R) := \arg\min_{\pi_P \in \Pi} \max_{r \in R} Regret(\pi_P, r) \qquad (3)$$

Our subsequent discussion will focus on identifying the sufficient conditions for PAGAR-based IL to mitigate task-reward misalignment as described in Definition 3. In particular, we consider the case where $\mathcal{J}_{IRL}(r) := U_r(E) - \max_{\pi} U_r(\pi)$. We use $L_r$ to denote the Lipschitz constant of $r(\tau)$, and $W_E$ to denote the smallest Wasserstein 1-distance $W_1(\pi, E)$ between $\tau \sim \pi$ of any $\pi$ and $\tau \sim E$, i.e., $W_E \triangleq \min_{\pi \in \Pi} W_1(\pi, E)$. Then, we have Theorem 2.

**Theorem 2** (Weak Acceptance). *If the following conditions (1) (2) hold for* $R_{E,\delta}$*, then the optimal protagonist policy* $\pi_P := MinimaxRegret(R_{E,\delta})$ *satisfies* $\forall r^+ \in R_{E,\delta}, U_{r^+}(\pi_P) \geq \underline{U}_{r^+}$.

*(1) There exists $r^+ \in R_{E,\delta}$s, and $\max\limits_{r^+\in R_{E,\delta}} \{\max\limits_{\pi\in\Pi} U_{r^+}(\pi) - \overline{U}_{r^+}\} < \min\limits_{r^+\in R_{E,\delta}} \{\overline{U}_{r^+} - \underline{U}_{r^+}\};$*

*(2) $\forall r^+ \in R_{E,\delta}$, $L_{r^+} \cdot W_E - \delta \leq \max\limits_{\pi\in\Pi} U_{r^+}(\pi) - \overline{U}_{r^+}$ and $\forall r^- \in R_{E,\delta}$, $L_{r^-} \cdot W_E - \delta < \min\limits_{r^+\in R_{E,\delta}} \{\overline{U}_{r^+} - \underline{U}_{r^+}\}.$*

This statement shows the conditions for PAGAR-based IL to attain the *'Weak Acceptance'* goal described in Definition 3. The condition (1) states that the task-aligned reward functions in $R_{E,\delta}$ all have a high level of alignment in matching $\preceq_{task}$ within their high utility ranges. The condition (2) requires that for the policy $\pi^* = \arg\min\limits_{\pi\in\Pi} W_1(\pi, E)$, the performance difference between $E$ and $\pi^*$ is small enough under all $r \in R_{E,\delta}$. Since for each reward function $r \in R_{E,\delta}$, the performance difference between $E$ and the optimal policy under $r$ is bounded by $\delta$, condition (2) implicitly requires that $\pi^*$ not only performs well under any task-aligned reward function $r^+$ (thus being acceptable in the task) but also achieve relatively low regret under task-misaligned reward function $r^-$. However, the larger the rage $[\overline{U}_{r^+}, \underline{U}_{r^+}]$ is across the task-aligned reward function $r^+$, the less strict the requirement for low regret under $r^-$ becomes. The proof can be found in Appendix A.5. The following theorem further suggests that a $\delta$ close to its upper-bound $\delta^* := \max\limits_r \mathcal{J}_{IRL}(r)$ can help $MinimaxRegret(R_{E,\delta})$ gain a better chance of finding an acceptable policy for the underlying task and attain the *'Strong Acceptance'* goal described in Definition 3.

**Theorem 3** (Strong Acceptance). *Assume that the condition (1) in Theorem 2 holds for $R_{E,\delta}$. If for any $r \in R_{E,\delta}$, $L_r \cdot W_E - \delta \leq \min\limits_{r^+\in R_{E,\delta}} \{\max\limits_{\pi\in\Pi} U_{r^+}(\pi) - \overline{U}_{r^+}\}$, then the optimal protagonist policy $\pi_P = MinimaxRegret(R_{E,\delta})$ satisfies $\forall r^+ \in R_{E,\delta}$, $U_{r^+}(\pi_P) \geq \overline{U}_{r^+}$.*

**When do these assumptions hold?** The condition (1) in Theorem 2 requires all the task-aligned reward functions in $R_{E,\delta}$ exhibit a high level of conformity with the policy order $\preceq_{task}$. Being task-aligned already sets a strong premise for satisfying this condition. We further posit that this condition is more easily satisfied when the task has a binary outcome, such as in reach-avoid tasks so that the aligned and misaligned reward functions tend to have higher discrepancy than tasks with quantitative outcomes. In the experimental section, we validate this hypothesis by evaluating tasks of this kind. Regarding condition (2) of Theorem 2 and the assumptions of Theorem 3, which basically require the existence of a policy with low regret across $R_{E,\delta}$ set, it is reasonable to assume that expert policy meets this criterion.

## 5.1 Comparing PAGAR-Based IL with IRL-Based IL

We illustrate the difference between IRL-based IL and PAGAR-based IRL in Fig.1(b). While IRL-based IL aims to learn the optimal policy $\pi_{r^*}$ under the IRL-optimal reward $r^*$, PAGAR-based IL learns a policy $\pi^*$ from the reward function set $R_{E,\delta}$. Both PAGAR-based IL and IRL-based IL are zero-sum games between a policy learner and a reward learner. However, while IRL-based IL only aims to reach equilibrium at a single reward function under strong assumptions, e.g., sufficient demonstrations, convex reward and policy spaces, etc., PAGAR-based IL can reach equilibrium with a **mixture of reward functions** without those assumptions.

**Proposition 2.** *Given arbitrary reward function set $R$, there exists a constant $c$ and a distribution $\mathcal{R}_\pi$ over $R$ such that $MinimaxRegret(R)$ yields the same policy as $\arg\max\limits_{\pi\in\Pi} \left\{ \frac{Regret(\pi, r_\pi^*)}{c - U_{r_\pi^*}(\pi)} \cdot U_{r_\pi^*}(\pi) + \mathbb{E}\limits_{r\sim\mathcal{R}_\pi(r)}[(1 - \frac{Regret(\pi, r)}{c - U_r(\pi)}) \cdot U_r(\pi)] \right\}$ where $r_\pi^* = \arg\max\limits_{r\in R} U_r(\pi)$ s.t. $r \in \arg\max\limits_{r'\in R} Regret(\pi, r')$.*

A detailed derivation can be found in Theorem 6 in Appendix A.4. In a nutshell, $\mathcal{R}_\pi(r)$ is a baseline distribution over $R$ such that (i) $c \equiv \mathbb{E}\limits_{r\sim\mathcal{R}_\pi}[U_r(\pi)]$ holds for all the $\pi$'s that do not always perform worse than any other policy under $r \in R$, (ii) among all the $\mathcal{R}_\pi$'s that satisfy the condition (i), we pick the one with the minimal $c$; and (iii) for any other policy $\pi$, $\mathcal{R}_\pi$ uniformly concentrates on $\arg\max\limits_{r\in R} U_r(\pi)$. Note that in PAGAR-based IL, where $R_{E,\delta}$ is used in place of arbitrary $R$, $\mathcal{R}_\pi$ is a distribution over $R_{E,\delta}$ and $r_\pi^*$ is constrained to be within $R_{E,\delta}$. Essentially, the mixed reward functions dynamically assign weights to $r \sim \mathcal{R}_\pi$ and $r_\pi^*$ depending on $\pi$. If $\pi$ performs worse under

$r_\pi^*$ than under many other reward functions ($U_{r_\pi^*}(\pi)$ falls below $c$), a higher weight will be allocated to using $r_\pi^*$ to train $\pi$. Conversely, if $\pi$ performs better under $r_\pi^*$ than under many other reward functions ($c$ falls below $Ur_\pi^*(\pi)$), a higher weight will be allocated to reward functions drawn from $\mathcal{R}_\pi$. Furthermore, we prove in Appendix A.7 that the $MinimaxRegret$ objective function defined in Eq.3 is a convex optimization w.r.t the protagonist policy $\pi_P$.

We also prove in Appendix A.8 that when there is no misalignment issue, i.e., under the ideal conditions for IRL, PAGAR-based IL can either guarantee inducing the same results as IRL-based IL with $\delta = \max_r \mathcal{J}_{IRL}(r)$, or guarantee inducing an acceptable $\pi_P$ by making $\max_r \mathcal{J}_{IRL}(r) - \delta$ no greater than $\max_{\pi \in \Pi} U_{r^+}(\pi) - \overline{U}_{r^+}$ for $r^+ \in R_{E,\delta}$.

# 6  A Practical Approach to Implementing PAGAR-based IL

We solve $MinimaxRegret(R_{E,\delta})$, by alternating between policy learning and reward search. Based on Eq.3, we introduce an on-and-off policy learning framework and an adversarial reward search objective. Moreover, we embed the constraint $r \in R_{E,\delta}$ into the reward search objective using IRL, resulting in a **meta-algorithm** compatible with various IRL methods.

## 6.1  Policy Optimization with On-and-Off Policy Samples

Given an intermediate learned reward function $r$, we use RL to train $\pi_P$ to minimize the regret $\min_{\pi_P} Regret(\pi_P, r) = \min_{\pi_P} \{\max_{\pi_A} U_r(\pi_A)\} - U_r(\pi_P)$ as indicated by Eq.3 where $\pi_A$ is trained to serve as the optimal policy under $r$ as noted in Section 5. Since we have to sample trajectories with $\pi_A$ and $\pi_P$, we propose to combine off-policy and on-policy samples to optimize $\pi_P$ so that we can leverage the samples maximally. **Off-Policy:** We leverage the Theorem 1 in Schulman et al. [2015] to derive a bound for the utility subtraction: $U_r(\pi_P) - U_r(\pi_A) \le \sum_{s \in \mathbb{S}} \rho_{\pi_A}(s) \sum_{a \in \mathbb{A}} \pi_P(a|s) A_{\pi_A}(s,a) +$
$C \cdot \max_s D_{TV}(\pi_A(\cdot|s), \pi_P(\cdot|s))^2$ where $\rho_{\pi_A}(s) = \sum_{t=0}^{T} \gamma^t Prob(s^{(t)} = s|\pi_A)$ is the discounted visitation frequency of $\pi_A$, $A_{\pi_A}$ is the advantage function without considering the entropy, and $C$ is some constant. Then we follow the derivation in Schulman et al. [2017], which is based on Theorem 1 in Schulman et al. [2015], to derive from the inequality an importance sampling-based objective function $\mathcal{J}_{\pi_A}(\pi_P; r) := \mathbb{E}_{s \sim \pi_A}[\min(\xi(s,a) \cdot A_{\pi_A}(s,a), clip(\xi(s,a), 1 - \sigma, 1 + \sigma) \cdot A_{\pi_A}(s,a)]$ where $\sigma$ is a clipping threshold, $\xi(s,a) = \frac{\pi_P(a|s)}{\pi_A(a|s)}$ is an importance sampling rate. The details can be found in Appendix B.1. This objective function allows us to train $\pi_P$ by using the trajectories of $\pi_A$. **On-Policy:** We also optimize $\pi_P$ with the standard RL objective function $\mathcal{J}_{RL}(\pi_P; r)$ by using the trajectories of $\pi_P$ itself. As a result, the objective function for optimizing $\pi_P$ is $\max_{\pi_P \in \Pi} \mathcal{J}_{\pi_A}(\pi_P; r) + \mathcal{J}_{RL}(\pi_P; r)$. As for $\pi_A$, we only use the standard RL objective function, i.e., $\max_{\pi_A \in \Pi} \mathcal{J}_{RL}(\pi_A; r)$. Although the computational complexity equals the sum of the complexities of RL update steps for $\pi_A$ and $\pi_P$, these two RL update steps can be executed in parallel.

## 6.2  Regret Maxmization with On-and-Off Policy Samples

Given the intermediate learned protagonist and antagonist policy $\pi_P$ and $\pi_A$, according to $MinimaxRegret$ in Eq.3, we need to optimize $r$ to maximize $U_r(\pi_A) - U_r(\pi_P)$. In practice, we found that the subtraction between the estimated $U_r(\pi_A)$ and $U_r(\pi_P)$ can have a high variance. To resolve this issue, we derive two reward improvement bounds to approximate this subtraction.

**Theorem 4.** *Suppose policy $\pi_2 \in \Pi$ is the optimal solution for $\mathcal{J}_{RL}(\pi; r)$. Then , the inequalities Eq.4 and 5 hold for any policy $\pi_1 \in \Pi$, where $\alpha = \max_s D_{TV}(\pi_1(\cdot|s), \pi_2(\cdot|s))$, $\epsilon = \max_{s,a} |\mathcal{A}_{\pi_2}(s,a)|$, and $\Delta\mathcal{A}(s) = \mathbb{E}_{a \sim \pi_1}[\mathcal{A}_{\pi_2}(s,a)] - \mathbb{E}_{a \sim \pi_2}[\mathcal{A}_{\pi_2}(s,a)].$*

$$\left| U_r(\pi_1) - U_r(\pi_2) - \sum_{t=0}^{\infty} \gamma^t \mathbb{E}_{s^{(t)} \sim \pi_1}\left[\Delta\mathcal{A}(s^{(t)})\right]\right| \le \frac{2\alpha\gamma\epsilon}{(1-\gamma)^2} \tag{4}$$

$$\left| U_r(\pi_1) - U_r(\pi_2) - \sum_{t=0}^{\infty} \gamma^t \mathbb{E}_{s^{(t)} \sim \pi_2}\left[\Delta\mathcal{A}(s^{(t)})\right]\right| \le \frac{2\alpha\gamma(2\alpha + 1)\epsilon}{(1-\gamma)^2} \tag{5}$$

**Algorithm 1** An Meta-Algorithm for Imitation Learning with PAGAR

**Input**: Expert demonstration set $E$, IRL objective function $J_{IRL}$, loss bound $\delta$, parameter $\lambda \geq 0$, initial protagonist policy $\pi_P$, antagonist policy $\pi_A$, reward function $r$, maximum iteration number $N$.
**Output**: $\pi_P$

1: **for** iteration $i = 0, 1, \ldots, N$ **do**
2:     Sample trajectory sets $\mathbb{D}_A \sim \pi_A$ and $\mathbb{D}_P \sim \pi_P$
3:     **Optimize** $\pi_A$: estimate $\mathcal{J}_{RL}(\pi_A; r)$ with $\mathbb{D}_A$; update $\pi_A$ to maximize $\mathcal{J}_{RL}(\pi_A; r)$
4:     **Optimize** $\pi_P$: estimate $\mathcal{J}_{RL}(\pi_P; r)$ with $\mathbb{D}_P$; estimate $\mathcal{J}_{\pi_A}(\pi_P; r)$ with $\mathbb{D}_A$; update $\pi_A$ to maximize $\mathcal{J}_{RL}(\pi_P; r) + \mathcal{J}_{\pi_A}(\pi_P; r)$
5:     **Optimize** $r$: estimate $\mathcal{J}_{PAGAR}(r; \pi_P, \pi_A)$ with $\mathbb{D}_P$ and $\mathbb{D}_A$; estimate $\mathcal{J}_{IRL}(r)$ with $\mathbb{D}_A$ and $E$; update $r$ to minimize $\mathcal{J}_{PAGAR}(r; \pi_P, \pi_A) + \lambda \cdot (\delta - \mathcal{J}_{IRL}(r))$; then update $\lambda$ to maximize $\delta - \mathcal{J}_{IRL}(r)$
6: **end for**
7: **return** $\pi_P$

By letting $\pi_P$ be $\pi_1$ and $\pi_A$ be $\pi_2$, Theorem 4 enables us to bound $U_r(\pi_A) - U_r(\pi_P)$ by using either only the samples of $\pi_A$ or only those of $\pi_P$. Following Fu et al. [2018], we let $r$ be a proxy of $\mathcal{A}_{\pi_2}$ in Eq.4 and 5. Then we derive two loss functions $\mathcal{J}_{R,1}(r; \pi_P, \pi_A)$ and $\mathcal{J}_{R,2}(r; \pi_P, \pi_A)$ for $r$ as shown in Eq.6 and 7 where $C_1$ and $C_2$ are constants proportional to the estimated maximum KL divergence between $\pi_A$ and $\pi_P$ (to bound $\alpha$ Schulman et al. [2015]). The objective function for $r$ is then $\mathcal{J}_{PAGAR} := \mathcal{J}_{R,1} + \mathcal{J}_{R,2}$. The complexity equals that of computing the reward along the trajectories sampled from $\pi_A$ and $\pi_P$.

$$\mathcal{J}_{R,1}(r; \pi_P, \pi_A) := \mathop{\mathbb{E}}_{\tau \sim \pi_A} \left[ \sum_{t=0}^{\infty} \gamma^t \left( \xi(s^{(t)}, a^{(t)}) - 1 \right) \cdot r(s^{(t)}, a^{(t)}) \right] + C_1 \cdot \max_{(s,a) \sim \pi_A} |r(s,a)| \quad (6)$$

$$\mathcal{J}_{R,2}(r; \pi_P, \pi_A) := \mathop{\mathbb{E}}_{\tau \sim \pi_P} \left[ \sum_{t=0}^{\infty} \gamma^t \left( 1 - \frac{1}{\xi(s^{(t)}, a^{(t)})} \right) \cdot r(s^{(t)}, a^{(t)}) \right] + C_2 \cdot \max_{(s,a) \sim \pi_P} |r(s,a)| \quad (7)$$

### 6.3 A Meta-Algorithm for Solving PAGAR-Based IL

Given an IRL objective function $\mathcal{J}_{IRL}$, we enforce the constraint $r \in R_{E,\delta}$ by adding to $\mathcal{J}_{PAGAR}(r; \pi_P, \pi_A)$ a penalty term $\lambda \cdot (\delta - \mathcal{J}_{IRL})$, where $\lambda$ is a Lagrangian parameter. The resulting objective function for optimizing $r$ becomes $\min_{r \in R} \mathcal{J}_{PAGAR}(r; \pi_P, \pi_A) + \lambda \cdot (\delta - \mathcal{J}_{IRL}(r))$.

We initialize $\lambda$ with a large value to prioritize satisfying the constraint $r \in R_{E,\delta}$ and update it based on $\delta - \mathcal{J}_{IRL}$ (the details can be found in Appendix B.4). Algorithm 1 outlines our meta-algorithm for PAGAR-based IL. The algorithm takes an IRL objective $\mathcal{J}_{IRL}$ as an input, and alternates between policy and reward learning. In line 3, $\pi_A$ is trained via RL with its own sample set $\mathbb{D}_A$. In line 4, we train $\pi_P$ via the on-and-off policy approach in Section 6.1 with both $\mathbb{D}_A$ and $\pi_P$'s sample set $\mathbb{D}_P$. Finally, in line 5, $\mathcal{J}_{PAGAR}$ is estimated from both $\mathbb{D}_A$ and $\mathbb{D}_P$, while $\mathcal{J}_{IRL}$ is from $\mathbb{D}_A$ and $E$.

## 7 Experiments

The goal of our experiments is to assess whether using PAGAR-based IL can efficiently mitigate reward misalignment under conditions that are not ideal for IRL. We present the main results below and provide details and additional results in Appendix C.

### 7.1 Discrete Navigation Tasks

**Benchmarks:** We consider a maze navigation environment where the task objective is compatible with Definition 1. Our benchmarks include two discrete domain tasks from the Mini-Grid environments Chevalier-Boisvert et al. [2023]: *DoorKey-6x6-v0*, and *SimpleCrossingS9N1-v0*. In both tasks, the agent needs to interact with the environmental objects which are **randomly positioned in every episode while the agent can only observe a small, unblocked area in front of it**. The default reward, which is always zero unless the agent reaches the target, is used to evaluate the performance of learned policies. Due to partial observability and the implicit hierarchical nature of the task, these environments are considered challenging for RL and IL, and have been extensively used for benchmarking curriculum RL and exploration-driven RL.

**Baselines:** We compare our approach with two standard baselines: GAIL Ho and Ermon [2016] and VAIL Peng et al. [2019]. GAIL has been introduced in Section 3. VAIL is based on GAIL but additionally optimizes a variational discriminator bottleneck (VDB) objective. Our approach uses the IRL techniques behind those two baseline algorithms, resulting in two versions of Algorithm 1, denoted as PAGAR-GAIL and PAGAR-VAIL, respectively. More specifically, if the baseline optimizes a $J_{IRL}$ objective, we use the same $J_{IRL}$ objective in Algorithm 1. Also, we extract the reward function $r$ from the discriminator $D$ as mentioned in Section 3. More details are in Appendix C.1. PPO Schulman et al. [2017] is used for policy training in GAIL, VAIL, and ours with a replay buffer of size $2048$. Additionally, we compare our algorithm with a state-of-the-art (SOTA) IL algorithm, IQ-Learn Garg et al. [2021], which, however, is not compatible with our algorithm because it does not explicitly optimize a reward function. The policy and the reward functions are all approximated using convolutional networks.

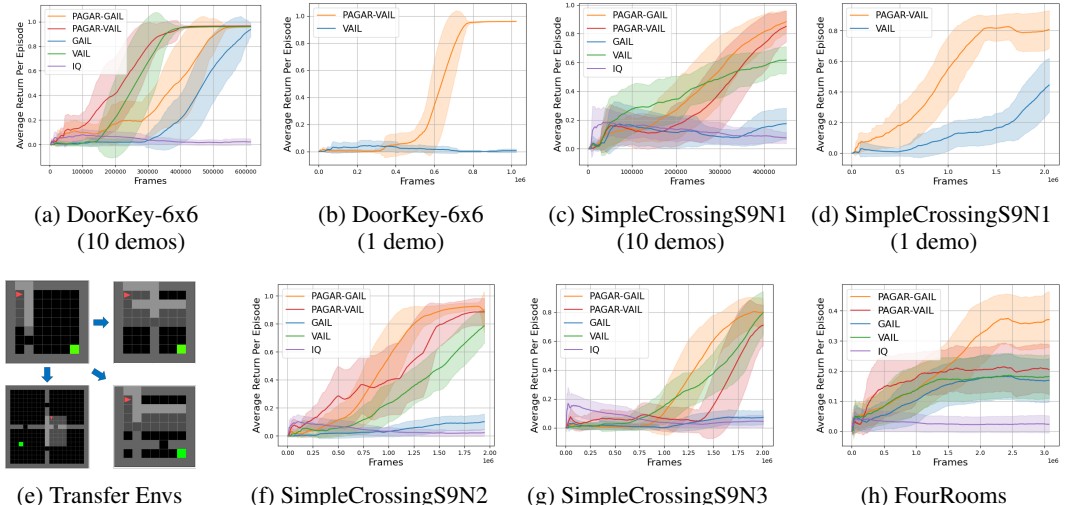

(a) DoorKey-6x6 (10 demos)    (b) DoorKey-6x6 (1 demo)    (c) SimpleCrossingS9N1 (10 demos)    (d) SimpleCrossingS9N1 (1 demo)

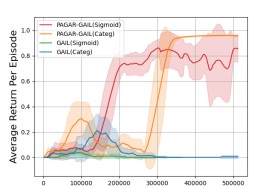

(e) Transfer Envs    (f) SimpleCrossingS9N2    (g) SimpleCrossingS9N3    (h) FourRooms

Figure 2: Comparing Algorithm 1 with baselines in partial observable navigation tasks. The suffix after each 'PAGAR-' indicates which IRL technique is used in Algorithm 1. The $y$ axis indicates the average return per episode. The $x$ axis indicates the number of time steps.

**IL with Limited Demonstrations.** By learning from 10 expert-demonstrated trajectories with high returns, PAGAR-based IL produces high-performance policies with high sample efficiencies as shown in Figure 2(a) and (c). Furthermore, we compare PAGAR-VAIL with VAIL by reducing the number of demonstrations from 10 to 1. As shown in Figure 2(b) and (d), PAGAR-VAIL produces high-performance policies with significantly higher sample efficiencies.

**IL under Dynamics Mismatch.** We demonstrate that PAGAR enables the agent to infer and accomplish the objective of a task even in environments that are substantially different from the one observed during expert demonstrations. As shown in Figure 2(e), we collect 10 expert demonstrations from the *SimpleCrossingS9N1-v0* environment. Then we apply Algorithm 1 and the baselines, GAIL, VAIL, and IQ-learn to learn policies in *SimpleCrossingS9N2-v0, SimpleCrossingS9N3-v0* and *FourRooms-v0*. The results in Figure 2(f)-(g) show that PAGAR-based IL outperforms the baselines in these challenging zero-shot settings.

Figure 3: PAGAR-GAIL in different reward spaces

**IL with Different Reward Hypothesis Sets**. The foundational theories of GAIL and AIRL indicate that different reward function hypothesis sets can affect the equilibrium of their GAN frameworks. We study whether choosing different reward hypothesis sets can influence the performance of Algorithm 1. We compare using a $Sigmoid$ function with a Categorical distribution in the output layer of the discriminator networks in GAIL and PAGAR-GAIL. When using the $Sigmoid$ function, the outputs of $D$ are not normalized, i.e., $\sum_{a \in \mathbb{A}} D(s, a) \neq 1$. When using a Categorical distribution, $\sum_{a \in \mathbb{A}} D(s, a) = 1$. We test GAIL and PAGAR-GAIL in *DoorKey-6x6-v0* environment. As shown in Figure 3, PAGAR-GAIL outperforms GAIL in both cases by using fewer samples.

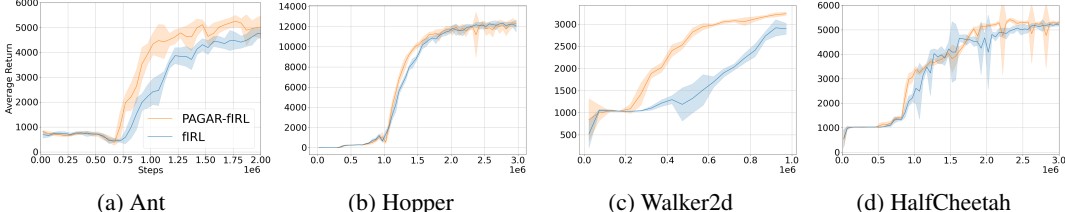

| | (a) Ant | (b) Hopper | (c) Walker2d | (d) HalfCheetah |

Figure 4: Comparing Algorithm 1 with f-IRL in continuous control tasks. 'PAGAR-fIRL' indicates f-IRL is used as the inverse RL algorithm in Algorithm 1. The $y$ axis indicates the average return per episode. The $x$ axis indicates the number of time steps in the environment.

| | RECOIL | PAGAR-RECOIL |
|---|---|---|
| hopper-random | $106.87 \pm 2.69$ | $\mathbf{111.16 \pm 0.51}$ |
| halfcheetah-random | $80.84 \pm 17.62$ | $\mathbf{92.94 \pm 0.10}$ |
| walker2d-random | $108.40 \pm 0.04$ | $108.40 \pm 0.12$ |
| ant-random | $113.34 \pm 2.78$ | $\mathbf{121 \pm 5.86}$ |

Table 1: Offline RL results obtained by combining PAGAR with RECOIL averaged over 4 seeds.

## 7.2 Continuous Control Tasks

We evaluate PAGAR-based IL on continuous control tasks in both online and offline RL settings, demonstrating its ability to improve IRL-based IL performance across different types of tasks.

**Benchmarks:** We use four continuous control environments from Mujoco. In the online RL setting, both protagonist and antagonist policies are permitted to explore the environment. In the offline RL setting, exploration by these policies is restricted. Especially, for offline RL we use the D4RL's 'expert' datasets as the expert demonstrations and the 'random' datasets as the offline suboptimal dataset. Policy performance is evaluated online in both settings by using the default reward function of the environment.

**Baselines:** We compare PAGAR-based IL against f-IRL Ni et al. [2021] in the online RL setting and compare with RECOIL Sikchi et al. [2024] in the offline RL setting. When comparing with f-IRL, we use f-IRL as the IRL algorithm in Algorithm 1. When comparing with RECOIL, as RECOIL does not directly learn the reward function but learns the Q and V functions, we develop another algorithm for the offline RL setting to combine PAGAR with RECOIL by explicitly learning a reward function and using the reward function and V function to represent the Q function. The details can be found in Appendix B.4.

**Results:** As shown in Figure 4, PAGAR-based IL achieves equivalent performance to the baselines with fewer iterations. Furthermore, on the *Ant* and *Walker2d* tasks, Algorithm 1 matches the performance level of f-IRL using significantly less iterations. Additional results of PAGAR with GAIL and VAIL across other continuous control benchmarks are provided in Appendix C.3. Table 1 further shows that when combined with RECOIL, PAGAR-based IL achieves higher performance in most of the tasks than the baseline. These results demonstrate the broader applicability of PAGAR-based IL in both online and offline settings and its effectiveness across different types of environments, further reinforcing the robustness of our approach.

## 8 Conclusion

In this paper, we propose to prioritize task alignment over conventional data alignment in IRL-based IL by treating expert demonstrations as weak supervision signals to derive a set of candidate reward functions that align with the task rather than only with the data. Our PAGAR-based IL adopts an adversarial mechanism to train a policy with this set of reward functions. Experimental results demonstrate that our algorithm can mitigate reward misalignment in challenging environments. Our future work will focus on employing the PAGAR paradigm to other task alignment problems.

## Acknowledgment

This work was supported in part by the U.S. National Science Foundation under grant CCF-2340776.

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

# A Reward Design with PAGAR

This paper does not aim to resolve the ambiguity problem in IRL but provides a way to circumvent it so that reward ambiguity does not lead to reward misalignment in IRL-based IL. PAGAR, the semi-supervised reward design paradigm proposed in this paper, tackles this problem from the perspective of semi-supervised reward design. But the nature of PAGAR is distinct from IRL and IL: assume that a set of reward functions is available for some underlying task, where some of those reward functions align with the task while others are misaligned, PAGAR provides a solution for selecting reward functions to train a policy that successfully performs the task, without knowing which reward function aligns with the task. Our research demonstrates that policy training with PAGAR is equivalent to learning a policy to maximize an affine combination of utilities measured under a distribution of the reward functions in the reward function set. With this understanding of PAGAR, we integrate it with IL to illustrate its advantages.

## A.1 Motivation: Failures in IRL-Based IL

For readers' convenience, we put the theorem from Abel et al. [2021] here for reference.

**Theorem 5.** *Viano et al. [2021] If the demonstration environment has a dynamics function $\mathcal{P}_{demo}$ different from the dynamics function $\mathcal{P}$ in the learning environment, the performance gap between the policies $\pi_E$ and $\pi_{r^*}$ under the ground true reward function $r_E$ satisfies $\mid U_{r_E}(\pi_E) - U_{r_E}(\pi_{r^*}) \mid \leq \frac{2 \cdot \gamma \cdot \max\limits_{s,a} \mid r_E(s,a) \mid}{(1-\gamma)^2} \cdot \max\limits_{s,a} D_{TV}(\mathcal{P}(\cdot \mid s,a), \mathcal{P}_{demo}(\cdot \mid s,a)).$*

Then, we prove that a limited number of demonstrations can also lead to the performance gap as described in Theorem 5. Essentially, we can construct a demonstration dynamics $\mathcal{P}_{demo}$ such that all the state transition probability estimated in $E$ match $\mathcal{P}_{demo}$ with zero error. Then a similar performance gap as that caused by dynamics mismatch can be derived.

**Proposition 3.** *Assume that the expert demonstration contains $m$ state-action pairs by only selecting the optimal actions, i.e., select $\arg\max\limits_{a} \pi_E(a|s)$ at each $s$. Then $\mid U_{r_E}(\pi_E) - U_{r_E}(\pi_{r^*}) \mid \leq \frac{2 \cdot \cdot d \cdot \gamma \cdot \max\limits_{s,a} \mid r_E(s,a) \mid}{(1-\gamma)^2}$ where $Prob(d \geq \epsilon) \leq 2 \cdot \exp(-2 \cdot m \cdot \epsilon^2)$ for any $\epsilon \in [\underline{p}, 1]$ where $\underline{p} = \frac{1}{\gamma} \cdot \frac{\min\limits_{s,a} r_E(s,a) - \min\limits_{s} r_E(s, \arg\max\limits_{a} \pi_E(a|s))}{\max\limits_{s,a} r_E(s,a) - \min\limits_{s,a} r_E(s,a)}.$*

*Proof.* We translate the limited demonstration case into a dynamics mismatch case where we will leverage the demonstrated state transitions to construct a dynamics $\mathcal{P}_{demo}(\cdot \mid s,a)$ and compare it with $\mathcal{P}(\cdot \mid s,a)$.

Assume that for each state-action pair $s,a$ in $E$, i.e., $m_{s,a} \in [m]$, the $m_{s,a}$ transition instances are $(s,a,s_1), (s,a,s_2), \ldots, (s,a,s_{m_{s,a}})$. For each $\hat{s} \in \mathbb{S}$, we let $\mathcal{P}_{demo}(\hat{s} \mid s,a) = \frac{1}{m_{s,a}} \sum_{i=1}^{m_{s,a}} \mathcal{I}[s_i = \hat{s}]$.

Then, we can view $D_{TV}(\mathcal{P}(\cdot \mid s,a), \mathcal{P}_{demo}(\cdot \mid s,a)) = \frac{1}{2} ||\mathcal{P}(\cdot \mid s,a) - \mathcal{P}_{demo}(\cdot \mid s,a))||_1$ as a function of $m$ independent random variables, i.e., $(s_1, a_1), (s_2, a_2), \ldots, (s_{m_{s,a}}, a_{m_{s,a}})$, sampled from the $\mathbb{S}$ domain. If one of the variables, $s_i$, is changed from $\hat{s}$ to another state $\hat{s}' \neq \hat{s}$, $\mathcal{P}_{demo}(\hat{s} \mid s,a)$ and $\mathcal{P}_{demo}(\hat{s}' \mid s,a)$ should decrease and increase by $\frac{1}{m_{s,a}}$ respectively. Then, $D_{TV}(\mathcal{P}(\cdot \mid s,a), \mathcal{P}_{demo}(\cdot \mid s,a))$ changes at most by $\frac{1}{m_{s,a}}$.

We can also view the $D_{TV}$ at each $(s,a) \in E$ as functions of the $m$ sampled state-action pairs in $E$, with a little abuse of notations, denoted as $(s_1, a_1), (s_2, a_2), \ldots, (s_m, a_m)$ by flattening all the demonstrated trajectories. If a state-action pair, $(s_i, a_i)$, in one of the trajectories, is changed from $(\hat{s}, \hat{a})$ to another state $(\hat{s}', \hat{a}')$, then for the state-action $(s_{i-1}, a_{i-1})$ that precedes $s_i$ if $s_i$ is not the initial state, $D_{TV}(\mathcal{P}(\cdot \mid s,a), \mathcal{P}_{demo}(\cdot \mid s,a))$ changes at most by $\frac{1}{m_{s_{i-1},a_{i-1}}}$ which can be as large as 1 since it is possible that $m_{s_{i-1},a_{i-1}} = 1$. Also, $D_{TV}(\mathcal{P}(\cdot \mid \hat{s}, \hat{a}), \mathcal{P}_{demo}(\cdot \mid \hat{s}, \hat{a}))$ and $D_{TV}(\mathcal{P}(\cdot \mid \hat{s}', \hat{a}'), \mathcal{P}_{demo}(\cdot \mid \hat{s}', \hat{a}'))$ change at most by $\frac{1}{m_{\hat{s},\hat{a}}}$ and $\frac{1}{m_{\hat{s}',\hat{a}'}}$. Note that if $m_{\hat{s}',\hat{a}'} = 0$, i.e., $(\hat{s}', \hat{a}') \notin E$, changing $\hat{s}, \hat{a}$ into $\hat{s}', \hat{a}'$ equivalently leads to taking $D_{TV}(\mathcal{P}(\cdot \mid \hat{s}, \hat{a}, \mathcal{P}_{demno}(\cdot \mid \hat{s}', \hat{a}'))$ into consideration when computing

$\max_{s,a} D_{TV}(\mathcal{P}(\cdot \mid s,a), \mathcal{P}_{demo}(\cdot \mid s,a))$. And $D_{TV}(\mathcal{P}(\cdot \mid \hat{s},\hat{a}, \mathcal{P}_{demno}(\cdot \mid \hat{s}',\hat{a}')) \leq 1 - \frac{\mathcal{P}(s_{i+1}|\hat{s}',\hat{a}'))}{2}$ where $s_{i+1}$ is the state that succeeds $s_i$ in the demonstrated trajectroy. Therefore, if changing a state-action pair in $E$ into another state-action pair, $\max_{(s,a)\in E} D_{TV}(\mathcal{P}(\cdot \mid s,a), \mathcal{P}_{demo}(\cdot \mid s,a))$ can be changed as large as by 1.

According to McDiarmid's Inequality, $Prob(|\max_{(s,a)\in E} D_{TV}(\mathcal{P}(\cdot \mid s,a), \mathcal{P}_{demo}(\cdot \mid s,a)) -$

$\mathbb{E}[\max_{(s,a)\in E} D_{TV}(\mathcal{P}(\cdot \mid s,a), \mathcal{P}_{demo}(\cdot \mid s,a))]| \geq \epsilon) \leq 2 \cdot \exp(-\frac{2\cdot\epsilon^2}{\sum_{i=1}^{m}(1)^2}) = 2 \cdot \exp(-\frac{2\epsilon^2}{m})$.
Note that $\mathbb{E}[\max_{s,a} D_{TV}(\mathcal{P}(\cdot \mid s,a), \mathcal{P}_{demo}(\cdot \mid s,a))] = 0$ since the estimation is un-biased. Hence,

$Prob(\max_{(s,a)\in E} D_{TV}(\mathcal{P}(\cdot \mid s,a), \mathcal{P}_{demo}(\cdot \mid s,a)) \geq \epsilon) \leq 2 \cdot \exp(-\frac{2\epsilon^2}{m})$ for $(s,a) \in E$.

For state-action pairs that are outside of $E$, we need to construct $\mathcal{P}_{demo}$ at those state-action pairs such that the action selection in $E$ is always optimal. To start with, we let $\mathcal{P}_{demo}(\cdot \mid a,s) \equiv \mathcal{P}(\cdot \mid a,s)$ for any state-action pairs that do not appear in $E$. For some state-action pairs, the states are not reached in $E$. We can disregard whether $\pi_E$ maintains optimal at those states and focus on the states that do appear in $E$. By denoting the Q-value of $\pi_E$ under $\mathcal{P}_{demo}$ as $Q_{\pi_E}^{demo}$, we need to make sure that the $Q_{\pi_E}^{demo}(s,a')$ for $a' \neq \arg\max_a \pi_E(a|s)$ is no greater than $Q_{\pi_E}^{demo}(s, \arg\max_a \pi_E(a|s))$. Since the state-actions in $E$ may have extremely low rewards stochastic nature of $\mathcal{P}$, we consider the worst case Q-value for $a^* = \arg\max_a \pi_E(a|s)$ i.e., $Q_{\pi_E}^{demo}(s,a^*) \geq \frac{1}{1-\gamma} \cdot \min_s r_E(s, \arg\max_a \pi_E(a|s))$. Meanwhile, for $a' \neq \arg\max_a \pi_E(a|s)$, we consider the best-case Q-value, i.e., $Q_{\pi_E}^{demo}(s,a') \leq \frac{1}{1-\gamma} \cdot \max_{(s,a)} r_E(s,a)$. We add a dummy, absorbing state $\underline{s}$ that is unreachable from any state-action under the dynamics $\mathcal{P}$ and also unreachable from any demonstrated state-action under the constructed dynamics $\mathcal{P}_{demo}$. But we let $\mathcal{P}_{demo}(\underline{s}|s,a') > 0$ if $Q_{\pi_E}^{demo}(s,a') > Q_{\pi_E}^{demo}(s,a^*)$. In other words, we add probability density on transitioning from such $(s,a')$ to $\underline{s}$ so that $Q_{\pi_E}^{demo}(s,a')$ drops below $Q_{\pi_E}^{demo}(s,a^*)$. We denote this density as $\underline{p}$. Then $\mathcal{P}_{demo}(\underline{s}|s,a') = \underline{p}$ and $\mathcal{P}_{demo}(s'|s,a') = (1-\underline{p}) \cdot \mathcal{P}(s'|s,a')$ for any other $s' \in \mathbb{S}$. As a result, $D_{TV}(\mathcal{P}(\cdot \mid s,a'), \mathcal{P}_{demo}(\cdot \mid s,a') = \underline{p}$. Note that adding $\underline{s}$ does not affect the Q-values of the state-action pairs that appear in $E$. Then we aim to find the $\underline{p}$ to ensure $Q_{\pi_E}^{demo}(s,a') \leq Q_{\pi_E}^{demo}(s,a^*)$. Considering the worst-case, $\underline{p} \cdot (\max_{s,a} r_E(s,a) +$

$\frac{\gamma}{1-\gamma} \cdot \min_{s,a} r_E(s,a)) + (1-\underline{p}) \cdot \frac{1}{1-\gamma} \cdot \max_{s,a} r_E(s,a) \leq \frac{1}{1-\gamma} \cdot \min_s r_E(s, \arg\max_a \pi_E(a|s))$ gives

$\underline{p} \leq \frac{1}{\gamma} \cdot \frac{\max_{s,a} r_E(s,a) - \min_s r_E(s, \arg\max_a \pi_E(a|s))}{\max_{s,a} r_E(s,a) - \min_{s,a} r_E(s,a)}$.

Combining the analysis on the state-action pairs that appear in $E$ and not in $E$, we can conclude that for $\epsilon > \underline{p}$, we can extend the confidence bound $Prob(\max_{(s,a)\in E} D_{TV}(\mathcal{P}(\cdot \mid s,a), \mathcal{P}_{demo}(\cdot \mid s,a)) \geq$

$\epsilon) \leq 2 \cdot \exp(-\frac{2\epsilon^2}{m})$ for $(s,a) \in E$ from $(s,a) \in E$ to all state-action pairs in $\mathbb{S} \times \mathbb{A}$. By using a variable $d$ to represent $\max_{s,a} D_{TV}(\mathcal{P}(\cdot \mid s,a) - \mathcal{P}_{demo}(\cdot \mid s,a))$, we can use the conclusion drawn from Theorem 5. Note that adding the dummy $\underline{s}$ does not affect the $\max_{s,a} |r_E(s,a)|$ since we let $r_E(\underline{s},a) \equiv \min_{s,a} r_E(s,a)$. Our proof is complete. $\qquad\square$

## A.2 Task-Reward Alignment

In this section, we provide proof of the properties of the task-reward alignment concept that we define in the main text. For readers' convenience, we include our definition of *task* for reference.

**Definition** 1 (**Task**) Given the policy hypothesis set $\Pi$, a **task** $(\Pi, \preceq_{task}, \Pi_{acc})$ is specified by a partial order over $\Pi$ and a non-empty set of acceptable policies $\Pi_{acc} \subseteq \Pi$ such that $\forall \pi_1 \in \Pi_{acc}$ and $\forall \pi_2 \notin \Pi_{acc}, \pi_2 \preceq_{task} \pi_1$ always hold.

We have defined in the main text that $\underline{U}_r := \min_{\pi \in \Pi_{acc}} U_r(\pi)$ is the lowest utility achieved by any acceptable policies under $r$, and $\overline{U}_r := \max_{\pi \in \Pi} U_r(\pi)$ s.t. $\forall \pi_1, \pi_2 \in \Pi, U_r(\pi_1) \leq U_r(\pi) \leq U_r(\pi_2) \Rightarrow (\pi_1 \preceq_{task} \pi) \wedge (\pi_1 \preceq_{task} \pi_2)$ is the highest utility threshold such that any policy achieving a utility

higher than $\overline{U}_r$ also has a higher order than those of which utilities are lower than $\overline{U}_r$. We call a reward function $r$ a $(\overline{U}_r, \underline{U}_r)$-aligned with the task.

**Lemma 1.** *For any task-aligned reward function $r^+$, $\overline{U}_{r+} \geq \underline{U}_{r+}$.*

*Proof.* If $\overline{U}_{r+} < \underline{U}_{r+}$, then for any policy $\pi$ with $U_r(\pi) \in [\overline{U}_{r+}, \underline{U}_{r+})$, it is guaranteed that $\pi \in \Pi \backslash \Pi_{acc}$ by definition of $\underline{U}_{r+}$. Then, $\underline{U}_{r+}$ is a better solution for $\max_{\pi \in \Pi} U_r(\pi)$ s.t. $\forall \pi_1, \pi_2 \in \Pi, U_r(\pi_1) \leq U_r(\pi) \leq U_r(\pi_2) \Rightarrow (\pi_1 \preceq_{task} \pi) \wedge (\pi_1 \preceq_{task} \pi_2)$ than $\overline{U}_{r+}$. Hence, $\overline{U}_{r+} \geq \underline{U}_{r+}$ must hold. $\square$

**Lemma 2.** *If $\pi_1 \preceq_{task} \pi_2 \Leftrightarrow U_r(\pi_2) \leq U_r(\pi_1)$ holds for any $\pi_1, \pi_2 \in \Pi$, then $\overline{U}_r = \underline{U}_r = \min_{\pi \in \Pi} U_r(\pi)$.*

*Proof.* Since the policy that has the highest order achieves the lowest utility $\min_{\pi \in \Pi} U_r(\pi)$, no other acceptable policy can achieve even lower utility. Hence, by definition $\underline{U}_r = \min_{\pi \in \Pi} U_r(\pi)$.

$\square$

Another example is that when optimal policy under $r$ has the lowest order in terms of $\preceq_{task}$, and the highest-order policy has the lowest utility under $r$, $\overline{U}_r = U_r$. In those extreme cases, the reward function can be considered completely misaligned. In the non-extreme cases, if a reward function $r$ exhibits misalignment with the task – the threshold $\overline{U}_r$ is lower than the utility of an unacceptable policy – $\overline{U}_r$ then must be also lower than the utilities of all the acceptable policies, which forms a lower-bound for the size of $\underline{U}_r$.

**Lemma 3.** *For any reward function $r$, if $\exists \pi \notin \Pi_{acc}, U_r(\pi) \geq \overline{U}_r$, then $\overline{U}_r \leq \underline{U}_r$.*

*Proof.* Since $\pi \notin \Pi_{acc}$ and $U_r(\pi) \geq \overline{U}_r$, then $\forall \pi' \in \Pi_{acc}, U_r(\pi') \geq \overline{U}_r$ must be true, otherwise $\exists \pi' \in \Pi_{acc}, \pi' \preceq_{task} \pi$, which contradicts the definition of $\Pi_{acc}$ and $\overline{U}_r$. $\square$

**Lemma 4.** *If a reward function $r$ has $\overline{U}_r \leq \underline{U}_r$, $r$ is a task-misaligned reward function.*

*Proof.* If $\overline{U}_r \leq \underline{U}_r$, then there must exists two policies $\pi_1, \pi_2$ where $\pi_1 \in \Pi \backslash \Pi_{acc}, U_r(\pi_1) \in [\overline{U}_r, \underline{U}_r]$ and $U_r(\pi_2) \geq \underline{U}_r \wedge \pi_2 \preceq_{task} \pi_1$. Such $\pi_2$ must not be an acceptable policy. Otherwise, it contradicts the definition of $\Pi_{acc}$. Hence, it must be unacceptable, leading to $r$ being a task-misaligned reward function. $\square$

**Proposition 1** For any two reward functions $r_1, r_2$, if $\{\pi : U_{r_1}(\pi) \geq \overline{U}_{r_1}\} \subseteq \{\pi : U_{r_2}(\pi) \geq \overline{U}_{r_2}\}$, then there must exist a $\pi_1 \in \{\pi : U_{r_1}(\pi) \geq \overline{U}_{r_1}\}$ and a $\pi_2 \in \{\pi : U_{r_2}(\pi) \geq \overline{U}_{r_2}\}$ that satisfy $U_{r_1}(\pi_2) \leq U_{r_1}(\pi_1)$ and $\pi_2 \preceq_{task} \pi_1$ while $U_{r_2}(\pi_1) \leq U_{r_2}(\pi_2)$.

*Proof.* According to the definition of $\overline{U}_{r_1}$, $\forall \pi_1 \in \{\pi : U_{r_2}(\pi) \geq \overline{U}_{r_1}\}$ and $\forall \pi_2 \in \{\pi : U_{r_2}(\pi) \geq \overline{U}_{r_2}\} / \{\pi : U_{r_1}(\pi) \geq \overline{U}_{r_1}\}$, $U_{r_1}(\pi_2) \leq U_{r_1}(\pi_1)$ and $\pi_2 \preceq_{task} \pi_1$ must be true. Furthermore, if for all pairs of $\pi_2 \in \{\pi : U_{r_2}(\pi) \geq \overline{U}_{r_2}\} / \{\pi : U_{r_1}(\pi) \geq \overline{U}_{r_1}\}$ and $\pi_1 \in \{\pi : U_{r_1}(\pi) \geq \overline{U}_{r_1}\}$, $U_{r_2}(\pi_1) > U_{r_2}(\pi_2)$ is true, then $\{U_{r_2}(\pi) \mid U_{r_1}(\pi) \geq \overline{U}_{r_1}\} \subset [\overline{U}_{r_2}, \max_{\pi \in \Pi} U_{r_2}(\pi)]$ is a smaller non-empty interval than $[\overline{U}_{r_2}, \max_{\pi \in \Pi} U_{r_2}(\pi)]$, contradicting the fact that $\overline{U}_{r_2}$ is the highest utility threshold under $r_2$. Hence, there must exist a $\pi_2 \in \{\pi : U_{r_2}(\pi) \geq \overline{U}_{r_2}\} / \{\pi : U_{r_1}(\pi) \geq \overline{U}_{r_1}\}$ and a $\pi_1 \in \{\pi : U_{r_1}(\pi) \geq \overline{U}_{r_1}\}$ that satisfy $U_{r_1}(\pi_2) \leq U_{r_1}(\pi_1)$ and $U_{r_2}(\pi_1) \leq U_{r_2}(\pi_2)$ while $\pi_2 \preceq_{task} \pi_1$ as aforementioned. $\square$

Note that from now on, we use the notation $r^+$ to denote *task-aligned reward functions* and $r^-$ to denote *task-misaligned reward functions* for short. Furthermore, if a reward function $r$ satisfies $U_r(\pi_1) \leq U_r(\pi_2) \Leftrightarrow \pi_1 \preceq_{task} \pi_2$, we call this reward function the *ground true reward function*, and denote it as $r_{task}$ for simplicity. Apparently, any $r_{task}$ is a task-aligned reward function, and it has the most trivial misalignment.

**Lemma 5.** $\overline{U}_{r_{task}} = \max_{\pi \in \Pi} U_{r_{task}}(\pi)$

*Proof.* By definition, $\arg \max_{\pi \in \Pi} U_{r_{task}}(\pi)$ has higher order and higher utility than any other policies.
$\qquad\square$

**Lemma 6.** *If $\pi_{r_{task}}$ is optimal under $r_{task}$, for any reward function $r$, $U_r(\pi_{r_{task}}) \geq \overline{U}_r$.*

*Proof.* If $U_r(\Pi_{acc}) < \overline{U}_r$, then there exists a $\pi$ with its utility $U_r(\pi) \geq \overline{U}_r$, thus $\pi_{r_{task}} \preceq_{task} \pi$, which contradicts the assumption that $\pi_{r_{task}}$ is the highest-order policy. $\qquad\square$

Furthermore, we have the following property.

**Theorem** 1 Let $\mathcal{I}$ be an indicator function. For any $k \geq \left\{ \min_{r^+} \sum_{\pi \in \Pi} \mathcal{I}\{U_{r^+}(\pi) \geq U_{r^+}(\pi_E)\} \right\}$, if $\pi^*$ satisfies $\left\{ \sum_{\pi \in \Pi} \mathcal{I}\{U_r(\pi) \geq U_r(\pi^*)\} \right\} < |\Pi_{acc}|$ for all $r \in R_{E,k} := \left\{ r \mid \sum_{\pi \in \Pi} \mathcal{I}\{U_r(\pi) \geq U_r(\pi_E)\} \leq k \right\}$, then $\pi^*$ is an acceptable policy, i.e., $\pi^* \in \Pi_{acc}$. Additionally, if $k < |\Pi_{acc}|$, such an acceptable policy $\pi^*$ is guaranteed to exist.

*Proof.* Since $\pi_E$ is an acceptable policy, there must be at least one task-aligned reward function $r^+$ that satisfies $|\{\pi : U_r(\pi) \geq U_r(\pi_E)\}| \leq k$ since $k \geq \min_{r^+} |\{\pi : U_{r^+}(\pi) \geq U_{r^+}(\pi_E)\}|$. The greater $k$ is, the more task-aligned reward functions tend to be included. If $\pi^*$ achieves $|\{\pi : U_{r^+}(\pi) \geq U_{r^+}(\pi^*)\}| < |\Pi_{acc}|$ under any task-aligned reward function $r^+$, $\pi^*$ must be acceptable policy. Because if $\pi^*$ is unacceptable, there must be an acceptable policy performing worse than the unacceptable $\pi^*$ under $r^+$, contradicting the definition of *task-aligned reward function*. Hence, $\pi^*$ must be acceptable policy. Furthermore, for any $k \in [\min_{r^+} \sum_{\pi \in \Pi} \mathcal{I}\{U_{r^+}(\pi) \geq U_{r^+}(\pi_E)\}, |\Pi_{acc}|)$, the policy $\pi_E$ itself satisfied $\sum_{\pi \in \Pi} \mathcal{I}\{U_r(\pi) \geq U_r(\pi_E)\} < |\Pi_{acc}|$ for all $r \in R_{E,k}$, which guarantees the existence of a feasible $\pi^*$. $\qquad\square$

### A.3 Semi-supervised Reward Design

Designing a reward function can be thought as deciding an ordering of policies. We adopt a concept, called *total domination*, from unsupervised environment design Dennis et al. [2020], and re-interpret this concept in the context of reward design. In this paper, we suppose that the function $U_r(\pi)$ is given to measure the performance of a policy and it does not have to be the utility function. While the measurement of policy performance can vary depending on the free variable $r$, *total dominance* can be viewed as an invariance regardless of such dependency.

**Definition 5** (Total Domination). A policy, $\pi_1$, is totally dominated by some policy $\pi_2$ w.r.t a reward function set $R$, if for every pair of reward functions $r_1, r_2 \in R$, $U_{r_1}(\pi_1) < U_{r_2}(\pi_2)$.

If $\pi_1$ totally dominate $\pi_2$ w.r.t $R$, $\pi_2$ can be regarded as being unconditionally better than $\pi_1$. In other words, the two sets $\{U_r(\pi_1) \mid r \in R\}$ and $\{U_r(\pi_2) \mid r \in R\}$ are disjoint, such that $\sup\{U_r(\pi_1) \mid r \in R\} < \inf\{U_r(\pi_2) \mid r \in R\}$. Conversely, if a policy $\pi$ is not totally dominated by any other policy, it indicates that for any other policy, say $\pi_2$, $\sup\{U_r(\pi_1) \mid r \in R\} \geq \inf\{U_r(\pi_2) \mid r \in R\}$.

**Definition 6.** A reward function set $R$ aligns with an ordering $\prec_R$ among policies such that $\pi_1 \prec_R \pi_2$ if and only if $\pi_1$ is totally dominated by $\pi_2$ w.r.t. $R$.

Especially, designing a reward function $r$ is to establish an ordering $\prec_{\{r\}}$ among policies. Total domination can be extended to policy-conditioned reward design, where the reward function $r$ is selected by following a decision rule $\omega(\pi)$ such that $\sum_{r \in R} \omega(\pi)(r) = 1$. We let $U_\omega(\pi) = \sum_{r \in R} \omega(\pi)(r) \cdot U_r(\pi)$ be an affine combination of $U_r(\pi)$'s with its coefficients specified by $\omega(\pi)$.

**Definition 7.** A policy conditioned decision rule $\omega$ is said to prefer a policy $\pi_1$ to another policy $\pi_2$, which is notated as $\pi_1 \prec^\omega \pi_2$, if and only if $U_\omega(\pi_1) < U_\omega(\pi_2)$.

Making a decision rule for selecting reward functions from a reward function set to respect the total dominance w.r.t this reward function set is an unsupervised learning problem, where no additional external supervision is provided. If considering expert demonstrations as a form of supervision

and using it to constrain the set $R_E$ of reward function via IRL, the reward design becomes semi-supervised.

## A.4 Solution to the MinimaxRegret

Without loss of generality, we use $R$ instead of $R_{E,\delta}$ in our subsequent analysis because solving $MinimaxRegret(R)$ does not depend on whether there are constraints for $R$. In order to show such an equivalence, we follow the same routine as in Dennis et al. [2020], and start by introducing the concept of *weakly total domination*.

**Definition 8** (Weakly Total Domination). A policy $\pi_1$ is *weakly totally dominated* w.r.t a reward function set $R$ by some policy $\pi_2$ if and only if for any pair of reward function $r_1, r_2 \in R$, $U_{r_1}(\pi_1) \leq U_{r_2}(\pi_2)$.

Note that a policy $\pi$ being totally dominated by any other policy is a sufficient but not necessary condition for $\pi$ being weakly totally dominated by some other policy. A policy $\pi_1$ being weakly totally dominated by a policy $\pi_2$ implies that $\sup\{U_r(\pi_1) \mid r \in R\} \leq \inf\{U_r(\pi_2) \mid r \in R\}$. We assume that there does not exist a policy $\pi$ that weakly totally dominates itself, which could happen if and only if $U_r(\pi)$ is a constant. We formalize this assumption as the following.

**Assumption 1.** For the given reward set $R$ and policy set $\Pi$, there does not exist a policy $\pi$ such that for any two reward functions $r_1, r_2 \in R$, $U_{r_1}(\pi) = U_{r_2}(\pi)$.

This assumption makes weak total domination a non-reflexive relation. It is obvious that weak total domination is transitive and asymmetric. Now we show that successive weak total domination will lead to total domination.

**Lemma 7.** *for any three policies $\pi_1, \pi_2, \pi_3 \in \Pi$, if $\pi_1$ is weakly totally dominated by $\pi_2$, $\pi_2$ is weakly totally dominated by $\pi_3$, then $\pi_3$ totally dominates $\pi_1$.*

*Proof.* According to the definition of weak total domination, $\max_{r \in R} U_r(\pi_1) \leq \min_{r \in R} U_r(\pi_2)$ and $\max_{r \in R} U_r(\pi_2) \leq \min_{r \in R} U_r(\pi_3)$. If $\pi_1$ is weakly totally dominated but not totally dominated by $\pi_3$, then $\max_{r \in R} U_r(\pi_1) = \min_{r \in R} U_r(\pi_3)$ must be true. However, it implies $\min_{r \in R} U_r(\pi_2) = \max_{r \in R} U_r(\pi_2)$, which violates Assumption 1. We finish the proof. $\square$

**Lemma 8.** *For the set $\Pi_{\neg wtd} \subseteq \Pi$ of policies that are not weakly totally dominated by any other policy in the whole set of policies w.r.t a reward function set $R$, there exists a range $U \subseteq \mathbb{R}$ such that for any policy $\pi \in \Pi_{\neg wtd}$, $U \subseteq [\min_{r \in R} U_r(\pi), \max_{r \in R} U_r(\pi)]$.*

*Proof.* For any two policies $\pi_1, \pi_2 \in \Pi_{\neg wtd}$, it cannot be true that $\max_{r \in R} U_r(\pi_1) = \min_{r \in R} U_r(\pi_2)$ nor $\min_{r \in R} U_r(\pi_1) = \max_{r \in R} U_r(\pi_2)$, because otherwise one of the policies weakly totally dominates the other. Without loss of generalization, we assume that $\max_{r \in R} U_r(\pi_1) > \min_{r \in R} U_r(\pi_2)$. In this case, $\max_{r \in R} U_r(\pi_2) > \min_{r \in R} U_r(\pi_1)$ must also be true, otherwise $\pi_1$ weakly totally dominates $\pi_2$. Inductively, $\min_{\pi \in \Pi_{\neg wtd}} \max_{r \in R} U_r(\pi) > \max_{\pi \in \Pi_{\neg wtd}} \min_{r \in R} U_r(\pi)$. Letting $ub = \min_{\pi \in \Pi_{\neg wtd}} \max_{r \in R} U_r(\pi)$ and $lb = \max_{\pi \in \Pi_{\neg wtd}} \min_{r \in R} U_r(\pi)$, any $U \subseteq [lb, ub]$ shall support the assertion. We finish the proof. $\square$

**Lemma 9.** *For a reward function set $R$, if a policy $\pi \in \Pi$ is weakly totally dominated by some other policy in $\Pi$ and there exists a subset $\Pi_{\neg wtd} \subseteq \Pi$ of policies that are not weakly totally dominated by any other policy in $\pi$, then $\max_{r \in R} U_r(\pi) < \min_{\pi' \in \Pi_{\neg wtd}} \max_{r \in R} U_r(\pi')$*

*Proof.* If $\pi_1$ is weakly totally dominated by a policy $\pi_2 \in \Pi$, then $\min_{r \in R} U_r(\pi_2) = \max_{r \in R} U_r(\pi)$. If $\max_{r \in R} U_r(\pi) \geq \min_{\pi' \in \Pi_{\neg wtd}} \max_{r \in R} U_r(\pi')$, then $\min_{r \in R} U_r(\pi_2) \geq \min_{\pi' \in \Pi_{\neg wtd}} \max_{r \in R} U_r(\pi')$, making at least one of the policies in $\Pi_{\neg wtd}$ being weakly totally dominated by $\pi_2$. Hence, $\max_{r \in R} U_r(\pi) < \min_{\pi' \in \Pi_{\neg wtd}} \max_{r \in R} U_r(\pi')$ must be true. $\square$

Given a policy $\pi$ and a reward function $r$, the regret is represented as Eq.8

$$Regret(\pi, r) \quad := \quad \max_{\pi'} U_r(\pi') - U_r(\pi) \tag{8}$$

Then we represent the $MinimaxRegret(R)$ problem in Eq.9.

$$MinimaxRegret(R) \quad := \quad \arg\min_{\pi \in \Pi} \left\{ \max_{r \in R} Regret(\pi, r) \right\} \tag{9}$$

We denote as $r_\pi^* \in R$ the reward function that maximizes $U_r(\pi)$ among all the $r$'s that achieve the maximization in Eq.9. Formally,

$$r_\pi^* \quad \in \quad \arg\max_{r \in R} U_r(\pi) \qquad s.t. \ r \in \arg\max_{r' \in R} Regret(\pi, r') \tag{10}$$

Then $MinimaxRegret$ can be defined as minimizing the worst-case regret as in Eq.9. Next, we want to show that for some decision rule $\omega$, the set of optimal policies that maximize $U_\omega$ are the solutions to $MinimaxRegret(R)$. Formally,

$$MinimaxRegret(R) = \arg\max_{\pi \in \Pi} U_\omega(\pi) \tag{11}$$

We design $\omega$ by letting $\omega(\pi) := \overline{\omega}(\pi) \cdot \delta_{r_\pi^*} + (1 - \overline{\omega}(\pi)) \cdot \mathcal{R}_\pi$ where $\mathcal{R}_\pi \in \Delta(R)$ is a policy conditioned distribution over reward functions, $\delta_{r_\pi^*}$ be a delta distribution centered at $r_\pi^*$, and $\overline{\omega}(\pi)$ is a coefficient. We show how to design $\mathcal{R}$ by using the following lemma.

**Lemma 10.** *Given that the reward function set is $R$, there exists a decision rule $\mathcal{R} : \Pi \to \Delta(R)$ which guarantees that: 1) for any policy $\pi$ that is not weakly totally dominated by any other policy in $\Pi$, i.e., $\pi \in \Pi_{\neg wtd} \subseteq \Pi$, $U_\mathcal{R}(\pi) \equiv c$ where $c = \max_{\pi' \in \Pi_{\neg wtd}} \min_{r \in R} U_r(\pi')$; 2) for any $\pi$ that is weakly totally dominated by some policy but not totally dominated by any policy, $U_\mathcal{R}(\pi) = \max_{r \in R} U_r(\pi)$; 3) if $\pi$ is totally dominated by some other policy, $\overline{\omega}(\pi)$ is a uniform distribution.*

*Proof.* Since the description of $\mathcal{R}$ for the policies in condition 2) and 3) are self-explanatory, we omit the discussion on them. For the none weakly totally dominated policies in condition 1), having a constant $U_\mathcal{R}(\pi) \equiv c$ is possible if and only if for any policy $\pi \in \Pi_{\neg wed}$, $c \in [\min_{r \in R} U_r(\pi'), \max_{r \in R} U_r(\pi')]$. As mentioned in the proof of Lemma 8, $c$ can exist within $[\min_{r \in R} U_r(\pi), \max_{r \in R} U_r(\pi)]$. Hence, $c = \max_{\pi' \in \Pi_{\neg wtd}} \min_{r \in R} U_r(\pi')$ is a valid assignment. $\qquad \square$

Then by letting $\overline{\omega}(\pi) := \frac{Regret(\pi, r_\pi^*)}{c - U_{r_\pi^*}(\pi)}$, we have the following theorem.

**Theorem 6.** *By letting $\omega(\pi) := \overline{\omega}(\pi) \cdot \delta_{r_\pi^*} + (1 - \overline{\omega}(\pi)) \cdot \mathcal{R}_\pi$ with $\overline{\omega}(\pi) := \frac{Regret(\pi, r_\pi^*)}{c - U_{r_\pi^*}(\pi)}$ and any $\mathcal{R}$ that satisfies Lemma 10,*

$$MinimaxRegret(R) = \arg\max_{\pi \in \Pi} U_\omega(\pi) \tag{12}$$

*Proof.* If a policy $\pi \in \Pi$ is totally dominated by some other policy, since there exists another policy with larger $U_\omega$, $\pi$ cannot be a solution to $\arg\max_{\pi \in \Pi} U_\omega(\pi)$. Hence, there is no need for further discussion on totally dominated policies. We discuss the none weakly totally dominated policies and the weakly totally dominated but not totally dominated policies (shortened to "weakly totally

dominated" from now on) respectively. First we expand $\arg\max_{\pi\in\Pi} U_\omega(\pi)$ as in Eq.13.

$$
\begin{aligned}
&\arg\max_{\pi\in\Pi} U_\omega(\pi) \\
=\ &\arg\max_{\pi\in\Pi} \sum_{r\in R}\omega(\pi)(r)\cdot U_r(\pi) \\
=\ &\arg\max_{\pi\in\Pi} \frac{Regret(\pi, r_\pi^*)\cdot U_{r_\pi^*}(\pi) + (U_{\mathcal{R}}(\pi) - U_{r_\pi^*}(\pi) - Regret(\pi, r_\pi^*))\cdot U_{\mathcal{R}}(\pi)}{c - U_{r_\pi^*}(\pi)} \\
=\ &\arg\max_{\pi\in\Pi} \frac{(U_{\mathcal{R}}(\pi) - U_{r_\pi^*}(\pi))\cdot U_{\mathcal{R}}(\pi) - (U_{\mathcal{R}}(\pi) - U_{r_\pi^*}(\pi))\cdot Regret(\pi, r_\pi^*))}{c - U_{r_\pi^*}(\pi)} \\
=\ &\arg\max_{\pi\in\Pi} \frac{U_{\mathcal{R}}(\pi) - U_{r_\pi^*}(\pi)}{c - U_{r_\pi^*}(\pi)}\cdot U_{\mathcal{R}}(\pi) - Regret(\pi, r_\pi^*) \qquad (13)
\end{aligned}
$$

1) For the none weakly totally dominated policies, since by design $U_{\mathcal{R}}\equiv c$, Eq.13 is equivalent to $\arg\max_{\pi\in\Pi_1} -Regret(\pi, r_\pi^*)$ which exactly equals $MinimaxRegret(R)$. Hence, the equivalence holds among the none weakly totally dominated policies. Furthermore, if a none weakly totally dominated policy $\pi\in\Pi_{\neg wtd}$ achieves optimality in $MinimaxRegret(R)$, its $U_\omega(\pi)$ is also no less than any weakly totally dominated policy. Because according to Lemma 9, for any weakly totally dominated policy $\pi_1$, its $U_{\mathcal{R}}(\pi_1)\le c$, hence $\frac{U_{\mathcal{R}}(\pi)-U_{r_\pi^*}(\pi)}{c-U_{r_\pi^*}(\pi)}\cdot U_{\mathcal{R}}(\pi_1)\le c$. Since $Regret(\pi, r_\pi^*)\le Regret(\pi_1, r_{\pi_1}^*)$, $U_\omega(\pi)\ge U_\omega(\pi_1)$. Therefore, we can can assert that if a none weakly totally dominated policy $\pi$ is a solution to $MinimaxRegret(R)$, it is also a solution to $\arg\max_{\pi\in\Pi} U_\omega(\pi)$. Additionally, to prove that if a none weakly totally dominated policy $\pi$ is a solution to $\arg\max_{\pi'\in\Pi} U_\omega(\pi')$, it is also a solution to $MinimaxRegret(R)$, it is only necessary to prove that $\pi$ achieve no larger regret than all the weakly totally dominated policies. But we delay the proof to 2).

2) If a policy $\pi$ is weakly totally dominated and is a solution to $MinimaxRegret(R)$, we show that it is also a solution to $\arg\max_{\pi\in\Pi} U_\omega(\pi)$, i.e., its $U_\omega(\pi)$ is no less than that of any other policy.

We start by comparing with non weakly totally dominated policy. for any weakly totally dominated policy $\pi_1\in MinimaxRegret(R)$, it must hold true that $Regret(\pi_1, r_{\pi_1}^*)\le Regret(\pi_2, r_{\pi_2}^*)$ for any $\pi_2\in\Pi$ that weakly totally dominates $\pi_1$. However, it also holds that $Regret(\pi_2, r_{\pi_2}^*)\le Regret(\pi_1, r_{\pi_2}^*)$ due to the weak total domination. Therefore, $Regret(\pi_1, r_{\pi_1}^*) = Regret(\pi_2, r_{\pi_2}^*) = Regret(\pi_1, r_{\pi_2}^*)$, implying that $\pi_2$ is also a solution to $MinimaxRegret(R)$. It also implies that $U_{r_{\pi_2}^*}(\pi_1) = U_{r_{\pi_2}^*}(\pi_2)\ge U_{r_{\pi_1}^*}(\pi_1)$ due to the weak total domination. However, by definition $U_{r_{\pi_1}^*}(\pi_1)\ge U_{r_{\pi_2}^*}(\pi_1)$. Hence, $U_{r_{\pi_1}^*}(\pi_1) = U_{r_{\pi_2}^*}(\pi_1) = U_{r_{\pi_2}^*}(\pi_2)$ must hold. Now we discuss two possibilities: a) there exists another policy $\pi_3$ that weakly totally dominates $\pi_2$; b) there does not exist any other policy that weakly totally dominates $\pi_2$. First, condition a) cannot hold. Because inductively it can be derived $U_{r_{\pi_1}^*}(\pi_1) = U_{r_{\pi_2}^*}(\pi_1) = U_{r_{\pi_2}^*}(\pi_2) = U_{r_{\pi_3}^*}(\pi_3)$, while Lemma 7 indicates that $\pi_3$ totally dominates $\pi_1$, which is a contradiction. Hence, there does not exist any policy that weakly totally dominates $\pi_2$, meaning that condition b) is certain. We note that $U_{r_{\pi_1}^*}(\pi_1) = U_{r_{\pi_2}^*}(\pi_1) = U_{r_{\pi_2}^*}(\pi_2)$ and the weak total domination between $\pi_1, \pi_2$ imply that $r_{\pi_1}^*, r_{\pi_2}^*\in\arg\max_{r\in R} U_r(\pi_1)$, $r_{\pi_2}^*\in\arg\min_{r\in R} U_r(\pi_2)$, and thus $\min_{r\in R} U_r(\pi_2)\le \max_{\pi\in\Pi_{\neg wtd}}\min_{r\in R} U_r(\pi) = c$. Again, $\pi_1\in MinimaxRegret(R)$ makes $Regret(\pi_1, r_\pi^*)\le Regret(\pi_1, r_{\pi_1}^*)\le Regret(\pi, r_\pi^*)$ not only hold for $\pi = \pi_2$ but also for any other policy $\pi\in\Pi_{\neg wtd}$, then for any policy $\pi\in\Pi_{\neg wtd}$, $U_{r_\pi^*}(\pi_1)\ge U_{r_\pi^*}(\pi)\ge \min_{r\in R} U_r(\pi)$. Hence, $U_{r_\pi^*}(\pi_1)\ge \max_{\pi\in\Pi_{\neg wtd}}\min_{r\in R} U_r(\pi) = c$. Since $U_{r_\pi^*}(\pi_1) = \min_{r\in R} U_r(\pi_2)$ as aforementioned, $\min_{r\in R} U_r(\pi_2) > \max_{\pi\in\Pi_{\neg wtd}}\min_{r\in R} U_r(\pi)$ will cause a contradiction. Hence, $\min_{r\in R} U_r(\pi_2) = \max_{\pi\in\Pi_{\neg wtd}}\min_{r\in R} U_r(\pi) = c$. As a result, $U_{\mathcal{R}}(\pi) = U_{r_\pi^*}(\pi) = \max_{\pi'\in\Pi_{\neg wtd}}\min_{r\in R} U_r(\pi') = c$, and $U_\omega(\pi) = c - Regret(\pi, r_\pi^*)\ge \max_{\pi'\in\Pi_{\neg wtd}} c - Regret(\pi', r_{\pi'}^*) = \max_{\pi'\in\Pi_{\neg wtd}} U_\omega(\pi')$. In other words, if a weakly totally dominated policy $\pi$ is a solution to $MinimaxRegret(R)$, then its $U_\omega(\pi)$ is no less than that of any non weakly totally dominated policy. This also complete the proof at the end of

1), because if a none weakly totally dominated policy $\pi_1$ is a solution to $\arg\max_{\pi\in\Pi} U_\omega(\pi)$ but not a solution to $MinimaxRegret(R)$, then $Regret(\pi_1, r^*_{\pi_1}) > 0$ and a weakly totally dominated policy $\pi_2$ must be the solution to $MinimaxRegret(R)$. Then, $U_\omega(\pi_2) = c > c - Regret(\pi_1, r^*_{\pi_1}) = U_\omega(\pi_1)$, which, however, contradicts $\pi_1 \in \arg\max_{\pi\in\Pi} U_\omega(\pi)$.

It is obvious that a weakly totally dominated policy $\pi \in MinimaxRegret(R)$ has a $U_\omega(\pi)$ no less than any other weakly totally dominated policy. Because for any other weakly totally dominated policy $\pi_1$, $U_\mathcal{R}(\pi_1) \le c$ and $Regret(\pi_1, r^*_{\pi_1}) \le Regret(\pi, r^*_\pi)$, hence $U_\omega(\pi_1) \le U_\omega(\pi)$ according to Eq.13.

So far we have shown that if a weakly totally dominated policy $\pi$ is a solution to $MinimaxRegret(R)$, it is also a solution to $\arg\max_{\pi'\in\Pi} U_\omega(\pi')$. Next, we need to show that the reverse is also true, i.e., if a weakly totally dominated policy $\pi$ is a solution to $\arg\max_{\pi\in\Pi} U_\omega(\pi)$, it must also be a solution to $MinimaxRegret(R)$. In order to prove its truthfulness, we need to show that if $\pi \notin MinimaxRegret(R)$, whether there exists: a) a none weakly totally dominated policy $\pi_1$, or b) another weakly totally dominated policy $\pi_1$, such that $\pi_1 \in MinimaxRegret(R)$ and $U_\omega(\pi_1) \le U_\omega(\pi)$. If neither of the two policies exists, we can complete our proof. Since it has been proved in 1) that if a none weakly totally dominated policy achieves $MinimaxRegret(R)$, it also achieves $\arg\max_{\pi'\in\Pi} U_\omega(\pi')$, the policy described in condition a) does not exist. Hence, it is only necessary to prove that the policy in condition b) also does not exist.

If such weakly totally dominated policy $\pi_1$ exists, $\pi \notin MinimaxRegret(R)$ and $\pi_1 \in MinimaxRegret(R)$ indicates $Regret(\pi, r^*_\pi) > Regret(\pi_1, r^*_{\pi_1})$. Since $U_\omega(\pi_1) \ge U_\omega(\pi)$, according to Eq.13, $U_\omega(\pi_1) = c - Regret(\pi_1, r^*_{\pi_1}) \le U_\omega(\pi) = \frac{U_\mathcal{R}(\pi) - U_{r^*_\pi}(\pi)}{c - U_{r^*_\pi}(\pi)} \cdot U_\mathcal{R}(\pi) - Regret(\pi, r^*_\pi)$. Thus $\frac{U_\mathcal{R}(\pi) - U_{r^*_\pi}(\pi)}{c - U_{r^*_\pi}(\pi)}(\pi) \cdot U_\mathcal{R} \ge c + Regret(\pi, r^*_\pi) - Regret(\pi_1, r^*_{\pi_1}) > c$, which is impossible due to $U_\mathcal{R} \le c$. Therefore, such $\pi_1$ also does not exist. In fact, this can be reasoned from another perspective. If there exists a weakly totally dominated policy $\pi_1$ with $U_{r^*_{\pi_1}}(\pi_1) = c = U_{r^*_\pi}(\pi)$ but $\pi_1 \notin MinimaxRegret(R)$, then $Regret(\pi, r^*_\pi) > Regret(\pi_1, r^*_{\pi_1})$. It also indicates $\max_{\pi'\in\Pi} U_{r^*_\pi}(\pi') > \max_{\pi'\in\Pi} U_{r^*_{\pi_1}}(\pi')$. Meanwhile, $Regret(\pi_1, r^*_\pi) := \max_{\pi'\in\Pi} U_{r^*_\pi}(\pi') - U_{r^*_\pi}(\pi_1) \le Regret(\pi_1, r^*_{\pi_1}) := \max_{\pi'\in\Pi} U_{r^*_{\pi_1}}(\pi') - U_{r^*_{\pi_1}}(\pi_1) := \max_{r\in R}\max_{\pi'\in\Pi} U_r(\pi') - U_r(\pi_1)$ indicates $\max_{\pi'\in\Pi} U_{r^*_\pi}(\pi') - \max_{\pi'\in\Pi} U_{r^*_{\pi_1}}(\pi') \le U_{r^*_\pi}(\pi_1) - U_{r^*_{\pi_1}}(\pi_1)$. However, we have proved that, for a weakly totally dominated policy, $\pi_1 \in MinimaxRegret(R)$ indicates $U_{r^*_{\pi_1}}(\pi_1) = \max_{r\in R} U_r(\pi_1)$. Hence, $\max_{\pi'\in\Pi} U_{r^*_\pi}(\pi') - \max_{\pi'\in\Pi} U_{r^*_{\pi_1}}(\pi') \le U_{r^*_\pi}(\pi_1) - U_{r^*_{\pi_1}}(\pi_1) \le 0$ and it contradicts $\max_{\pi'\in\Pi} U_{r^*_\pi}(\pi') > \max_{\pi'\in\Pi} U_{r^*_{\pi_1}}(\pi')$. Therefore, such $\pi_1$ does not exist. In summary, we have exhausted all conditions and can assert that for any policies, being a solution to $MinimaxRegret(R)$ is equivalent to a solution to $\arg\max_{\pi\in\Pi} U_\omega(\pi)$. We complete our proof.

$\square$

## A.5   Collective Validation of Similarity Between Expert and Agent

In Definition 2 and our definition of $Regret$ in Eq.2, we use the utility function $U_r$ to measure the performance of a policy. We now show that we can replace $U_r$ with other functions.

**Lemma 11.** *The solution of $MinimaxRegret(R_{E,\delta^*})$ does not change when $U_r$ in $MinimaxRegret$ is replace with $U_r(\pi) - f(r)$ where $f$ can be arbitrary function of $r$.*

*Proof.* When using $U_r(\pi) - f(r)$ instead of $U_r(\pi)$ to measure the policy performance, solving $MinimaxRegret(R)$ is to solve Eq. 14, which is the same as Eq.9.

$$
\begin{aligned}
MimimaxRegret(R) &= \arg\max_{\pi\in\Pi}\min_{r\in R} Regret(\pi, r) \\
&= \arg\max_{\pi\in\Pi}\min_{r\in R}\max_{\pi'\in\Pi} \{U_r(\pi') - f(r)\} - (U_r(\pi) - f(r)) \\
&= \arg\max_{\pi\in\Pi}\min_{r\in R}\max_{\pi'\in\Pi} U_r(\pi') - U_r(\pi) \quad (14)
\end{aligned}
$$

$\square$

Lemma 11 implies that we can use the policy-expert margin $U_r(\pi) - U_r(E)$ as a measurement of policy performance. This makes the rationale of using PAGAR-based IL for collective validation of similarity between $E$ and $\pi$ more intuitive.

### A.6 Criterion for Successful Policy Learning

To analyze the sufficient conditions for $MinimaxRegret$ to mitigate task-reward misalignment, we start by analyzing the general properties of $MinimaxRegret$ on arbitrary input $R$.

**Proposition 4.** *If the following conditions (1) (2) hold for $R$, then the optimal protagonist policy $\pi_P := MinimaxRegret(R)$ satisfies $\forall r^+ \in R, U_{r^+}(\pi_P) \geq \underline{U}_{r^+}$.*

*(1) There exists $r^+ \in R$, and $\max\limits_{r^+ \in R} \{\max\limits_{\pi \in \Pi} U_{r^+}(\pi) - \bar{U}_{r^+}\} < \min\limits_{r^+ \in R} \{\bar{U}_{r^+} - \underline{U}_{r^+}\};$*

*(2) There exists a policy $\pi^*$ such that $\forall r^+ \in R$, $U_{r^+}(\pi^*) \geq \bar{U}_{r^+}$, and $\forall r^- \in R$, $Regret(\pi^*, r^-) < \min\limits_{r^+ \in R} \{\bar{U}_{r^+} - \underline{U}_{r^+}\}$.*

*Proof.* Suppose the conditions are met, and a policy $\pi_1$ satisfies the property described in conditions 2). Then for any policy $\pi_2 \in MinimaxRegret(R)$, if $\pi_2$ does not satisfy the mentioned property, there exists a task-aligned reward function $r^+ \in R$ such that $U_{r^+}(\pi_2) \leq \underline{U}_{r^+}$. In this case $Regret(\pi_2, r^+) = \max\limits_{\pi \in \Pi} U_{r^+}(\pi) - U_{r^+}(\pi_2) \geq \overline{U}_{r^+} - \underline{U}_{r^+} \geq \min\limits_{r^+{}' \in R} \overline{U}_{r^+{}'} - \underline{U}_{r^+{}'}$. However, for $\pi_1$, it holds for any task-aligned reward function $\hat{r}^+ \in R$ that $Regret(\pi_1, \hat{r}^+) \leq \max\limits_{\pi \in \Pi} U_{\hat{r}^+}(\pi) - \overline{U}_{\hat{r}^+} \leq \max\limits_{r^+{}' \in R} \{\max\limits_{\pi \in \Pi} U_{r^+{}'}(\pi) - \overline{U}_{r^+{}'}\} < \min\limits_{r^+{}'' \in R} \{\overline{U}_{r^+{}'} - \underline{U}_{r^+{}'}\} \leq Regret(\pi_2, r^+)$, and it also holds for any misaligned reward function $r^- \in R$ that $Regret(\pi_1, r^-) < \min\limits_{r^+{}' \in R} \{\overline{U}_{r^+{}'} - \underline{U}_{r^+{}'}\} \leq Regret(\pi_2, \hat{r}^+)$. Hence, $Regret(\pi_1, r^+) < Regret(\pi_2, r^+)$, contradicting $\pi_2 \in MinimaxRegret(R)$. We complete the proof. $\square$

In Proposition 4, condition (1) states that the task-aligned reward functions in $R$ all have a low extent of misalignment while condition (2) states that there exists a $\pi^*$ that not only performs well under all $r^+$'s (thus being acceptable in the task) but also achieves relatively low regret under all $r^-$'s. Note that the more aligned the $r^+$'s, the more forgiving the tolerance for high regret on $r^-$. Furthermore, Proposition 5 shows that, under a stronger condition on the existence of a policy $\pi^*$ performing well under all reward functions in $R$, $MinimaxRegret(R)$ can guarantee to induce an acceptable policy, i.e., satisfying the condition (2) in Definition 3.

**Proposition 5** (Strong Acceptance)**.** *Assume that condition (1) in Proposition 4 is satisfied. In addition, if there exists a policy $\pi^*$ such that $\forall r \in R$, $Regret(\pi^*, r) < \max\limits_{r^+ \in R} \{\max\limits_{\pi \in \Pi} U_{r^+}(\pi) - \bar{U}_{r^+}\}$, then the optimal protagonist policy $\pi_P := MinimaxRegret(R)$ satisfies $\forall r^+ \in R$, $U_{r^+}(\pi_P) \geq \bar{U}_{r^+}$.*

*Proof.* Since $\max\limits_{r \in R} \max\limits_{\pi} U_r(\pi) - U_r(\pi_P) \leq \max\limits_{r \in R} \max\limits_{\pi} U_r(\pi) - U_r(\pi^*) < \max\limits_{r^+ \in R} \{\max\limits_{\pi \in \Pi} U_{r^+}(\pi) - \overline{U}_{r^+}\}$, we can conclude that for any $r^+ \in R$, $U_{r^+}(\pi_P) \geq \overline{U}_{r^+}$. The proof is complete. $\square$

Note that the assumptions in Proposition 4 and 5 are not trivially satisfiable for arbitrary $R$, e.g., if $R$ contains two reward functions with opposite signs, i.e., $r, -r \in R$, no policy can perform well under both $r$ and $-r$. However, in PAGAR-based IL, using $R_{E,\delta}$ in place of arbitrary $R$ is equivalent to using $E$ and $\delta$ to constrain the selection of reward functions, which can lead to additional implications.

**Theorem 2.** (Weak Acceptance) If the following conditions (1) (2) hold for $R_{E,\delta}$, then the optimal protagonist policy $\pi_P := MinimaxRegret(R_{E,\delta})$ satisfies $\forall r^+ \in R_{E,\delta}, U_{r^+}(\pi_P) \geq \underline{U}_{r^+}$.

(1) The condition (1) in Proposition 4 holds

(2) $\forall r^+ \in R_{E,\delta}$, $L_{r^+} \cdot W_E - \delta \leq \max\limits_{\pi \in \Pi} U_{r^+}(\pi) - \overline{U}_{r^+}$ and $\forall r^- \in R_{E,\delta}$, $L_{r^-} \cdot W_E - \delta < \min\limits_{r^+ \in R_{E,\delta}} \{\overline{U}_{r^+} - \underline{U}_{r^+}\}$.

*Proof.* We consider $U_r(\pi) = \mathbb{E}_{\tau \sim \pi}[r(\tau)]$. Since $W_E \triangleq \min_{\pi \in \Pi} W_1(\pi, E) = \frac{1}{K} \sup_{|r|_L \leq K} U_r(E) - U_r(\pi)$

for any $K > 0$, let $\pi^*$ be the policy that achieves the minimality in $W_E$. Then for any $r^+ \in R$, the term $L_{r^+} \cdot W_E - \delta \geq L_{r^+} \cdot \frac{1}{L_{r^+}} \sup_{|r|_L \leq L_{r^+}} U_r(E) - U_r(\pi) - \delta \geq U_{r^+}(E) - U_{r^+}(\pi) - (U_{r^+}(E) -$

$\max_{\pi' \in \Pi} U_{r^+}(\pi')) = \max_{\pi' \in \Pi} U_{r^+}(\pi') - U_{r^+}(\pi)$. Hence, for all $r^+ \in R$, $\max_{\pi' \in \Pi} U_{r^+}(\pi') - U_{r^+}(\pi) <$

$\max_{\pi' \in \Pi} U_{r^+}(\pi') - \bar{U}_{r^+}$, i.e., $U_{r^+}(\pi^*) \geq \bar{U}_{r^+}$. Likewise, $L_{r^-} \cdot W_E - \delta < \min_{r^+ \in R_{E,\delta}} \overline{U}_{r^+} - \underline{U}_{r^+}$ indicates

that for all $r^- \in R$, $\max_{\pi' \in \Pi} U_{r^+}(\pi') - U_{r^+}(\pi) < \min_{r^+ \in R_{E,\delta}} \overline{U}_{r^+} - \underline{U}_{r^+}$. Then, we have recovered the

condition (2) in Proposition 4. As a result, we deliver the same guarantees in Proposition 4. $\qquad\square$

Theorem 2 delivers the same guarantee as that of Proposition 4 but differs from Proposition 4 in that Condition (2) implicitly requires that for the policy $\pi^* = \arg\min_{\pi \in \Pi} W_1(\pi, E)$, the performance difference between $E$ and $\pi^*$ is small enough under all $r \in R_{E,\delta}$.

**Theorem 3.** (Strong Acceptance) Assume that the condition (1) in Theorem 4 holds for $R_{E,\delta}$. If for any $r \in R_{E,\delta}$, $L_r \cdot W_E - \delta \leq \min_{r^+ \in R_{E,\delta}} \{\max_{\pi \in \Pi} U_{r^+}(\pi) - \overline{U}_{r^+}\}$, then the optimal protagonist policy

$\pi_P = MinimaxRegret(R_{E,\delta})$ satisfies $\forall r^+ \in R_{E,\delta}$, $U_{r^+}(\pi_P) \geq \overline{U}_{r^+}$.

*Proof.* Again, we let $\pi^*$ be the policy that achieves the minimality in $W_E$. Then, we have $L_r \cdot W_E - \delta \geq L_r \cdot \frac{1}{L_r} \sup_{|r|_L \leq L_r} U_r(E) - U_r(\pi^*) - (U_{r^+}(E) - \max_{\pi' \in \Pi} U_{r^+}(\pi')) \geq \max_{\pi' \in \Pi} U_{r^+}(\pi') - U_{r^+}(\pi^*)$ for

any $r \in R_{E,\delta}$. We have recovered the condition in Proposition 5. The proof is complete. $\qquad\square$

## A.7 Stationary Solutions

In this section, we show that $MinimaxRegret$ is convex for $\pi_P$.

**Proposition 6.** $\max_{r \in R} Regret(\pi_P, r)$ *is convex in* $\pi_P$.

*Proof.* For any $\alpha \in [0,1]$ and $\pi_{P,1}, \pi_{P,2}$, there exists a $\pi_{P,3} = \alpha\pi_{P,1} + (1-\alpha)\pi_{P,2}$. Let $r_1, \pi_{A,1}$ and $r_2, \pi_{A,2}$ be the optimal reward and antagonist policy for $\pi_{P,1}$ and $\pi_{P,2}$ Then $\alpha \cdot (\max_{r \in R} \max_{\pi_A \in \Pi} U_r(\pi_A) -$

$U_r(\pi_{P,1})) + (1-\alpha) \cdot (\max_{r \in R} \max_{\pi_A \in \Pi} U_r(\pi_A) - U_r(\pi_{P,2})) = \alpha(U_{r_1}(\pi_{A,1}) - U_{r_1}(\pi_{P,1})) + (1-\alpha)(U_{r_2}(\pi_{A,2}) - U_{r_2}(\pi_{P,2})) \geq \alpha(U_{r_3}(\pi_{A,3}) - U_{r_3}(\pi_{P,1})) + (1-\alpha)(U_{r_2}(\pi_{A,3}) - U_{r_3}(\pi_{P,2})) = U_{r_3}(\pi_{A,3}) - U_{r_3}(\pi_{P,3})$. Therefore, $\max_{r \in R} \max_{\pi_A \in \Pi} U_r(\pi_A) - U_r(\pi_P)$ is convex in $\pi_P$. $\qquad\square$

## A.8 Compare PAGAR-Based IL with IRL-Based IL

**Assumption 2.** $\max_r \mathcal{J}_{IRL}(r)$ can reach Nash Equilibrium at an optimal reward function $r^*$ and its optimal policy $\pi_{r^*}$.

We make this assumption only to demonstrate how PAGAR-based IL can prevent performance degradation w.r.t IRL-based IL, which is preferred when IRL-based IL does not have a reward misalignment issue under ideal conditions. We draw two assertions from this assumption. The first one considers Maximum Margin IRL-based IL and shows that if using the optimal reward function set $R_{E,\delta^*}$ as input to $MinimaxRegret$, PAGAR-based IL and Maximum Margin IRL-based IL have the same solutions.

**Proposition 7.** $\pi_{r^*} = MinimiaxRegret(R_{E,\delta^*})$.

*Proof.* The reward function set $R_{E,\delta^*}$ and the policy set $\Pi_{acc}$ achieving Nash Equilibrium for $\arg\min_{r \in R} J_{IRL}(r)$ indicates that for any $r \in R_{E,\delta^*}, \pi \in \Pi_{acc}, \pi \in \arg\max_{\pi \in \Pi} U_r(\pi) - U_r(E)$. Then

$\Pi_{acc}$ will be the solution to $\arg\max_{\pi_P \in \Pi} \min_{r \in R_{E,\delta^*}} \left\{ \max_{\pi_A \in \Pi} U_r(\pi_A) - U_r(E) \right\} - (U_r(\pi_P) - U_r(E))$

because the policies in $\Pi_{acc}$ achieve zero regret. Then Lemma 11 states that $\Pi_{acc}$ will also be the solution to $\arg\max_{\pi_P\in\Pi} \min_{r\in R_{E,\delta^*}} \left\{\max_{\pi_A\in\Pi} U_r(\pi_A)\right\} - U_r(\pi_P)$. We finish the proof. $\qquad\square$

The proof can be found in Appendix A.6. The second assertion shows that if IRL-based IL can learn a policy to succeed in the task, $MinimaxRegret(R_{E,\delta})$ with $\delta < \delta^*$ can also learn a policy that succeeds in the task under certain condition. The proof can be found in Appendix A.6. This assertion also suggests that the designer should select a $\delta$ smaller than $\delta^*$ while making $\delta^* - \delta$ no greater than the expected size of the high-order policy utility interval.

**Proposition 8.** *If $r^*$ is a task-aligned reward function and $\delta \geq \delta^* - (\max_{\pi\in\Pi} U_{r^*}(\pi) - \overline{U}_{r^*})$, the optimal protagonist policy $\pi_P = MinimiaxRegret(R_{E,\delta})$ is guaranteed to be acceptable for the task.*

*Proof.* If $\pi_{r^*} \in MinimiaxRegret(R_{E,\delta})$, then $\pi_{r^*}$ can succeed in the task by definition. Now assume that $\pi_P \neq \pi_{r^*}$. Since $J_{IRL}$ achieves Nash Equilibrium at $r^*$ and $\pi_{r^*}$, for any other reward function $r$ we have $\max_{\pi\in\Pi} U_r(\pi) - U_r(\pi_{r^*}) \leq \delta^* - (U_r(E) - \max_{\pi\in\Pi} U_r(\pi)) \leq \delta^* - \delta$. We also have $\max_{r'\in R_{E,\delta}} Regret(r', \pi_P) \leq \max_{r'\in R_{E,\delta}} Regret(r', \pi_{r^*}) \leq \delta^* - \delta$. Furthermore, $Regre(r^*, \pi_P) \leq \max_{r'\in R_{E,\delta}} Regret(r', \pi_P)$. Hence, $Regre(r^*, \pi_P) \leq \delta - \delta^* \leq \max_{\pi\in\Pi} U_{r^+}(\pi) - \overline{U}_{r^+}$. In other words, $U_{r^*}(\pi_P) \in [\overline{U}_{r^*}, \max_{\pi\in\Pi} U_{r^*}(\pi)]$, indicating $\pi_P$ can succeed in the task. The proof is complete. $\qquad\square$

## A.9 Example 1

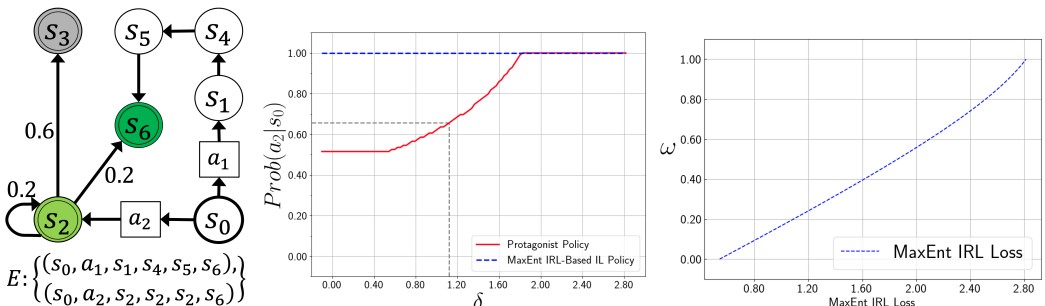

Figure 5: **Left:** Consider an MDP where there are two available actions $a_1, a_2$ at initial state $s_0$. In other states, actions make no difference: the transition probabilities are either annotated at the transition edges or equal 1 by default. States $s_3$ and $s_6$ are terminal states. Expert demonstrations are in $E$. **Middle**: x-axis indicates the MaxEnt IRL loss bound $\delta$ for $R_{E,\delta}$ as defined in Section A.3. The y-axis indicates the probability of the protagonist policy learned via $MinimaxRegret(R_{E,\delta})$ choosing $a_2$ at $s_0$. The red curve shows how different $\delta$'s lead to different protagonist policies. The blue dashed curve is for reference, showing the optimal policy under the optimal reward learned via MaxEnt IRL. **Right**: The curve shows how the MaxEnt IRL Loss changes with $\omega$.

*Example* 1. Figure 5 Left shows an illustrative example of how PAGAR-based IL mitigates reward misalignment in IRL-based IL. The task requires that *a policy must visit $s_2$ and $s_6$ with probabilities no less than* $0.5$ *within 5 steps*, i.e. $Prob(s_2 \mid \pi) \geq 0.5 \wedge Prob(s_6 \mid \pi) \geq 0.5$ where $Prob(s \mid \pi)$ is the probability of $\pi$ generating a trajectory that contains $s$ within the first 5 steps. It can be derived analytically that a successful policy must choose $a_2$ at $s_0$ with a probability within $[\frac{1}{2}, \frac{125}{188}]$. The derivation is as follows.

The trajectories that reach $s_6$ after choosing $a_2$ at $s_0$ include: $(s_0, a_2, s_2, s_6), (s_0, a_2, s_2, s_2, s_6), (s_0, a_2, s_2, s_2, s_2, s_6)$. The total probability equals $Prob(s_6 \mid \pi; s_0, a_2) = \frac{1}{5} + \frac{1}{5}^2 + \frac{1}{5}^3 = \frac{31}{125}$. Then the total probability of reaching $s_6$ equals $Prob(s_6 \mid \pi) = (1 - \pi(a_2 \mid s_0)) + \frac{31}{125} \cdot \pi(a_2 \mid s_0)$. For $Prob(s_6 \mid \pi)$ to be no less than 0.5, $\pi(a_2 \mid s_0)$ must be no greater than $\frac{125}{188}$.

The reward function hypothesis space is $\{r_\omega \mid r_\omega(s,a) = \omega \cdot r_1(s,a) + (1 - \omega) \cdot r_2(s,a)\}$ where $\omega \in [0,1]$ is a parameter, $r_1, r_2$ are two features. Specifically, $r_1(s,a)$ equals 1 if $s = s_2$ and equals 0 otherwise, and $r_2(s,a)$ equals 1 if $s = s_6$ and equals 0 otherwise. Given the demonstrations and the MDP, the maximum negative MaxEnt IRL loss $\delta^* \approx 2.8$ corresponds to the optimal parameter $\omega^* = 1$. This is computed based on Eq.6 in Ziebart et al. [2008]. The discount factor is $\gamma = 0.99$. When computing the normalization term $Z$ in Eq.4 of Ziebart et al. [2008], we only consider the trajectories within 5 steps. The optimal policy under $r_{\omega^*}$ chooses $a_2$ at $s_0$ with probability 1 and reaches $s_6$ with probability less than 0.25, thus failing to accomplish the task. The optimal protagonist policy $\pi_P = MinimaxRegret(R_{E,\delta})$ can succeed in the task as indicated by the grey dashed lines in Figure 5 Middle. It chooses $a_2$ at $s_0$ with probability 1 when $\delta$ is close to its maximum $\delta^*$. However, $\pi_P(a_2 \mid s_2)$ decreases as $\delta$ decreases. It turns out that for any $\delta < 1.1$ the optimal protagonist policy can succeed in the task. In Figure 5 Right, we further show how the MaxEnt IRL loss changes with $\omega$.

# B    Approach to Solving MinimaxRegret

In this section, we show how we derive the off-policy RL objective function for $\pi_P$. Also, we develop a series of theories that lead to two bounds of the Protagonist Antagonist Induced Regret. By using those bounds, we formulate objective functions for solving Imitation Learning problems with PAGAR.

## B.1    Off-Policy Objective Function for Protagonist Policy Training

For reader's convenience, we put the Theorem 1 in Schulman et al. [2015] here.

**Theorem 7** (Schulman et al. [2015]). *Let* $\alpha = \max_s D_{TV}(\pi_{old}, \pi_{new})$, *and let* $\epsilon = \max_s \mathbb{E}_{a \sim \pi_{new}}[A_{\pi_{old}}(s,a)]$, *then Eq.15 holds.*

$$U_r(\pi_{new}) \leq U_r(\pi_{old}) + \sum_{s \in \mathbb{S}} \rho_{\pi_{old}}(s) \sum_{a \in \mathbb{A}} \pi_{new}(a|s) A_{\pi_{old}}(s,a) + \frac{2\epsilon\gamma}{(1-\gamma)^2} \alpha^2 \qquad (15)$$

*where* $\rho_{\pi_{old}}(s) = \sum_{t=0}^{T} \gamma^t Prob(s^{(t)} = s | \pi_{old})$ *is the discounted visitation frequency of* $\pi_{old}$, $A_{\pi_{old}}$ *is the advantage function without considering the entropy.*

Algorithm 1 in Schulman et al. [2015] learns $\pi_{new}$ by maximizing the r.h.s of the inequality Eq.15, which only involves the trajectories and the advantage function of $\pi_{old}$. By moving $U_r(\pi_{old})$ from r.h.s of Eq.15 to the left, and replacing $\pi_{new}$ with $\pi_P$ and $\pi_{old}$ with $\pi_A$, we obtain a bound for $U_r(\pi_P) - U_r(\pi_A)$ as mentioned in Section 6.1. The PPO in Schulman et al. [2017] further simplifies the r.h.s of the inequality Eq.15 with a clipped importance sampling rate. We derived $J_{\pi_A}(\pi_P)$ by using the same trick.

## B.2    Protagonist Antagonist Induced Regret Bounds

Our theories are inspired by the on-policy policy improvement methods in Schulman et al. [2015]. The theories in Schulman et al. [2015] are under the setting where entropy regularizer is not considered. In our implementation, we always consider entropy regularized RL of which the objective is to learn a policy that maximizes $J_{RL}(\pi; r) = U_r(\pi) + \mathcal{H}(\pi)$. Also, since we use GAN-based IRL algorithms, the learned reward function $r$ as proved by Fu et al. [2018] is a distribution. Moreover, it is also proved in Fu et al. [2018] that a policy $\pi$ being optimal under $r$ indicates that $\log \pi \equiv r \equiv \mathcal{A}_\pi$. We omit the proof and let the reader refer to Fu et al. [2018] for details. Although all our theories are about the relationship between the Protagonist Antagonist Induced Regret and the soft advantage function $\mathcal{A}_\pi$, the equivalence between $\mathcal{A}_\pi$ and $r$ allows us to use the theories to formulate our reward optimization objective functions. To start off, we denote the reward function to be optimized as $r$. Given the intermediate learned reward function $r$, we study the Protagonist Antagonist Induced Regret between two policies $\pi_1$ and $\pi_2$.

**Lemma 12.** *Given a reward function $r$ and a pair of policies $\pi_1$ and $\pi_2$,*

$$U_r(\pi_1) - U_r(\pi_2) = \mathop{\mathbb{E}}_{\tau \sim \pi_1} \left[ \sum_{t=0}^{\infty} \gamma^t \mathcal{A}_{\pi_2}(s^{(t)}, a^{(t)}) \right] + \mathop{\mathbb{E}}_{\tau \sim \pi} \left[ \sum_{t=0}^{\infty} \gamma^t \mathcal{H}\left(\pi_2(\cdot | s^{(t)})\right) \right] \qquad (16)$$

*Proof.* This proof follows the proof of Lemma 1 in Schulman et al. [2015] where RL is not entropy-regularized. For entropy-regularized RL, since $\mathcal{A}_\pi(s, a^{(t)}) = \mathbb{E}_{s' \sim \mathcal{T}(\cdot|s,a^{(t)})} \left[ r(s, a^{(t)}) + \gamma \mathcal{V}_\pi(s') - \mathcal{V}_\pi(s) \right]$,

$$
\mathbb{E}_{\tau \sim \pi_1} \left[ \sum_{t=0}^{\infty} \gamma^t \mathcal{A}_{\pi_2}(s^{(t)}, a^{(t)}) \right]
$$

$$
= \mathbb{E}_{\tau \sim \pi_1} \left[ \sum_{t=0}^{\infty} \gamma^t \left( r(s^{(t+1)}, a^{(t+1)}) + \gamma \mathcal{V}_{\pi_2}(s^{(t+1)}) - \mathcal{V}_{\pi_2}(s^{(t)}) \right) \right]
$$

$$
= \mathbb{E}_{\tau \sim \pi_1} \left[ \sum_{t=0}^{\infty} \gamma^t r(s^{(t)}, a^{(t)}) - \mathcal{V}_{\pi_2}(s^{(0)}) \right]
$$

$$
= \mathbb{E}_{\tau \sim \pi_1} \left[ \sum_{t=0}^{\infty} \gamma^t r(s^{(t)}, a^{(t)}) \right] - \mathbb{E}_{s^{(0)} \sim d_0} \left[ \mathcal{V}_{\pi_2}(s^{(0)}) \right]
$$

$$
= \mathbb{E}_{\tau \sim \pi_1} \left[ \sum_{t=0}^{\infty} \gamma^t r(s^{(t)}, a^{(t)}) \right] - \mathbb{E}_{\tau \sim \pi_2} \left[ \sum_{t=0}^{\infty} \gamma^t r(s^{(t)}, a^{(t)}) + \mathcal{H}\left( \pi_2(\cdot|s^{(t)}) \right) \right]
$$

$$
= U_r(\pi_1) - U_r(\pi_2) - \mathbb{E}_{\tau \sim \pi_2} \left[ \sum_{t=0}^{\infty} \gamma^t \mathcal{H}\left( \pi_2(\cdot|s^{(t)}) \right) \right]
$$

$$
= U_r(\pi_1) - U_r(\pi_2) - \mathcal{H}(\pi_2)
$$

$\square$

*Remark* 1. Lemma 12 confirms that $\mathbb{E}_{\tau \sim \pi} \left[ \sum_{t=0}^{\infty} \gamma^t \mathcal{A}_\pi(s^{(t)}, a^{(t)}) \right] = U_r(\pi) - U_r(\pi) + \mathcal{H}(\pi) = \mathcal{H}(\pi)$.

We follow Schulman et al. [2015] and denote $\Delta \mathcal{A}(s) = \mathbb{E}_{a \sim \pi_1(\cdot|s)} [\mathcal{A}_{\pi_2}(s, a)] - \mathbb{E}_{a \sim \pi_2(\cdot|s)} [\mathcal{A}_{\pi_2}(s, a)]$ as the difference between the expected advantages of following $\pi_2$ after choosing an action respectively by following policy $\pi_1$ and $\pi_2$ at any state $s$. Although the setting of Schulman et al. [2015] differs from ours by having the expected advantage $\mathbb{E}_{a \sim \pi_2(\cdot|s)} [\mathcal{A}_{\pi_2}(s, a)]$ equal to 0 due to the absence of entropy regularization, the following definition and lemmas from Schulman et al. [2015] remain valid in our setting.

**Definition 9.** Schulman et al. [2015], the protagonist policy $\pi_1$ and the antagonist policy $\pi_2$) are $\alpha$-coupled if they defines a joint distribution over $(a, \tilde{a}) \in \mathbb{A} \times \mathbb{A}$, such that $Prob(a \neq \tilde{a}|s) \leq \alpha$ for all $s$.

**Lemma 13.** *Schulman et al. [2015] Given that the protagonist policy $\pi_1$ and the antagonist policy $\pi_2$ are $\alpha$-coupled, then for all state $s$,*

$$
|\Delta \mathcal{A}(s)| \leq 2\alpha \max_a |\mathcal{A}_{\pi_2}(s, a)| \tag{17}
$$

**Lemma 14.** *Schulman et al. [2015] Given that the protagonist policy $\pi_1$ and the antagonist policy $\pi_2$ are $\alpha$-coupled, then*

$$
\left| \mathbb{E}_{s^{(t)} \sim \pi_1} \left[ \Delta \mathcal{A}(s^{(t)}) \right] - \mathbb{E}_{s^{(t)} \sim \pi_2} \left[ \Delta \mathcal{A}(s^{(t)}) \right] \right| \leq 4\alpha (1 - (1-\alpha)^t) \max_{s,a} |\mathcal{A}_{\pi_2}(s, a)| \tag{18}
$$

**Lemma 15.** *Given that the protagonist policy $\pi_1$ and the antagonist policy $\pi_2$ are $\alpha$-coupled, then*

$$
\mathbb{E}_{\substack{s^{(t)} \sim \pi_1 \\ a^{(t)} \sim \pi_2}} \left[ \mathcal{A}_{\pi_2}(s^{(t)}, a^{(t)}) \right] - \mathbb{E}_{\substack{s^{(t)} \sim \pi_2 \\ a^{(t)} \sim \pi_2}} \left[ \mathcal{A}_{\pi_2}(s^{(t)}, a^{(t)}) \right] \leq 2(1 - (1-\alpha)^t) \max_{(s,a)} |\mathcal{A}_{\pi_2}(s, a)| \tag{19}
$$

*Proof.* The proof is similar to that of Lemma 14 in Schulman et al. [2015]. Let $n_t$ be the number of times that $a^{(t')} \sim \pi_1$ does not equal $a^{(t')} \sim \pi_2$ for $t' < t$, i.e., the number of times that $\pi_1$ and $\pi_2$

disagree before timestep $t$. Then for $s^{(t)} \sim \pi_1$, we have the following.

$$\mathbb{E}_{s^{(t)} \sim \pi_1} \left[ \mathbb{E}_{a^{(t)} \sim \pi_2} \left[ \mathcal{A}_{\pi_2}(s^{(t)}, a^{(t)}) \right] \right]$$

$$= P(n_t = 0) \mathbb{E}_{\substack{s^{(t)} \sim \pi_1 \\ n_t = 0}} \left[ \mathbb{E}_{a^{(t)} \sim \pi_2} \left[ \mathcal{A}_{\pi_2}(s^{(t)}, a^{(t)}) \right] \right] + P(n_t > 0) \mathbb{E}_{\substack{s^{(t)} \sim \pi_1 \\ n_t > 0}} \left[ \mathbb{E}_{a^{(t)} \sim \pi_2} \left[ \mathcal{A}_{\pi_2}(s^{(t)}, a^{(t)}) \right] \right]$$

The expectation decomposes similarly for $s^{(t)} \sim \pi_2$.

$$\mathbb{E}_{\substack{s^{(t)} \sim \pi_2 \\ a^{(t)} \sim \pi_2}} \left[ \mathcal{A}_{\pi_2}(s^{(t)}, a^{(t)}) \right]$$

$$= P(n_t = 0) \mathbb{E}_{\substack{s^{(t)} \sim \pi_2 \\ a^{(t)} \sim \pi_2 \\ n_t = 0}} \left[ \mathcal{A}_{\pi_2}(s^{(t)}, a^{(t)}) \right] + P(n_t > 0) \mathbb{E}_{\substack{s^{(t)} \sim \pi_2 \\ a^{(t)} \sim \pi_2 \\ n_t > 0}} \left[ \mathcal{A}_{\pi_2}(s^{(t)}, a^{(t)}) \right]$$

When computing $\mathbb{E}_{s^{(t)} \sim \pi_1} \left[ \mathbb{E}_{a^{(t)} \sim \pi_2} \left[ \mathcal{A}_{\pi_2}(s^{(t)}, a^{(t)}) \right] \right] - \mathbb{E}_{\substack{s^{(t)} \sim \pi_2 \\ a^{(t)} \sim \pi_2}} \left[ \mathcal{A}_{\pi_2}(s^{(t)}, a^{(t)}) \right]$, the terms with $n_t = 0$ cancel each other because $n_t = 0$ indicates that $\pi_1$ and $\pi_2$ agreed on all timesteps less than $t$. That leads to the following.

$$\mathbb{E}_{s^{(t)} \sim \pi_1} \left[ \mathbb{E}_{a^{(t)} \sim \pi_2} \left[ \mathcal{A}_{\pi_2}(s^{(t)}, a^{(t)}) \right] \right] - \mathbb{E}_{\substack{s^{(t)} \sim \pi_2 \\ a^{(t)} \sim \pi_2}} \left[ \mathcal{A}_{\pi_2}(s^{(t)}, a^{(t)}) \right]$$

$$= P(n_t > 0) \mathbb{E}_{\substack{s^{(t)} \sim \pi_1 \\ n_t > 0}} \left[ \mathbb{E}_{a^{(t)} \sim \pi_2} \left[ \mathcal{A}_{\pi_2}(s^{(t)}, a^{(t)}) \right] \right] - P(n_t > 0) \mathbb{E}_{\substack{s^{(t)} \sim \pi_2 \\ a^{(t)} \sim \pi_2 \\ n_t > 0}} \left[ \mathcal{A}_{\pi_2}(s^{(t)}, a^{(t)}) \right]$$

By definition of $\alpha$, the probability of $\pi_1$ and $\pi_2$ agreeing at timestep $t'$ is no less than $1 - \alpha$. Hence, $P(n_t > 0) \leq 1 - (1 - \alpha^t)^t$. Hence, we have the following bound.

$$\left| \mathbb{E}_{s^{(t)} \sim \pi_1} \left[ \mathbb{E}_{a^{(t)} \sim \pi_2} \left[ \mathcal{A}_{\pi_2}(s^{(t)}, a^{(t)}) \right] \right] - \mathbb{E}_{\substack{s^{(t)} \sim \pi_2 \\ a^{(t)} \sim \pi_2}} \left[ \mathcal{A}_{\pi_2}(s^{(t)}, a^{(t)}) \right] \right|$$

$$= \left| P(n_t > 0) \mathbb{E}_{\substack{s^{(t)} \sim \pi_1 \\ n_t > 0}} \left[ \mathbb{E}_{a^{(t)} \sim \pi_2} \left[ \mathcal{A}_{\pi_2}(s^{(t)}, a^{(t)}) \right] \right] - P(n_t > 0) \mathbb{E}_{\substack{s^{(t)} \sim \pi_2 \\ a^{(t)} \sim \pi_2 \\ n_t > 0}} \left[ \mathcal{A}_{\pi_2}(s^{(t)}, a^{(t)}) \right] \right|$$

$$\leq P(n_t > 0) \left( \left| \mathbb{E}_{\substack{s^{(t)} \sim \pi_1 \\ a^{(t)} \sim \pi_2 \\ n_t \geq 0}} \left[ \mathcal{A}_{\pi_2}(s^{(t)}, a^{(t)}) \right] \right| + \left| \mathbb{E}_{\substack{s^{(t)} \sim \pi_2 \\ a^{(t)} \sim \pi_2 \\ n_t > 0}} \left[ \mathcal{A}_{\pi_2}(s^{(t)}, a^{(t)}) \right] \right| \right)$$

$$\leq 2(1 - (1 - \alpha)^t) \max_{(s,a)} |\mathcal{A}_{\pi_2}(s, a)| \tag{20}$$

$\square$

The preceding lemmas lead to the proof for Theorem 4 in the main text.

**Theorem 4.** Suppose that $\pi_2$ is the optimal policy in terms of entropy regularized RL under $r$. Let $\alpha = \max_s D_{TV}(\pi_1(\cdot|s), \pi_2(\cdot|s))$, $\epsilon = \max_{s,a} |\mathcal{A}_{\pi_2}(s, a^{(t)})|$, and $\Delta \mathcal{A}(s) = \mathbb{E}_{a \sim \pi_1} [\mathcal{A}_{\pi_2}(s, a)] -$

$\mathbb{E}_{a\sim\pi_2}[\mathcal{A}_{\pi_2}(s,a)]$. For any policy $\pi_1$, the following bounds hold.

$$\left| U_r(\pi_1) - U_r(\pi_2) - \sum_{t=0}^{\infty} \gamma^t \mathbb{E}_{s^{(t)}\sim\pi_1}\left[\Delta\mathcal{A}(s^{(t)})\right]\right| \leq \frac{2\alpha\gamma\epsilon}{(1-\gamma)^2} \tag{21}$$

$$\left| U_r(\pi_1) - U_r(\pi_2) - \sum_{t=0}^{\infty} \gamma^t \mathbb{E}_{s^{(t)}\sim\pi_2}\left[\Delta\mathcal{A}(s^{(t)})\right]\right| \leq \frac{2\alpha\gamma(2\alpha+1)\epsilon}{(1-\gamma)^2} \tag{22}$$

*Proof.* We first leverage Lemma 12 to derive Eq.23. Note that since $\pi_2$ is optimal under $r$, Remark 1 confirmed that $\mathcal{H}(\pi_2) = -\sum_{t=0}^{\infty}\gamma^t \mathbb{E}_{s^{(t)}\sim\pi_2}\left[\mathbb{E}_{a^{(t)}\sim\pi_2}\left[\mathcal{A}_{\pi_2}(s^{(t)},a^{(t)})\right]\right]$.

$$
\begin{aligned}
& U_r(\pi_1) - U_r(\pi_2) \\
=~ & (U_r(\pi_1) - U_r(\pi_2) - \mathcal{H}(\pi_2)) + \mathcal{H}(\pi_2) \\
=~ & \mathbb{E}_{\tau\sim\pi_1}\left[\sum_{t=0}^{\infty}\gamma^t\mathcal{A}_{\pi_2}(s^{(t)},a^{(t)})\right] + \mathcal{H}(\pi_2) \\
=~ & \mathbb{E}_{\tau\sim\pi_1}\left[\sum_{t=0}^{\infty}\gamma^t\mathcal{A}_{\pi_2}(s^{(t)},a^{(t)})\right] - \sum_{t=0}^{\infty}\gamma^t\mathbb{E}_{s^{(t)}\sim\pi_2}\left[\mathbb{E}_{a^{(t)}\sim\pi_2}\left[\mathcal{A}_{\pi_2}(s^{(t)},a^{(t)})\right]\right] \\
=~ & \sum_{t=0}^{\infty}\gamma^t\mathbb{E}_{s^{(t)}\sim\pi_1}\left[\mathbb{E}_{a^{(t)}\sim\pi_1}\left[\mathcal{A}_{\pi_2}(s^{(t)},a^{(t)})\right] - \mathbb{E}_{a^{(t)}\sim\pi_2}\left[\mathcal{A}_{\pi_2}(s^{(t)},a^{(t)})\right]\right] + \\
& \sum_{t=0}^{\infty}\gamma^t\left(\mathbb{E}_{s^{(t)}\sim\pi_1}\left[\mathbb{E}_{a^{(t)}\sim\pi_2}\left[\mathcal{A}_{\pi_2}(s^{(t)},a^{(t)})\right]\right] - \mathbb{E}_{s^{(t)}\sim\pi_2}\left[\mathbb{E}_{a^{(t)}\sim\pi_2}\left[\mathcal{A}_{\pi_2}(s^{(t)},a^{(t)})\right]\right]\right) \\
=~ & \sum_{t=0}^{\infty}\gamma^t\mathbb{E}_{s^{(t)}\sim\pi_1}\left[\Delta\mathcal{A}(s^{(t)})\right] + \\
& \sum_{t=0}^{\infty}\gamma^t\left(\mathbb{E}_{s^{(t)}\sim\pi_1}\left[\mathbb{E}_{a^{(t)}\sim\pi_2}\left[\mathcal{A}_{\pi_2}(s^{(t)},a^{(t)})\right]\right] - \mathbb{E}_{s^{(t)}\sim\pi_2}\left[\mathbb{E}_{a^{(t)}\sim\pi_2}\left[\mathcal{A}_{\pi_2}(s^{(t)},a^{(t)})\right]\right]\right)
\end{aligned} \tag{23}
$$

We switch terms between Eq.23 and $U_r(\pi_1) - U_r(\pi_2)$, then use Lemma 15 to derive Eq.24.

$$
\begin{aligned}
& \left| U_r(\pi_1) - U_r(\pi_2) - \sum_{t=0}^{\infty}\gamma^t\mathbb{E}_{s^{(t)}\sim\pi_1}\left[\Delta\mathcal{A}(s^{(t)})\right]\right| \\
=~ & \left|\sum_{t=0}^{\infty}\gamma^t\left(\mathbb{E}_{s^{(t)}\sim\pi_1}\left[\mathbb{E}_{a^{(t)}\sim\pi_2}\left[\mathcal{A}_{\pi_2}(s^{(t)},a^{(t)})\right]\right] - \mathbb{E}_{s^{(t)}\sim\pi_2}\left[\mathbb{E}_{a^{(t)}\sim\pi_2}\left[\mathcal{A}_{\pi_2}(s^{(t)},a^{(t)})\right]\right]\right)\right| \\
\leq~ & \sum_{t=0}^{\infty}\gamma^t \cdot 2\max_{(s,a)}|\mathcal{A}_{\pi_2}(s,a)|\cdot(1-(1-\alpha)^t) \leq \frac{2\alpha\gamma\max_{(s,a)}|\mathcal{A}_{\pi_2}(s,a)|}{(1-\gamma)^2}
\end{aligned} \tag{24}
$$

Alternatively, we can expand $U_r(\pi_2) - U_r(\pi_1)$ into Eq.25. During the process, $\mathcal{H}(\pi_2)$ is converted into $-\sum_{t=0}^{\infty} \gamma^t \underset{s^{(t)} \sim \pi_2}{\mathbb{E}} \left[ \underset{a^{(t)} \sim \pi_2}{\mathbb{E}} \left[ \mathcal{A}_{\pi_2}(s^{(t)}, a^{(t)}) \right] \right]$.

$$
\begin{aligned}
& U_r(\pi_1) - U_r(\pi_2) \\
= & \; (U_r(\pi_1) - U_r(\pi_2) - \mathcal{H}(\pi_2)) + \mathcal{H}(\pi_2) \\
= & \; \underset{\tau \sim \pi_1}{\mathbb{E}} \left[ \sum_{t=0}^{\infty} \gamma^t \mathcal{A}_{\pi_2}(s^{(t)}, a^{(t)}) \right] + \mathcal{H}(\pi_2) \\
= & \; \sum_{t=0}^{\infty} \gamma^t \underset{s^{(t)} \sim \pi_1}{\mathbb{E}} \left[ \underset{a^{(t)} \sim \pi_1}{\mathbb{E}} \left[ \mathcal{A}_{\pi_2}(s^{(t)}, a^{(t)}) \right] \right] + \mathcal{H}(\pi_2) \\
= & \; \sum_{t=0}^{\infty} \gamma^t \underset{s^{(t)} \sim \pi_1}{\mathbb{E}} \left[ \Delta A(s^{(t)}) + \underset{a^{(t)} \sim \pi_2}{\mathbb{E}} \left[ \mathcal{A}_{\pi_2}(s^{(t)}, a^{(t)}) \right] \right] + \mathcal{H}(\pi_2) \\
= & \; \sum_{t=0}^{\infty} \gamma^t \underset{s^{(t)} \sim \pi_2}{\mathbb{E}} \left[ \underset{a^{(t)} \sim \pi_1}{\mathbb{E}} \left[ \mathcal{A}_{\pi_2}(s^{(t)}, a^{(t)}) \right] - \underset{a^{(t)} \sim \pi_2}{\mathbb{E}} \left[ \mathcal{A}_{\pi_2}(s^{(t)}, a^{(t)}) \right] - \Delta \mathcal{A}(s^{(t)}) \right] + \\
& \; \underset{s^{(t)} \sim \pi_1}{\mathbb{E}} \left[ \Delta \mathcal{A}(s^{(t)}) + \underset{a^{(t)} \sim \pi_2}{\mathbb{E}} \left[ \mathcal{A}_{\pi_2}(s^{(t)}, a^{(t)}) \right] \right] - \underset{\substack{s^{(t)} \sim \pi_2 \\ a^{(t)} \sim \pi_2}}{\mathbb{E}} \left[ \mathcal{A}_{\pi_2}(s^{(t)}, a^{(t)}) \right] \\
= & \; \sum_{t=0}^{\infty} \gamma^t \left( \underset{s^{(t)} \sim \pi_1}{\mathbb{E}} \left[ \underset{a^{(t)} \sim \pi_2}{\mathbb{E}} \left[ \mathcal{A}_{\pi_2}(s^{(t)}, a^{(t)}) \right] \right] - 2 \underset{s^{(t)} \sim \pi_2}{\mathbb{E}} \left[ \underset{a^{(t)} \sim \pi_2}{\mathbb{E}} \left[ \mathcal{A}_{\pi_2}(s^{(t)}, a^{(t)}) \right] \right] \right) + \\
& \; \sum_{t=0}^{\infty} \gamma^t \left( \underset{s^{(t)} \sim \pi_2}{\mathbb{E}} \left[ \underset{a^{(t)} \sim \pi_1}{\mathbb{E}} \left[ \mathcal{A}_{\pi_2}(s^{(t)}, a^{(t)}) \right] \right] - ( \underset{s^{(t)} \sim \pi_2}{\mathbb{E}} \left[ \Delta \mathcal{A}(s^{(t)}) \right] - \underset{s^{(t)} \sim \pi_1}{\mathbb{E}} \left[ \Delta \mathcal{A}(s^{(t)}) \right] ) \right)
\end{aligned}
$$

(25)

We switch terms between Eq.25 and $U_r(\pi_1) - U_r(\pi_2)$, then base on Lemma 14 and 15 to derive the inequality in Eq.26.

$$
\begin{aligned}
& \left| U_r(\pi_1) - U_r(\pi_2) - \sum_{t=0}^{\infty} \gamma^t \underset{s^{(t)} \sim \pi_2}{\mathbb{E}} \left[ \Delta \mathcal{A}_{\pi}(s^{(t)}, a^{(t)}) \right] \right| \\
= & \; \Bigg| U_r(\pi_1) - U_r(\pi_2) - \\
& \quad \sum_{t=0}^{\infty} \gamma^t \left( \underset{s^{(t)} \sim \pi_2}{\mathbb{E}} \left[ \underset{a^{(t)} \sim \pi_1}{\mathbb{E}} \left[ \mathcal{A}_{\pi_2}(s^{(t)}, a^{(t)}) \right] \right] - \underset{s^{(t)} \sim \pi_2}{\mathbb{E}} \left[ \underset{a^{(t)} \sim \pi_2}{\mathbb{E}} \left[ \mathcal{A}_{\pi_2}(s^{(t)}, a^{(t)}) \right] \right] \right) \Bigg| \\
= & \; \Bigg| \sum_{t=0}^{\infty} \gamma^t \left( \underset{s^{(t)} \sim \pi_2}{\mathbb{E}} \left[ \Delta \mathcal{A}(s^{(t)}) \right] - \underset{s^{(t)} \sim \pi_1}{\mathbb{E}} \left[ \Delta \mathcal{A}(s^{(t)}) \right] \right) - \\
& \quad \sum_{t=0}^{\infty} \gamma^t \left( \underset{s^{(t)} \sim \pi_1}{\mathbb{E}} \left[ \underset{a^{(t)} \sim \pi_2}{\mathbb{E}} \left[ \mathcal{A}_{\pi_2}(s^{(t)}, a^{(t)}) \right] \right] - \underset{s^{(t)} \sim \pi_2}{\mathbb{E}} \left[ \underset{a^{(t)} \sim \pi_2}{\mathbb{E}} \left[ \mathcal{A}_{\pi_2}(s^{(t)}, a^{(t)}) \right] \right] \right) \Bigg| \\
\leq & \; \Bigg| \sum_{t=0}^{\infty} \gamma^t \left( \underset{s^{(t)} \sim \pi_2}{\mathbb{E}} \left[ \Delta \mathcal{A}(s^{(t)}) \right] - \underset{s^{(t)} \sim \pi_1}{\mathbb{E}} \left[ \Delta \mathcal{A}(s^{(t)}) \right] \right) \Bigg| + \\
& \quad \Bigg| \sum_{t=0}^{\infty} \gamma^t \left( \underset{s^{(t)} \sim \pi_1}{\mathbb{E}} \left[ \underset{a^{(t)} \sim \pi_2}{\mathbb{E}} \left[ \mathcal{A}_{\pi_2}(s^{(t)}, a^{(t)}) \right] \right] - \underset{s^{(t)} \sim \pi_2}{\mathbb{E}} \left[ \underset{a^{(t)} \sim \pi_2}{\mathbb{E}} \left[ \mathcal{A}_{\pi_2}(s^{(t)}, a^{(t)}) \right] \right] \right) \Bigg| \\
\leq & \; \sum_{t=0}^{\infty} \gamma^t \left( (1 - (1-\alpha)^t)(4\alpha \underset{s,a}{\max}|\mathcal{A}_{\pi_2}(s, a)| + 2\underset{(s,a)}{\max}|\mathcal{A}_{\pi_2}(s, a)|) \right) \\
\leq & \; \frac{2\alpha\gamma(2\alpha + 1)\underset{s,a}{\max}|\mathcal{A}_{\pi_2}(s, a)|}{(1 - \gamma)^2}
\end{aligned}
$$

(26)

It is stated in Schulman et al. [2015] that $\max\limits_{s} D_{TV}(\pi_2(\cdot|s), \pi_1(\cdot|s)) \leq \alpha$. Hence, by letting $\alpha := \max\limits_{s} D_{TV}(\pi_2(\cdot|s), \pi_1(\cdot|s))$, Eq.23 and 26 still hold. Then, we have proved Theorem 4. $\qquad\square$

### B.3 Objective Functions of Reward Optimization

To derive $J_{R,1}$ and $J_{R,2}$, we let $\pi_1 = \pi_P$ and $\pi_2 = \pi_A$. Then based on Eq.21 and 22 we derive the following upper-bounds of $U_r(\pi_P) - U_r(\pi_A)$.

$$U_r(\pi_P) - U_r(\pi_A) \leq \sum_{t=0}^{\infty} \gamma^t \mathop{\mathbb{E}}_{s^{(t)}\sim\pi_P} \left[\Delta\mathcal{A}(s^{(t)})\right] + \frac{2\alpha\gamma(2\alpha+1)\epsilon}{(1-\gamma)^2} \tag{27}$$

$$U_r(\pi_P) - U_r(\pi_A) \geq \sum_{t=0}^{\infty} \gamma^t \mathop{\mathbb{E}}_{s^{(t)}\sim\pi_A} \left[\Delta\mathcal{A}(s^{(t)})\right] - \frac{2\alpha\gamma\epsilon}{(1-\gamma)^2} \tag{28}$$

By our assumption that $\pi_A$ is optimal under $r$, we have $\mathcal{A}_{\pi_A} \equiv r$ Fu et al. [2018]. This equivalence enables us to replace $\mathcal{A}_{\pi_A}$'s in $\Delta\mathcal{A}$ with $r$. As for the $\frac{2\alpha\gamma(2\alpha+1)\epsilon}{(1-\gamma)^2}$ and $\frac{2\alpha\gamma\epsilon}{(1-\gamma)^2}$ terms, since the objective is to maximize $U_r(\pi_A) - U_r(\pi_B)$, we heuristically estimate the $\epsilon$ in Eq.27 by using the samples from $\pi_P$ and the $\epsilon$ in Eq.28 by using the samples from $\pi_A$. As a result we have the objective functions defined as Eq.29 and 30 where $\xi_1(s,a) = \frac{\pi_P(a^{(t)}|s^{(t)})}{\pi_A(a^{(t)}|s^{(t)})}$ and $\xi_2 = \frac{\pi_A(a^{(t)}|s^{(t)})}{\pi_P(a^{(t)}|s^{(t)})}$ are the importance sampling probability ratio derived from the definition of $\Delta\mathcal{A}$; $C_1 \propto -\frac{\gamma\hat{\alpha}}{(1-\gamma)}$ and $C_2 \propto \frac{\gamma\hat{\alpha}}{(1-\gamma)}$ where $\hat{\alpha}$ is either an estimated maximal KL-divergence between $\pi_A$ and $\pi_B$ since $D_{KL} \geq D_{TV}^2$ according to Schulman et al. [2015], or an estimated maximal $D_{TV}^2$ depending on whether the reward function is Gaussian or Categorical. We also note that for finite horizon tasks, we compute the average rewards instead of the discounted accumulated rewards in Eq.30 and 29.

$$J_{R,1}(r;\pi_P,\pi_A) := \mathop{\mathbb{E}}_{\tau\sim\pi_A} \left[\sum_{t=0}^{\infty}\gamma^t\left(\xi_1(s^{(t)},a^{(t)})-1\right)\cdot r(s^{(t)},a^{(t)})\right] + C_1 \max_{(s,a)\sim\pi_A}|r(s,a)| \tag{29}$$

$$J_{R,2}(r;\pi_P,\pi_A) := \mathop{\mathbb{E}}_{\tau\sim\pi_P} \left[\sum_{t=0}^{\infty}\gamma^t\left(1-\xi_2(s^{(t)},a^{(t)})\right)\cdot r(s^{(t)},a^{(t)})\right] + C_2 \max_{(s,a)\sim\pi_P}|r(s,a)| \tag{30}$$

Beside $J_{R,1}, J_{R,2}$, we additionally use two more objective functions based on the derived bounds. W $J_{R,r}(r;\pi_A,\pi_P)$. By denoting the optimal policy under $r$ as $\pi^*$, $\alpha^* = \max\limits_{s\in\mathbb{S}} D_{TV}(\pi^*(\cdot|s), \pi_A(\cdot|s))$, $\epsilon^* = \max\limits_{(s,a^{(t)})}|\mathcal{A}_{\pi^*}(s,a^{(t)})|$, and $\Delta\mathcal{A}_A^*(s) = \mathop{\mathbb{E}}_{a\sim\pi_A}[\mathcal{A}_{\pi^*}(s,a)] - \mathop{\mathbb{E}}_{a\sim\pi^*}[\mathcal{A}_{\pi^*}(s,a)]$, we have the following.

$$
\begin{aligned}
& U_r(\pi_P) - U_r(\pi^*) \\
= \; & U_r(\pi_P) - U_r(\pi_A) + U_r(\pi_A) - U_r(\pi^*) \\
\leq \; & U_r(\pi_P) - U_r(\pi_A) + \sum_{t=0}^{\infty}\gamma^t \mathop{\mathbb{E}}_{s^{(t)}\sim\pi_A}\left[\Delta\mathcal{A}_A^*(s^{(t)})\right] + \frac{2\alpha^*\gamma\epsilon^*}{(1-\gamma)^2} \\
= \; & U_r(\pi_P) - \sum_{t=0}^{\infty}\gamma^t \mathop{\mathbb{E}}_{s^{(t)}\sim\pi_A}\left[\mathop{\mathbb{E}}_{a^{(t)}\sim\pi_A}\left[r(s^{(t)},a^{(t)})\right]\right] + \\
& \sum_{t=0}^{\infty}\gamma^t \mathop{\mathbb{E}}_{s^{(t)}\sim\pi_A}\left[\mathop{\mathbb{E}}_{a^{(t)}\sim\pi_A}\left[\mathcal{A}_{\pi^*}(s^{(t)},a^{(t)})\right] - \mathop{\mathbb{E}}_{a^{(t)}\sim\pi^*}\left[\mathcal{A}_{\pi^*}(s^{(t)},a^{(t)})\right]\right] + \frac{2\alpha^*\gamma\epsilon^*}{(1-\gamma)^2} \\
= \; & U_r(\pi_P) - \sum_{t=0}^{\infty}\gamma^t \mathop{\mathbb{E}}_{s^{(t)}\sim\pi_A}\left[\mathop{\mathbb{E}}_{a^{(t)}\sim\pi^*}\left[\mathcal{A}_{\pi^*}(s^{(t)},a^{(t)})\right]\right] + \frac{2\alpha^*\gamma\epsilon^*}{(1-\gamma)^2} \\
= \; & \mathop{\mathbb{E}}_{\tau\sim\pi_P}\left[\sum_{t=0}^{\infty}\gamma^t r(s^{(t)},a^{(t)})\right] - \mathop{\mathbb{E}}_{\tau\sim\pi_A}\left[\sum_{t=0}^{\infty}\gamma^t \frac{\exp(r(s^{(t)},a^{(t)}))}{\pi_A(a^{(t)}|s^{(t)})}r(s^{(t)},a^{(t)})\right] + \frac{2\alpha^*\gamma\epsilon^*}{(1-\gamma)^2} \quad (31)
\end{aligned}
$$

Let $\xi_3 = \frac{\exp(r(s^{(t)},a^{(t)}))}{\pi_A(a^{(t)}|s^{(t)})}$ be the importance sampling probability ratio. It is suggested in Schulman et al. [2017] that instead of directly optimizing the objective function Eq.31, optimizing a surrogate

objective function as in Eq.32, which is an upper-bound of Eq.31, with some small $\delta \in (0,1)$ can be much less expensive and still effective.

$$J_{R,3}(r; \pi_P, \pi_A) := \underset{\tau \sim \pi_P}{\mathbb{E}} \left[ \sum_{t=0}^{\infty} \gamma^t r(s^{(t)}, a^{(t)}) \right] -$$
$$\underset{\tau \sim \pi_A}{\mathbb{E}} \left[ \sum_{t=0}^{\infty} \gamma^t \min \left( \xi_3 \cdot r(s^{(t)}, a^{(t)}), clip(\xi_3, 1-\delta, 1+\delta) \cdot r(s^{(t)}, a^{(t)}) \right) \right] \tag{32}$$

Alternatively, we let $\Delta \mathcal{A}_P^*(s) = \underset{a \sim \pi_P}{\mathbb{E}} [\mathcal{A}_{\pi^*}(s,a)] - \underset{a \sim \pi^*}{\mathbb{E}} [\mathcal{A}_{\pi^*}(s,a)]$. The according to Eq.27, we have the following.

$$U_r(\pi_P) - U_r(\pi^*)$$
$$\leq \sum_{t=0}^{\infty} \gamma^t \underset{s^{(t)} \sim \pi_P}{\mathbb{E}} \left[ \Delta \mathcal{A}_P^*(s^{(t)}) \right] + \frac{2\alpha^* \gamma (2\alpha^* + 1)\epsilon^*}{(1-\gamma)^2}$$
$$= \sum_{t=0}^{\infty} \gamma^t \underset{s^{(t)} \sim \pi_P}{\mathbb{E}} \left[ \underset{a^{(t)} \sim \pi_P}{\mathbb{E}} \left[ \mathcal{A}_{\pi^*}(s^{(t)}, a^{(t)}) \right] - \underset{a^{(t)} \sim \pi^*}{\mathbb{E}} \left[ \mathcal{A}_{\pi^*}(s^{(t)}, a)^{(t)} \right] \right] + \frac{2\alpha^* \gamma (2\alpha^* + 1)\epsilon^*}{(1-\gamma)^2}$$
$$\tag{33}$$

Then a new objective function $J_{R,4}$ is formulated in Eq.34 where $\xi_4 = \frac{\exp(r(s^{(t)}, a^{(t)}))}{\pi_P(a^{(t)} | s^{(t)})}$.

$$J_{R,4}(r; \pi_P, \pi_A) := \underset{\tau \sim \pi_P}{\mathbb{E}} \left[ \sum_{t=0}^{\infty} \gamma^t r(s^{(t)}, a^{(t)}) \right] -$$
$$\underset{\tau \sim \pi_P}{\mathbb{E}} \left[ \sum_{t=0}^{\infty} \gamma^t \min \left( \xi_4 \cdot r(s^{(t)}, a^{(t)}), clip(\xi_4, 1-\delta, 1+\delta) \cdot r(s^{(t)}, a^{(t)}) \right) \right] \tag{34}$$

### B.4   Incorporating IRL Algorithms

**Online RL Setting.** In our implementation, we combine PAGAR with GAIL Ho and Ermon [2016], VAIL Peng et al. [2019], and f-IRL Ni et al. [2021], respectively. In this section, we use $J_{IRL}$ to indicate the IRL loss to be minimized in place of the notation $\mathcal{J}_{IRL}$ in the main text. Accordingly, $\delta$ is the target IRL loss.

- **f-IRL**. We use the FKL of f-IRL in our experiments. Since FKL is explicitly models a reward function with a neural network, when PAGAR is combined with FKL, the meta-algorithm Algorithm 1 can be directly implemented without changes except for letting $J_{IRL}$ be the FKL loss.

- **GAIL and VAIL**. We take additional steps to incorporate GAN-based algorithms, as they do not explicitly learn reward functions but instead train discriminators. When PAGAR is combined with GAIL, the meta-algorithm Algorithm 1 becomes Algorithm 2. When PAGAR is combined with VAIL, it becomes Algorithm 3. Both of the two algorithms are GAN-based IRL, indicating that both algorithms use Eq.1 as the IRL objective function. In these two cases, we use a neural network to approximate $D$, the discriminator in Eq.1. To get the reward function $r$, we follow Fu et al. [2018] and denote $r(s,a) = \log \left( \frac{\pi_A(a|s)}{D(s,a)} - \pi_A(a|s) \right)$ as mentioned in Section 1. Hence, the only difference between Algorithm 2 and Algorithm 1 is in the representation of the reward function. Regarding VAIL, since it additionally learns a representation for the state-action pairs, a bottleneck constraint $J_{IC}(D) \leq i_c$ is added where the bottleneck $J_{IC}$ is estimated from policy roll-outs. VAIL introduces a Lagrangian parameter $\beta$ to integrate $J_{IC}(D) - i_c$ in the objective function. As a result its objective function becomes $J_{IRL}(r) + \beta \cdot (J_{IC}(D) - i_c)$. VAIL not only learns the policy and the discriminator but also optimizes $\beta$. In our case, we utilize the samples from both protagonist and antagonist policies to optimize $\beta$ as in line 10 following Peng et al. [2019].

In our implementation, depending on the difficulty of the benchmarks, we choose to maintain $\lambda$ as a constant or update $\lambda$ with the IRL loss $J_{IRL}(r)$ in most of the continuous control tasks. In *HalfCheetah-v2* and all the maze navigation tasks, we update $\lambda$ by introducing a hyperparameter $\mu$. As described in the maintext, we treat $\delta$ as the target IRL loss of $J_{IRL}(r)$, i.e., $J_{IRL}(r) \leq \delta$.

---

**Algorithm 2** GAIL w/ PAGAR

**Input**: Expert demonstration $E$, discriminator loss bound $\delta$, initial protagonist policy $\pi_P$, antagonist policy $\pi_A$, discriminator $D$ (representing $r(s,a) = \log\left(\frac{\pi_A(a|s)}{D(s,a)} - \pi_A(a|s)\right)$), Lagrangian parameter $\lambda$, iteration number $i = 0$, maximum iteration number $N$

**Output**: $\pi_P$

1: **while** iteration number $i < N$ **do**
2:      Sample trajectory sets $\mathbb{D}_A \sim \pi_A$ and $\mathbb{D}_P \sim \pi_P$
3:      Estimate $J_{RL}(\pi_A; r)$ with $\mathbb{D}_A$
4:      Optimize $\pi_A$ to maximize $J_{RL}(\pi_A; r)$.
5:      Estimate $J_{RL}(\pi_P; r)$ with $\mathbb{D}_P$; $J_{\pi_A}(\pi_P; \pi_A, r)$ with $\mathbb{D}_P$ and $\mathbb{D}_A$;
6:      Optimize $\pi_P$ to maximize $J_{RL}(\pi_P; r) + J_{\pi_A}(\pi_P; \pi_A, r)$.
7:      Estimate $J_{PAGAR}(r; \pi_P, \pi_A)$ with $\mathbb{D}_P$ and $\mathbb{D}_A$
8:      Estimate $J_{IRL}(\pi_A; r)$ with $\mathbb{D}_A$ and $E$ by following the IRL algorithm
9:      Optimize $D$ to minimize $J_{PAGAR}(r; \pi_P, \pi_A) + \lambda \cdot max(J_{IRL}(r) + \delta, 0)$
10: **end while**
11: **return** $\pi_P$

---

**Algorithm 3** VAIL w/ PAGAR

**Input**: Expert demonstration $E$, discriminator loss bound $\delta$, initial protagonist policy $\pi_P$, antagonist policy $\pi_A$, discriminator $D$ (representing $r(s,a) = \log\left(\frac{\pi_A(a|s)}{D(s,a)} - \pi_A(a|s)\right)$), Lagrangian parameter $\lambda$ for PAGAR, iteration number $i = 0$, maximum iteration number $N$, Lagrangian parameter $\beta$ for bottleneck constraint, bounds on the bottleneck penalty $i_c$, learning rate $\mu$.

**Output**: $\pi_P$

1: **while** iteration number $i < N$ **do**
2:      Sample trajectory sets $\mathbb{D}_A \sim \pi_A$ and $\mathbb{D}_P \sim \pi_P$
3:      Estimate $J_{RL}(\pi_A; r)$ with $\mathbb{D}_A$
4:      Optimize $\pi_A$ to maximize $J_{RL}(\pi_A; r)$.
5:      Estimate $J_{RL}(\pi_P; r)$ with $\mathbb{D}_P$; $J_{\pi_A}(\pi_P; \pi_A, r)$ with $\mathbb{D}_P$ and $\mathbb{D}_A$;
6:      Optimize $\pi_P$ to maximize $J_{RL}(\pi_P; r) + J_{\pi_A}(\pi_P; \pi_A, r)$.
7:      Estimate $J_{PAGAR}(r; \pi_P, \pi_A)$ with $\mathbb{D}_P$ and $\mathbb{D}_A$
8:      Estimate $J_{IRL}(\pi_A; r)$ with $\mathbb{D}_A$ and $E$ by following the IRL algorithm
9:      Estimate $J_{IC}(D)$ with $\mathbb{D}_A, \mathbb{D}_P$ and $E$
10:      Optimize $D$ to minimize $J_{PAGAR}(r; \pi_P, \pi_A) + \lambda \cdot max(J_{IRL}(r) - \delta, 0) + \beta \cdot J_{IC}(D)$
11:      Update $\beta := \max\left(0, \beta - \mu \cdot \left(\frac{J_{IC}(D)}{3} - i_c\right)\right)$
12: **end while**
13: **return** $\pi_P$

---

In all the maze navigation tasks, we initialize $\lambda$ with some constant $\lambda_0$ and update $\lambda$ by $\lambda := \lambda \cdot \exp(\mu \cdot (J_{IRL}(r) - \delta))$ after every iteration. In *HalfCheetah-v2*, we update $\lambda$ by $\lambda := max(\lambda_0, \lambda \cdot \exp(\mu \cdot (J_{IRL}(r) - \delta)))$ to avoid $\lambda$ being too small. Besides, we use PPO Schulman et al. [2017] to train all policies in Algorithm 2 and 3.

**Offline RL Setting.** We incorporate PAGAR with RECOIL Sikchi et al. [2024]. The original RECOIL algorithm does not learn a reward function but learns a $Q$, $V$ value functions and a policy $\pi$ with neural networks. In order to combine PAGAR with RECOIL, we made the following modification to RECOIL:

- Instead of learning the $Q$ value function, we explicitly learn a reward function $r : \mathbb{S} \times \mathbb{A} \times \mathbb{S} \to \mathbb{R}$, which takes the current state $s$, action $a$ and the next state $s'$ as input, and outputs a real number as the reward

- We use the same loss function as that for optimizing $Q$ in RECOIL to optimize $r$ by replacing $Q(s,a)$ with $r(s,a,s') + \gamma V(s')$ for every $(s,a,s')$ sampled from an offline dataset $\mathbb{D}$. We denote this loss function for $r$ as $\mathcal{J}_{IRL}(r)$ as in Eq.35 where $S$ is an offline trajectory sample set of some sub-optimal behavioral policy, $d_{mix}^{E,S}$ indicates a mixture between $S$ and $E$ sets,

---

**Algorithm 4** RECOIL w/ PAGAR

---

**Input**: Expert demonstration $E$, behavioral policy sample set $S$, reward function $r$, value function $V$, initial protagonist policy $\pi_P$, antagonist policy $\pi_A$, maximum iteration number $N$.

**Output**: $\pi_P$

1: **while** iteration number $i < N$ **do**
2:      Train $r$ using $\min_r J_{IRL}(r) + 0.001 \cdot J_{PAGAR}(r)$
3:      Train $V$ using $\min_V J_V(V)$
4:      Train $\pi_A$ using $\max_{\pi_A} J_{RL}(\pi_A)$
5:      Train $\pi_P$ using $\max_{\pi_A} J_{RL}(\pi_P)$
6: **end while**
7: **return** $\pi_P$

---

and $\beta$ is the *mixing ratio* as defined in Sikchi et al. [2024].

$$\mathcal{J}_{IRL}(r) := \beta \left( \mathbb{E}_{(s,a,s') \sim \mathcal{S}}[r(s,a,s') + \gamma V(s')] - \mathbb{E}_{(s,a,s') \sim E}[r(s,a) + \gamma V(s')] \right) + \\ 0.25 \cdot \mathbb{E}_{(s,a,s') \sim d_{mix}^{E,S}} \left[ \left( r(s,a,s') \right)^2 \right] \quad (35)$$

- We use the same loss function for optimizing the value function $V$ as in RECOIL to still optimize $V$, except for replacing the target $Q(s,a)$ with target $r(s,a,s') + \gamma V(s')$ for every $(s,a,s')$ experience sampled from the offline dataset. Consequently, we have the loss function for $V$ as defined in Eq.36 where $\sigma$ is a conservatism parameter defined in Sikchi et al. [2024].

$$\mathcal{J}_V(V) := \mathbb{E}_{(s,a,s') \sim d_{mix}^{E,S}} \Big[ \exp\left( (r(s,a,s') + \gamma V(s') - V(s))/\sigma \right) + \\ \left( r(s,a,s') + \gamma V(s') - V(s) \right) \Big] \quad (36)$$

- Instead of learning a single policy as in RECOIL, we learn a protagonist and antagonist policies $\pi_P$ and $\pi_A$ by using the same SAC-like policy update rule as in RECOIL, except for replacing $Q(s,a)$ with $r(s,a,s') + \gamma V(s')$ for every $(s,a,s')$ experience sampled from the offline dataset. The objective function for learning $\pi_A$ and $\pi_P$ are defined in Eq.37.

$$\mathcal{J}_{RL}(\pi) := \mathbb{E}_{(s,a,s') \sim d_{mix}^{E,S}} \Big[ \exp\left( (r(s,a,s') + \gamma V(s') - V(s)) \right) \log \pi(a|s) \Big] \quad (37)$$

- With some heuristic, we construct a PAGAR-loss as follows.

$$\mathcal{J}_{PAGAR}(r) \quad := \quad \mathbb{E}_{(s,a,s') \sim E} \left[ r(s,a,s') \cdot \exp(clip(max(0, \log \frac{\pi_P(a|s)}{\pi_A(a|s)}), -1, 1) \right] + \\ \mathbb{E}_{(s,a,s') \sim S} \left[ r(s,a,s') \cdot \exp(clip(min(0, \log \frac{\pi_A(a|s)}{\pi_P(a|s)}), -1, 1) \right] \quad (38)$$

- For simplicity, we multiply this PAGAR-loss $\mathcal{J}_{PAGAR}$ with a fixed Lagrangian parameter $\lambda = 1e - 3$ and add it to the aforementioned loss $\mathcal{J}_{IRL}$ for optimizing $r$.

We summarize this RECOIL w/ PAGAR algorithm in Algorithm 4.

## C Experiment Details

This section presents some details of the experiments and additional results.

### C.1 Experimental Details

Hardware. All experiments are carried out on a quad-core i7-7700K processor running at 3.6 GHz with a NVIDIA GeForce GTX 1050 Ti GPU and a 16 GB of memory. **Network Architectures**. Our algorithm involves a protagonist policy $\pi_P$, and an antagonist policy $\pi_A$. In our implementation, the two policies have the same structures. Each structure contains two neural networks, an actor network, and a critic network. When associated with GAN-based IRL, we use a discriminator $D$ to represent the reward function as mentioned in Appendix B.4.

- **Protagonist and Antagonist policies**. We prepare two versions of actor-critic networks, a fully connected network (FCN) version, and a CNN version, respectively, for the Mujoco and Mini-Grid benchmarks. The FCN version, the actor and critic networks have 3 layers. Each hidden layer has 100 neurons and a $tanh$ activation function. The output layer output the mean and standard deviation of the actions. In the CNN version, the actor and critic networks share 3 convolutional layers, each having 5, 2, 2 filters, $2 \times 2$ kernel size, and $ReLU$ activation function. Then 2 FCNs are used to simulate the actor and critic networks. The FCNs have one hidden layer, of which the sizes are 64.

- **Discriminator $D$ for PAGAR-based GAIL in Algorithm 2**. We prepare two versions of discriminator networks, an FCN version and a CNN version, respectively, for the Mujoco and Mini-Grid benchmarks. The FCN version has 3 linear layers. Each hidden layer has 100 neurons and a $tanh$ activation function. The output layer uses the $Sigmoid$ function to output the confidence. In the CNN version, the actor and critic networks share 3 convolutional layers, each having 5, 2, 2 filters, $2 \times 2$ kernel size, and $ReLU$ activation function. The last convolutional layer is concatenated with an FCN with one hidden layer with 64 neurons and $tanh$ activation function. The output layer uses the $Sigmoid$ function as the activation function.

- **Discriminator $D$ for PAGAR-based VAIL in Algorithm 3**. We prepare two versions of discriminator networks, an FCN version and a CNN version, respectively, for the Mujoco and Mini-Grid benchmarks. The FCN version uses 3 linear layers to generate the mean and standard deviation of the embedding of the input. Then a two-layer FCN takes a sampled embedding vector as input and outputs the confidence. The hidden layer in this FCN has 100 neurons and a $tanh$ activation function. The output layer uses the $Sigmoid$ function to output the confidence. In the CNN version, the actor and critic networks share 3 convolutional layers, each having 5, 2, 2 filters, $2 \times 2$ kernel size, and $ReLU$ activation function. The last convolutional layer is concatenated with a two-layer FCN. The hidden layer has 64 neurons and uses $tanh$ as the activation function. The output layer uses the $Sigmoid$ function as the activation function.

**Hyperparameters** The hyperparameters that appear in Algorithm 3 and 3 are summarized in Table 2 where we use N/A to indicate using $\delta^*$, in which case we let $\mu = 0$. Otherwise, the values of $\mu$ and $\delta$ vary depending on the task and IRL algorithm. The parameter $\lambda_0$ is the initial value of $\lambda$ as explained in Appendix B.4.

| Parameter | Continuous Control Domain | Partially Observable Domain |
|---|---|---|
| Policy training batch size | 64 | 256 |
| Discount factor | 0.99 | 0.99 |
| GAE parameter | 0.95 | 0.95 |
| PPO clipping parameter | 0.2 | 0.2 |
| $\lambda_0$ | 1e3 | 1e3 |
| $\sigma$ | 0.2 | 0.2 |
| $i_c$ | 0.5 | 0.5 |
| $\beta$ | 0.0 | 0.0 |
| $\mu$ | VAIL(HalfCheetah): 0.5; others: 0.0 | VAIL: 1.0; GAIL: 1.0 |
| $\delta$ | VAIL(HalfCheetah): 1.0; others: N/A | VAIL: 0.8; GAIL: 1.2 |

Table 2: Hyperparameters used in the training processes

**Expert Demonstrations.** Our expert demonstrations all achieve high rewards in the task. The number of trajectories and the average trajectory total rewards are listed in Table 3.

### C.2 Additional Results

**Continuous Tasks with Non-Binary Outcomes** We test PAGAR-based IRL in 5 Mujuco tasks where the task objectives do not have binary outcomes. We append the results in three Mujoco benchmarks: *Walker2d-v2*, *HalfCheeta-v2*, *Hopper-v2*, *InvertedPendulum-v2* and *Swimmer-v2* in Figure 6 and 7. Algorithm 1 performs similarly to VAIL and GAIL in those two benchmarks. The results show that PAGAR-based IL takes fewer iterations to achieve the same performance as the baselines. In particular, in the *HalfCheetah-v2* task, Algorithm 1 achieves the same level of performance compared

| Task | Number of Trajectories | Average Tot.Rewards |
|---|---|---|
| Walker2d-v2 | 10 | 4133 |
| HalfCheetah-v2 | 100 | 1798 |
| Hopper-v2 | 100 | 3586 |
| InvertedPendulum-v2 | 10 | 1000 |
| Swimmer-v2 | 10 | 122 |
| DoorKey-6x6-v0 | 10 | 0.92 |
| SimpleCrossingS9N1-v0 | 10 | 0.93 |

Table 3: The number of demonstrated trajectories and the average trajectory rewards

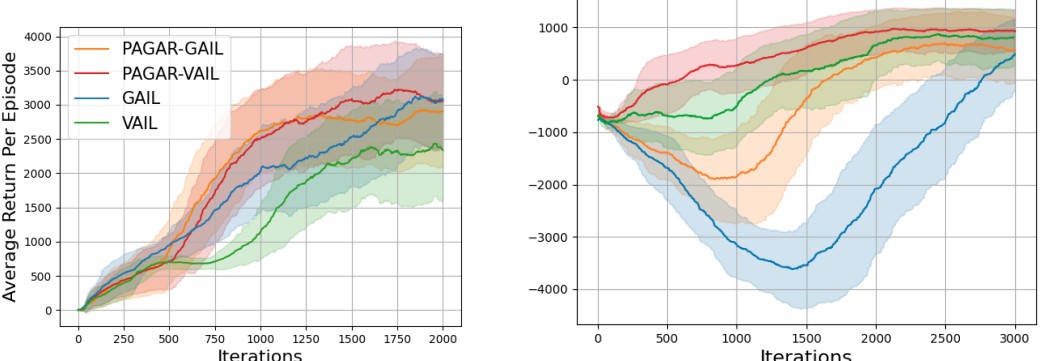

Figure 6: (Left: Walker2d-v2. Right: HalfCheeta-v2) The $y$ axis indicates the average return per episode.

with GAIL and VAIL by using only half the numbers of iterations. IQ-learn does not perform well in Walker2d-v2 but performs better than ours and other baselines by a large margin.

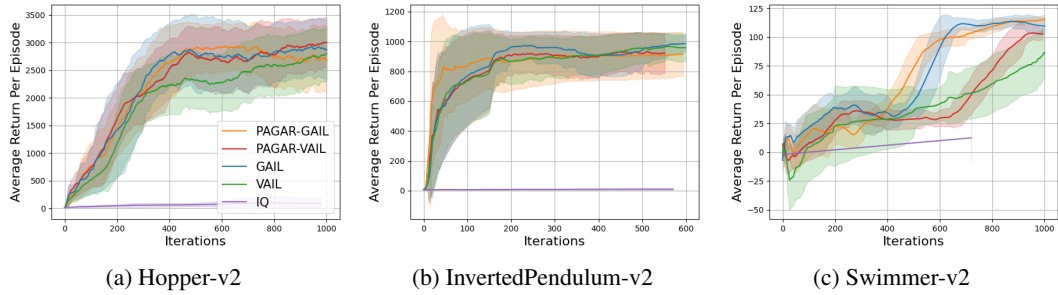

(a) Hopper-v2        (b) InvertedPendulum-v2        (c) Swimmer-v2

Figure 7: Comparing Algorithm 1 with baselines. The suffix after each 'PAGAR-' indicates which IRL algorithm is utilized in Algorithm 1. The $y$ axis is the average return per step. The $x$ axis is the number of iterations in GAIL, VAIL, and ours. The policy is executed between each iteration for 2048 timesteps for sample collection. One exception is that IQ-learn updates the policy at every timestep, making its actual number of iterations 2048 times larger than indicated in the figures.

## C.3 Influence of Reward Hypothesis Space

In addition to the *DoorKey-6x6-v0* environment, we also tested PAGAR-GAIL and GAIL in *SimpleCrossingS9N2-v0* environment. The results are shown in Figure 6 and 8.

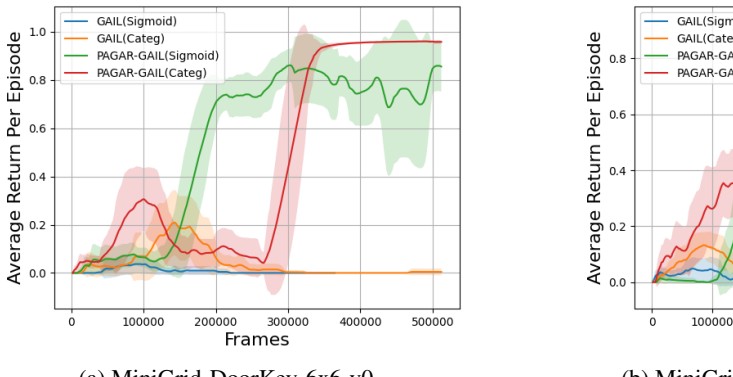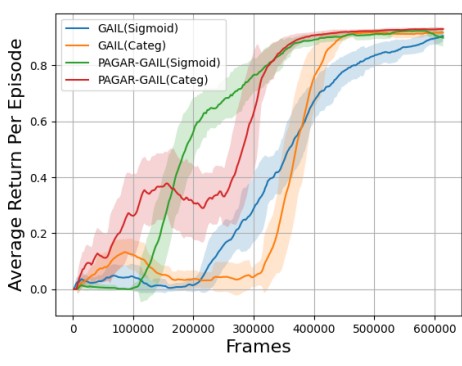

(a) MiniGrid-DoorKey-6x6-v0                    (b) MiniGrid-SimpleCrossingS9N2-v0

Figure 8: Comparing Algorithm 1 with baselines. The prefix 'protagonist_GAIL' indicates that the IRL algorithm utilized in Algorithm 1 is the same as in GAIL. The '_Sigmoid' and '_Categ' suffixes indicate whether the output layer of the discriminator is using the $Sigmoid$ function or Categorical distribution. The $x$ axis is the number of sampled frames. The $y$ axis is the average return per episode.

