# OpenReview forum: "Rethinking Inverse Reinforcement Learning: from Data Alignment to Task Alignment"
_NeurIPS.cc/2024/Conference — NeurIPS 2024 poster_

### Official Review · Reviewer_fY7t · 2024-07-10

**Soundness:** 3
**Presentation:** 2
**Contribution:** 3
**Rating:** 5
**Confidence:** 3

**Summary:**

This paper aims to alleviate the reward-misalignment issue by introducing task-related objective. The authors theoretically derive sufficient conditions to mitigate the task-reward misalignment issue and design algorithms accordingly. Theoretical analysis of the algorithm is provided to guarantee the reward learning improvement. Two discrete grid-world experiments are provided to empirically validate the proposed algorithms.

**Strengths:**

1. The idea of introducing task objective to help alleviate the reward ambiguity issue is novel and interesting.
2. This paper has a complete and solid theoretical structure, which is also the biggest strength.
3. The empirical results show superior performance over GAIL and VAIL.

**Weaknesses:**

The main weakness lies in the empirical evaluation:
1. The environment setups are too simple. The paper only studies two discrete settings where continuous environments are missing, e.g., MuJoCo.

2. The experiment only has two comparison baselines, which is not convincing enough to show the superiority of the proposed algorithm. The paper mentions the IRL based IL and aims to alleviate the reward ambiguity issue, why not directly use some IRL algorithms as baselines to compare, e.g., MaxEnt IRL, ML-IRL, f-IRL, etc.

I know that this paper is a theory paper where the theoretical analysis is the major contribution and should be valued the most. However, the empirical evaluation should at least meet the minimum standard of the current majority, and two discrete gridworlds with fewer than 100 states apparently do not. I highly suggest the authors to add some continuous environments and baselines.

**Questions:**

Please see weakness.

**Limitations:**

The author discusses the limitation in the paper, which is reasonable.

---

> ### Author Rebuttal · Authors · 2024-08-07
>
> > ## ... continuous environments are missing, e.g., MuJoCo.
>
> We have included continuous control tasks in Appendix C.2.
>
> > ## ... environment setups are too simple ... fewer than 100 states
>
> The MiniGrid environments are challenging benchmarks for RL and IRL although they may look deceptively simple. The agent only makes **partial observation** of the environment -- the 7x7 cells in front of it. **Furethermore, all the objects/walls/doors are randomly placed in every episode.** As a result, it has **far more than 100 states**. In fact, due to its massive state space, it has become a standard benchmark for exploration-driven RL algorithms. For instance, AGAC [1],  the SOTA exploration-driven RL technique, uses this environment as the main benchmark. It takes less than **10 million steps** to attain high performance for a task in VizDoom (a more feature-rich, perception-based environment), but takes **200 million steps** to only achieve a moderate performance on one of the MiniGrid environments. The MiniGrid Github repository also has a long list of papers that use MiniGrid in their empirical evaluations.
>
>
>
> > ## ... not directly use some IRL algorithms as baselines to compare, e.g., MaxEnt IRL, ML-IRL, f-IRL, etc.
>
> Our approach, named PAGAR-based IL, is an IL algorithm that uses existing IRL-based IL algorithm to specify the set $R_{E,\delta}$ as mentioned in line 172. We design the experiments mainly to show the compatibility of our algorithm with the IRL parts of the existing IRL-based IL benchmarks, and whether our algorithm can improve the imitation performance by using the same IRL parts as those of the benchmarks.
>
> However, we appreciate the reviewrs' suggestions on trying other IRL algorithms. Therefore, we have incorporated our algorithm with the most advanced IRL/IL algorithm, RECOIL [2].  We have attached our results in our general response to all reviewers. Please refer to it for details.
>
>
> [1] Flet-Berliac, Y., Ferret, J., Pietquin, O., Preux, P.,  Geist, M. Adversarially guided actor-critic. ICLR 2021
>
> [2] Sikchi et al.; Dual RL: Unification and new methods for reinforcement and imitation learning, ICLR 2024

---

> ### Author Response · Authors · 2024-08-12
> **Suppplementary Experimental Results: Comparing PAGAR with f-IRL**
>
> Thank you once again for your valuable comments. We hope that our previous response has addressed most of your concerns. To further support our rebuttal, we would like to share some additional experimental results that directly address the concerns that you raised. We would greatly appreciate it if you could re-evaluate our submission based on this response, or let us know if you have any other questions. Thank you!
>
> > ##  (cont'd) ... directly use some IRL algorithms as baselines to compare, e.g., MaxEnt IRL, ML-IRL, f-IRL, etc...
>
> We have conducted experiments combining PAGAR with **f-IRL**, and the results are summarized in our most recent **General Response**. This combination has achieved higher performance than **f-IRL** baseline with fewer exploration iterations in the environment across multiple standard IRL/IL benchmarks. These results will be presented in the appropriate format in the revised version of the paper.

---

> ### Author Response · Authors · 2024-08-12
> **Please let us know if further clarification or additional evaluations would help in revisiting the score**
>
> Dear Reviewer,
>
> Thank you once again for your valuable comments.
>
> We hope that our explanation, along with the experiments in **continuous control tasks** with GAIL/VAIL in Appendix C.2, and the **additional experiments** with  **f-IRL**  (provided in the **general responses**) as well as the **SOTA** offline IRL/IL baseline **RECOIL**  (also provided in the **general responses**) have addressed your concerns about experimental evaluation.
>
> ## Please let us know if further clarification or additional evaluations would help in revisiting the score.
>
> Your support would make a significant difference.
>
> Thank you!
>
> Best,
>
> Authors

---

> > ### Comment · Reviewer_fY7t · 2024-08-12
> >
> > Thanks for the response. I'll keep the current rating given that  my current rating is already positive.

---

> ### Author Response · Authors · 2024-08-14
>
> Thank you for your thoughtful review and for taking the time to consider our response. We’re pleased that our clarification was helpful.
>
> We would like to reiterate our first point from the initial rebuttal that **MiniGrid** environment we use is a well-established RL benchmark with **its own publication in NeurIPS 2023** [1]. As mentioned in our initial response, this environment is particularly complex even for SOTA IRL/IL algorithms due to its **massive state space (much more than 100 states)**.
>
> #### Given that the current score is quite close to the borderline, we would be grateful if you could revisit our explanations and the additional results.
>
> In our revised version, we will not only include the additional experiments with **f-IRL** and **RECOIL** on continuous control tasks but also focus on presenting the main idea of the paper as follows to ensure our contributions are effectively communicated:
>
> * **Goal**: Learn a policy to fulfill an unknown task that fits the description in Definition 1.
> * **Insight**: Achieving high utility under task-aligned reward functions is essential for task fulfillment.
> * **Problem**: The specific task-aligned reward function is unknown.
> * **Solution**: By treating expert demonstrations as weak supervision, we learn a set of reward functions that encompass task-aligned rewards and then train a policy to achieve high utility across this set of reward functions.
>
> Of course, we are happy to address any further questions.
>
>
> [1] Chevalier-Boisvert, Maxime, et al. "Minigrid & miniworld: Modular & customizable reinforcement learning environments for goal-oriented tasks." NeurIPS 2023

---

### Official Review · Reviewer_QeNS · 2024-07-11

**Soundness:** 2
**Presentation:** 2
**Contribution:** 2
**Rating:** 5
**Confidence:** 4

**Summary:**

This paper introduces a novel approach to inverse RL (IRL) by identify weaknesses in current IRL algorithms and proposing the respective solution:  optimizing for *task-related* reward functions. The authors provide a clean theoretical framework for task-related rewards, but lack applicability in practice from my perspective (see below).

**Strengths:**

- inverse RL with task-aligned reward functions is an interesting research direction.
- the paper presents a good theoretical basis for future work.
- the theoretical work looks plausible to me (only reading the main paper).

**Weaknesses:**

- The storyline of the paper is not clear. I do not see the emphasis on task-relatedness (see comments below).
- The paper is cluttered in theory, making it hard to follow. Instead, more focus should be on the story, the motivation, and the experiments.
- The algorithm looks quite complex; not sure how scaleable it is. (suggestions below)
- The experimental campaign is concentrated on simple tasks; more complex tasks could prove scalability.

**Questions:**

- How is $\Pi_{acc}$ actually spanned? It is not really talked about it, which made the following sentence a bit confusing “It is crucial to recognize that rE might not meet the task-aligned reward function criteria specified in Definition 2, even though its optimal policy πE is acceptable. ”
- Looking at Def 2, the “task relatedness” is only induced by the definition of $\Pi_{acc}$, is this correct?

**Limitations:**

My biggest problem is that I dont see why this paper (in practice) proposes a technique for task-aligned reward functions. $k$ and $\Pi_{acc}$ define task-relatedness in this framework to my understanding. And as the authors say:
“As the task alignment of each reward function typically remains unknown in IRL settings, this paper proposes treating k as an adjustable parameter – starting with a small k and adjusting based on empirical learning outcome, allowing for iterative refinement for alignment with task requirements.” So what defines task-relatedness is actually set as a hyperparameter in this framework. For me, the method sounds more like a regularization technique for finding less overfitting reward functions. It looks like the auhors also say so:
“Given the uncertainty in identifying which reward function is aligned, our solution is to train a policy to achieve high utilities under all reward functions in RE,δ to satisfy the conditions in Definition 3.” --> all reward functions does not mean task-related reward functions. Hence, my claim of being a regularization technique rather than an active search for task-related search. For me, this is breaking the storyline of the paper.

Further points:
- In section 4.1, consider clarifying the difference between r* and r_E. It took me a bit do understand it.
- Algorithm 1, line 4: I guess it should be $\pi_P$ instead of $\pi_A$.
- an ablation on $\delta$ would be interesting
- add more complex environments. One good fit could be the new locomotion benchmark LocoMuJoCo. It provides motion capture datasets mapped towards different robots. So, there is a dynamics mismatch between the expert dataset and the environment. Also, all baselines (Gail, Vail, IQ-Learn) are already available.
- Since you have been using IQ-Learn [1] as a baseline, I would also add it together with similar methods using implicit rewards [2, 3] to the related work section with some discussion. Otherwise, it looks out of place in the experiment section.
- General remark: shorten the theory and make a more comprehensive experimental campaign.


[1] Garg et al.; IQ-Learn: Inverse soft-Q Learning for Imitation \
[2] Al-Hafez et al.; LS-IQ: Implicit Reward Regularization for Inverse Reinforcement Learning \
[3] Sikchi et al.;  Dual RL: Unification and new methods for reinforcement and imitation learning

---

> ### Author Rebuttal · Authors · 2024-08-07
>
> > ## The experimental campaign is concentrated on simple tasks... more complex tasks could prove scalability.
>
> The MiniGrid tasks we employ are challenging benchmarks for RL and IRL although they may look deceptively simple. The agent only makes **partial observation** of the environment -- the 7x7 cells in front of it. **All the objects/walls/doors are randomly placed after every episode.** As a result, it has prohibitively large state space, which has made it a standard benchmark for exploration-driven RL algorithms. For instance, AGAC [1],  the SOTA exploration-driven RL technique, uses this environment as the main benchmark. It takes less than **10 million steps** to attain high performance for a task in VizDoom (a more feature-rich, perception-based environment), but takes **200 million steps** to only achieve a moderate performance on one of the MiniGrid environments. The MiniGrid Github repository also has a long list of papers that use MiniGrid in their empirical evaluations.
>
>
> Besides discrete navigation tasks, we **have also included continuous control tasks, e.g., Mujoco, in Appendix C.2.** Furthermore, we have experimentd on **D4RL dataset** and attached results later in this rebuttal.
>
>
> > ## How is $\Pi_{acc}$ spanned ...
>
> $\Pi_{acc}$ is the set of policies that can accomplish the `task`. How it is spanned depends on what the underlying `task` is. For instance, in some reachability tasks, any policy that can reach a goal state with a higher probability than a threshold can be deemed acceptable. As mentioned in our general response, our definition of `task` is inspired by the NeurIPS 2021 best paper [2], which shows that this definition can characterize a wide range of real-world tasks.
>
> > ## Definition 2 task-relatedness is only induced by the definition of $\Pi_{acc}$, is this correct?
>
> Whether a reward function is `task-aligned` is determined by $\Pi_{acc}$: a reward function $r$ is `task-aligned` if **all** the acceptable policies from $\Pi_{acc}$ achieve higher utilities under $r$ than **all** those not from $\Pi_{acc}$. Even if the expert policy $\pi_E$ belongs inside $\Pi_{acc}$, it does not guarantee that its optimal reward function $r_E$ is `task-aligned`.
>
> > ## ... hyperparameter $k$ ... what defines task-relatedness is actually set as a hyperparameter in this framework.
>
> The hyperparameter $k$ **does not determine task-relatedness** ('task-alignment'). It is used to determine the **size of the candidate reward functions set $R_{E,k}$**.
>
> One way to assist understanding the role of $k$ is to draw an analogy to adversarially robust machine learning: the learning model should maintain high accuracy/low error not only at the training data points but also in the 'vicinity' of these points. As the range of adversarial perturbation is generally unknown, the range of this 'vicinity' is often set as a hyperparameter. But this hyperparamter does not determine which point is prone to be attacked.
>
> > ## ...‘finding less overfitting reward functions’ ... regularization technique rather than an active search for task-aligned ...
>
> We do not actively search for task-aligned reward functions. Our goal is to find an acceptable policy. However, **the concept of task alignment is crucial as it is the backbone of Theorem 1** which justifies two key aspects of our approach:
>
>     1. Collect a set of reward functions, including IRL's sub-optimal rewards, to encompass task-aligned rewards, and
>     2. Train a policy to attain high utility across this reward set.
>
> We have not seen literature supporting that **"finding less overfitting reward functions"** justifies either of these aspects.
>
> Our formulation in Equation 3 is a **constrained optimization problem**, distinct from merely adding **regularization** to IRL/IL’s loss. The objective is to find a policy that minimizes worst-case regret while the constraint allows us to consider all reward functions with low IRL loss.
>
> > ## ... breaking the storyline ...
>
> The main idea of this paper can be summarized as follows:
> * **Goal**: Learn a policy to fulfill an unknown task that fits the description in Definition 1.
> * **Insight**: Only achieving high utility under task-aligned reward functions can ensure task fulfillment.
> * **Problem**: Unknown which reward function is task-aligned
> * **Solution**:
>     1. Treating expert demos as weak supervision, learn a set of reward functions to encompass task-aligned rewards
>     2. Train a policy to achieve high utility across this set of reward functions
>
> > ## the difference between r* and r_E
>
> $r^*$ is the optimal reward function solved via IRL while $r_E$ is the expert reward function.
>
> > ## line 4: I guess it should be $\pi_P$ instead of $\pi_A$
>
> Yes, we meant to update $\pi_P$ instead of $\pi_A$ in line 4.
>
> > ## new locomotion benchmark LocoMuJoCo
>
> As mentioned earlier, we included continuous control tasks, e.g., Mujoco, in Appendix C.2. However, we appreciate the suggestion and will conduct experiments on LocoMujoco and update the results in the paper.
>
> > ## an ablation on $\delta$ would be interesting
>
> In our experiments, we observe that when $\delta$ is too large, the algorithm does not induce any meaningful results. This has been described in line 164. We plan to include this observation in the updated version of the paper.
>
>
> > ## ...similar methods using implicit rewards
>
> We will add them to our related work section. In our general response, we have attached our experiemtns on combining our framework with RECOIL in [3]. This marks that compatibility with **implicit reward** and **offline IL** as a new feature of our method.
>
>
> [1] Flet-Berliac, Y., Ferret, J., Pietquin, O., Preux, P.,  Geist, M. Adversarially guided actor-critic. ICLR 2021
>
> [2] Abel, David, et al. On the expressivity of markov reward, NeurIPS 2021
>
> [3] Sikchi et al.; Dual RL: Unification and new methods for reinforcement and imitation learning, ICLR 2024

---

> > ### Comment · Reviewer_QeNS · 2024-08-13
> >
> > I appreciate the author's response.
> >
> > I still disagree with the authors about the complexity of MiniGrid environment, and very much hope to see more complex experiments in the final paper (e.g., LocoMujoco). It seems also like the other reviewers similarily found the experiments too simple.
> >
> > The storyline of the paper became a bit clearer thanks to your additional explanation. **Please consider writing it more clearly in the final version of the paper.** (similar to your response)
> >
> > > A significant characteristic of these tasks is the presence of a binary success or failure criterion between policies. Our method aims to learn an acceptable policy that succeeds in such a task, rather than an optimal policy under some 'ground truth' reward function. As a result, our approach's effectiveness is more pronounced in tasks where success and failure are distinctly binary, such as in MiniGrid, than in tasks with more quantitative performance metrics, such as continuous control tasks.
> >
> > This is not only a characteristic but a limitation of this algorithm! Please make this clear in the paper.
> >
> > Given that the promised updates are yet to be included to the paper, I will only slightly raise my score.

---

> ### Author Response · Authors · 2024-08-12
> **Reminder for Re-evaluation and Further Discussion**
>
> Dear reviewer,
>
> Thank you once again for your valuable comments. We hope that our explanation and the new experimental results on **RECOIL** and **f-IRL** with **continuous control benchmarks** (provided in the **General Responses**) have addressed your concerns. We would greatly appreciate it if you could re-evaluate our submission based on this response, or let us know if you have any further questions. Thank you!
>
> Best,
>
> Authors

---

> ### Author Response · Authors · 2024-08-14
>
> We sincerely thank the reviewer for raising our score and appreciate the constructive feedback. We will incorporate the suggested improvements as outlined in our response.
>
> In addition, we would like to highlight that the **MiniGrid** environment we use is a well-established RL benchmark with **its own publication in NeurIPS 2023** [1]. As mentioned in our initial response, this environment is particularly complex, even for SOTA IRL/IL algorithms due to its **massive state space**.
>
> Also, we need to clarify that our **Task** definition in Definition 1 is not intended to **limit the scope of benchmarks**. This definition inherits concepts from [2] . This abstract task definition is very general and encompasses many real-world examples, as also demonstrated in [2]. In fact, it is designed to address considerations from dimensions beyond simple classifications such as ‘continuous/discrete,’ ‘finite/infinite states,’ or ‘fixed/infinite horizon.’  We now demonstrate that our **Task** definition is expressive enough to characterize standard RL tasks (learning the optimal policy from a reward function $r$).
>
> > Proof: Given a reward function $r$ and a policy hypothesis set $\Pi$, the corresponding **Task** is $(\Pi, \preceq_{task}, \Pi_{acc})$ where the partial order $\preceq_{task}$ is defined as $\forall \pi_1,\pi_2\in \Pi,\pi_1\preceq \pi_2 \Leftrightarrow U_r(\pi_1)\leq U_r(\pi_2)$, and $\Pi_{acc}=\\{\pi|\pi'\preceq_{task} \pi\ \forall \pi'\in \Pi\\}$ if only considering the optimal policies.
>
> Our theories are based on this task definition.  Our algorithm can work on a variety of standard RL benchmarks, and our experimental results in MiniGrid and continuous control have demonstrated this.
>
> [1] Chevalier-Boisvert, Maxime, et al. "Minigrid & miniworld: Modular & customizable reinforcement learning environments for goal-oriented tasks." NeurIPS 2023
>
> [2] Abel, David, et al. On the expressivity of Markov reward, NeurIPS 2021

---

### Official Review · Reviewer_MkJQ · 2024-07-14

**Soundness:** 3
**Presentation:** 3
**Contribution:** 3
**Rating:** 7
**Confidence:** 2

**Summary:**

This paper addresses the reward misalignment in inverse reinforcement learning and introduces a novel approach to tackle this problem: PAGAR. It introduces a protagonist and an antagonist policy and treats the expert demonstrations as weak supervision to derive a set of reward functions rather than a single reward function. The reward functions and policies are used in a min-max scheme to minimize the regret induced by the two policies. The antagonist policy should maximize the regret whereas the protagonist policy should minimize it. PAGAR is used on adversarial imitation learning settings (more specifically on top of GAIL and VAIL) in combination with PPO on two discrete domain tasks.

**Strengths:**

* The problem of solving the underlying tasks instead of adapting to the expert data is very relevant.
* Even though the idea takes inspiration from UED, the proposed approach is novel and significant for the IRL community.
* The authors provide theoretical results.

**Weaknesses:**

* Even though the relationship and inspiration to UED is mentioned once, the basis of UED should be explained more.
* The experimental evaluation is limited. E.g. the experiments only contain environments with discrete action spaces. Since GAIL and VAIL as well as PPO work in continuous action spaces, it should not be a problem to include additional experiments on continuous control tasks.
* Due to the introduced notation and presented theoretical results, the paper is difficult to follow. Even though the appendix already includes most of the theoretical derivations and proofs, some parts of the main paper could still be included there to give more space for explanation, intuition and experiments. E.g. section 5.1. could be moved to the appendix or integrated in a reduced version in the main text.

* Minor:
- Missing } in line 270.
- Line 4 in algorithm 1: It should be "update $\pi_{P}$ to maximize..." instead of $\pi_{A}$ ?

**Questions:**

See weaknesses.

**Limitations:**

Even though the authors mention a small amount of limitations in the checklist, I think that setting the parameter
$k$ is not the only limitation of the method (which is the only limitation addressed in the main text). An individual section about the limitations of the approach is missing.

---

> ### Author Rebuttal · Authors · 2024-08-07
>
> > ## the basis of UED should be explained more.
>
>
> Our Eq. 3, which minimizes the worst-case Protagonist Antagonist Induced Regret $Regret$, is inspired by UED in [1]. However, the core of our approach focuses on learning policy with a set of candidate reward functions rather than searching for diverse environments to test the robustness of policies as in [1]. We have developed **novel theories for reward search**.
>
> Furthermore, as mentioned in line 47~52, our contribution is **three-fold**. Besides formulating Eq. 3 for reward search, our **two other contributions are independent of UED**.
> * Our first contribution in establishing the concepets and properties of task-alignment that ultimately lead us to Theorem 1, which justifies the usage of Eq.3.
> * Our third contribution is to develope a practical implementation for solving MinimaxRegret in Section 5.1.
>
> Therefore, we cited [1] but omitted its details to focus on the distinct aspects of our work.
>
>
> > ## ...additional experiments on continuous control tasks
>
> In our main text (lines 226-234) and in the general response, we have explained why we did not focus on continuous tasks.
>
> However, we did include **continuous control tasks in Appendix C.2**. Furthermore, we have conducted additional experiments on Offline IL with **D4RL continuous control datasets**. The details can be found in our general response
>
>
> >  ## ...E.g. section 5.1. could be moved to the appendix or integrated in a reduced version in the main text.
>
> We intended for Section 5.1, especially Proposition 2, to highlight the difference between PAGAR-based IL and IRL-based IL: IRL-based IL uses a single reward function derived from an inner optimization problem for policy training, whereas our method employs a mixture of reward functions for policy training. We will consider restructuring our manuscript to provide better clarification rather than presenting all the information as it currently stands.
>
> > ## Missing } in line 270.
>
> The } is in the middle of the equation.
>
>
> [1] Dennis, M, et al., Emergent Complexity and Zero-shot Transfer via Unsupervised Environment Design, NeurIPS 20202.

---

> > ### Comment · Reviewer_MkJQ · 2024-08-13
> >
> > Thank you for taking the time to respond to my review and for addressing my questions. Furthermore, I thank the authors for mentioning the results in the appendix and for presenting additional results in the Author Rebuttal. I must have missed the results in the appendix.
> >
> > Therefore, I raised my rating to 7.
> >
> > Regarding UED: I never doubted that two of your contributions are independent of UED. I only want to have a deeper explanation of UED in your paper since your second contribution is related (as you mention yourself).

---

> ### Author Response · Authors · 2024-08-12
> **Your support would make a significant difference.**
>
> Dear Reviewer,
>
> Thank you once again for your valuable comments.
>
> In our revised version, we will focus on presenting the main idea of the paper as follows to ensure our contributions are effectively communicated:
>
> * **Goal**: Learn a policy to fulfill an unknown task that fits the description in Definition 1.
> * **Insight**: Achieving high utility under task-aligned reward functions is essential for task fulfillment.
> * **Problem**: The specific task-aligned reward function is unknown.
> * **Solution**: By treating expert demonstrations as weak supervision, we learn a set of reward functions that encompass task-aligned rewards and then train a policy to achieve high utility across this set of reward functions.
>
> We also hope that our experiments with GAIL/VAIL  in **continuous control tasks** in Appendix C.2, and the **additional experiments** with the more recent IRL/IL baseline **f-IRL**  (provided in the **general responses**) and SOTA offline IL technique **RECOIL**  (also provided in the **general responses**) have addressed your concerns about experimental evaluation.
>
> Please let us know if further clarification or additional evaluations would help in revisiting the score.
>
> ### Your support would make a significant difference.
>
> Thank you!
>
> Best,
>
> Authors

---

### Official Review · Reviewer_yv3o · 2024-07-22

**Soundness:** 3
**Presentation:** 3
**Contribution:** 3
**Rating:** 6
**Confidence:** 3

**Summary:**

This work investigates the inverse reinforcement learning problem. Previous methods suffer from the weakness: fail to capture the true task objectives from demonstrations. This study proposes deriving a set of candidate reward functions that align with the task rather than merely the data. Using an adversarial learning approach, the policy is trained with this set of reward functions. Then the proposed framework collectively validates the policy’s ability for task accomplishment. The proposed method is theoretically derived with detailed derivation and experimentally implemented on MiniGrid environments with impressive empirical results.

**Strengths:**

The paper is well-structured, clearly explained, and nicely presented.

The approach has a solid theoretical foundation.

The method shows significant advantages over baseline approaches on MiniGrid DoorKey, especially with a single demonstration.

**Weaknesses:**

The experiments on MiniGrid are not very extensive. Beyond DoorKey, MiniGrid includes more challenging environments such as ObstructedMaze and UnLock. Demonstrating advantages in these environments would be more convincing.

The proposed approach requires a set of candidate reward functions. While it achieves a higher success rate than baselines, it is unclear if this comes at a higher time cost. Comparing the time efficiency with prior work would be beneficial.

**Questions:**

In Appendix Figure 5, the proposed approach does not significantly better than the baselines in locomotion tasks (Walker, HalfCheetah, Hopper, Swimmer, etc.). Would the method show more advantages if only 1 demonstration is used instead of 10 or 100?

The MiniGrid DoorKey 6x6 experiment just requires a demonstration with a short horizon. In larger mazes or more challenging environments with complicated demonstrations, would the proposed approach still perform impressively with only 1 demo?

**Limitations:**

There is no analysis explaining why the proposed method performs better on MiniGrid (Figure 2) than on locomotion tasks (Appendix Figure 5). Studying the scenarios where the method does not work well and providing explanations would help researchers determine the appropriateness of using this approach.

---

> ### Author Rebuttal · Authors · 2024-08-07
>
> > ## MiniGrid includes more challenging environments
>
> As our approach builds upon existing IRL/IL methods (PAGAR-GAIL, PAGAR-VAIL), its performance is influenced by the integrated IRL/IL algorithms. We found that the more complex MiniGrid environments, such as ObstructedMaze and KeyCorridor, are too challenging for our integrated IRL/IL methods and the baselines to produce meaningful results. Consequently, we did not include these environments. Future work will focus on enhancing our method and the underlying IRL/IL algorithms to address these more challenging tasks.
>
> > ## ... requires a set of candidate reward functions
>
> We do not explicitly build this set of reward functions. We mention in Section 6.3 that we incorporate the $r\in R_{E,\delta}$ constraint into the objective function. Algorithm 1 shows that the policy optimization and reward searching processes are performed alternately, similar to those in GAN-based IRL/IL methods.
>
> > ## Comparing the time efficiency with prior work would be beneficial.
>
> We appreciate this suggestion and will analyze the time efficiency in the updated version of the paper. The primary potential increase in time complexity arises from the need to optimize two policies, $\pi_A$ and $\pi_P$. However, these two policies can explore the environment and be optimized in parallel, mitigating the redundant time complexity. Additionally, the estimation of Equations 6 and 7 introduces some overhead: we evaluate policy $\pi_A$'s output along $\pi_P$'s samples and vice versa. This increased time complexity is linear in the size of the replay buffers of the two policies.
>
> > ## would the method show more advantages if only 1 demonstration
>
> We did attempt to use only one demonstration. However, our approach, like the integrated GAN-based IRL/IL algorithms, suffers when only one demo is provided. Our conjecture is that the $R_{E,\delta}$ set can be overwhelmed by task-misaligned reward functions that overfit the single trajectory, leading to poor performance. Our approach aims to improve task alignment under general non-ideal situations, but it is not optimized to excel in one-shot IL across baselines.
>
> > ## In larger mazes or more challenging environments with complicated demonstrations, would the proposed approach still perform impressively with only 1 demo?
>
> In environments like S9N1, IRL/IL baselines struggle with only one demo, while our approach shows better results. However, as tasks become more complex, the partial observability significantly hinders the IRL/IL baselines from producing meaningful results, which can affect the performance of our method, as mentioned earlier. We have tested our algorithm and the baselines in the KeyDoorS3 and ObstructedMaze environments but did not achieve satisfying results.
>
>
> > ## There is no analysis explaining why the proposed method performs better on MiniGrid (Figure 2) than on locomotion tasks
>
> We discussed this briefly in lines 226-234. The intuition is that MiniGrid tasks, which are primarily about reachability, have a clear binary success or failure criterion. Such tasks present a distinct discrepancy between task-aligned and misaligned reward functions, as well as between acceptable and unacceptable policies. In contrast, locomotion tasks exhibit more quantitative and nuanced differences between policies. Our method aims to learn an **acceptable** policy, which is not necessarily an **optimal** policy under some 'ground truth' reward function -- in Proposition 2, we show that the policy is optimal under a mixture of rewards.
>
> Another probable explanation is that we did not use high-quality demos in locomotion to simulate non-ideal situations, as shown by the expert's total reward in Table 2. In MiniGrid, even a low-quality demo ends up reaching the goal, and our algorithm captures this property. However, in locomotion, low-quality demos may exhibit different behaviors, such as low speed or perfect balance followed by a crash.

---

> ### Comment · Reviewer_yv3o · 2024-08-12
> **Thanks for Author Response**
>
> Thanks for the detailed response. I'd keep my score.

---

> ### Author Response · Authors · 2024-08-12
>
> Dear Reviewer,
>
> Thank you for recognizing the effort in our response.
>
> Please let us know if further clarification or additional evaluations would help in revisiting the score.
>
> Your support would make a significant difference.
>
> Bests,
>
> Authors

---

### Author Rebuttal · Authors · 2024-08-07

We appreciate the reviewers' insights and suggestions. In this general response, we would like to reiterate the motivation of this paper and share our additional experimental results.


> ## PAGAR-Based IL for Task-Alignment

In this paper, we focus on aligning with tasks that fit the format described in Definition 1, which is based on the **NeurIPS 2021 best paper** [1].  A significant characteristic of these tasks is the presence of a binary success or failure criterion between policies. Our method aims to learn an **acceptable** policy that succeeds in such a task, rather than an **optimal** policy under some 'ground truth' reward function. As a result, our approach's effectiveness is more pronounced in tasks where success and failure are distinctly binary, such as in MiniGrid, than in tasks with more quantitative performance metrics, such as continuous control tasks. Therefore, we included our MiniGrid experimental results in the main text and placed the Mujoco experimental results in Appendix C.2.


> ## Applying PAGAR to offline IL tasks

Our additional experiments combine PAGAR with RECOIL [2], a SOTA **offline IL** algorithm. In these experiments, we used the standard offline datasets from D4RL as baselines. Our results below demonstrate the compatibility of PAGAR with offline IL algorithms and highlight this as a new feature of our method.

|        | RECOIL | (Ours) PAGAR_RECOIL|
|--------|------- | ------------|
| hopper-random |$106.87 \pm 2.69$ | $111.16\pm 0.51$ |
| halfcheetah-random | $80.84\pm17.62$  | $92.94\pm 0.10$ |
| walker2d-random | $108.40\pm 0.04$ | $108.40\pm 0.12$|
| ant-random | $113.34\pm2.78$ | $121\pm 5.86$ |


Here are some details. The original RECOIL algorithm learns a $Q$, $V$ value functions and a policy $\pi$.  In order to combine PAGAR with RECOIL, we made the following modification.
* Instead of learning the $Q$ value function, we **explicitly** learn the reward  function $r$:
    * The $r$ reward function takes the current state $s$, action $a$ and the next state $s'$ as input, and outputs a real number -- the reward.
    * We use the same loss function as that for optimizing $Q$ in RECOIL to optimize $r$ by replacing $Q(s,a)$ with $r(s,a,s')+\gamma V(s')$ for every $(s,a,s')$ sampled from the offline dataset.
    * We use the same loss function for optimizaing the $V$ value function as in RECOIL to still optimize $V$, except for replacing the target $Q(s,a)$ with target $r(s,a,s') + \gamma V(s')$ for every $(s,a,s')$ experience sampled from the offline dataset.
* Instead of learning a single policy as in RECOIL, we learn a protagonist and antagonist policies $\pi_P$ and $\pi_A$. We use the same SAC-like policy update rule as in RECOIL to train each policy, except for replacing $Q(s,a)$ with $r(s,a,s')+\gamma V(s')$ for every $(s,a,s')$ experience sampled from the offline dataset.
* With some heuristic, we construct a PAGAR-loss that is proportional to $R(s,a,s') * max(0, \frac{\pi_P(a|s)}{\pi_A(a|s)})$ for $(s,a,s')$ in the expert demonstrations plus $R(s,a,s') *  min(0, \frac{\pi_P(a|s)}{\pi_A(a|s)})$ for $(s,a,s')$ in the offline random sample set. For simplicity, we multiply this PAGAR-loss with a fixed Lagrangian parameter $\lambda=1e-3$ and add it to the aforementioned loss for optimizing $r$.
* We directly build our implementation upon the code base realsed by RECOIL's author, and tested it on D4RL datasets, with the same configurations as reported in the RECOIL paper [2]. In particular, offline IL uses expert and sub-optimal sample sets to learn policies. We use the D4RL's 'expert' datasets as the expert demonstrations and the 'random' datasets for Mujoco environments as the offline suboptimal dataset. The results in the table above are averaged from 4 seeds.


While our experimental results also show improved performance in continuous tasks, such as those in Appendix C.2 and the offline IL tasks introduced above, our primary focus remains on task alignment. This paper serves as a debut and theoretical introduction to our task-alignment ideology.

[1] Abel, David, et al. On the expressivity of Markov reward, NeurIPS 2021

[2] Sikchi et al.; Dual RL: Unification and new methods for reinforcement and imitation learning, ICLR 2024

---

### Author Response · Authors · 2024-08-12
**Suppplementary Experimental Results: Comparing PAGAR with f-IRL in Continuous Control Tasks**

We thank the reviewers again for their valuable comments. We hope that our previous responses have addressed most of your concerns. To further support our rebuttal, we would like to share some additional experimental results. We have conducted experiments combining PAGAR with **f-IRL**[1], and the results are summarized in the following table. Our PAGAR-based approach substantially outperforms the baseline with fewer exploration iterations in the environment across multiple standard IRL/IL benchmarks in **continuous control**. Ideally, these should be presented as **plots** to better illustrate the performance trends. In the revised version of the paper, we will ensure that these results are presented appropriately.

|   Hopper | f-IRL (FKL) | (Ours) PAGAR-f-IRL (FKL) |
|--------           |-------                 | ------------|
|Env Step 0 ~ 2.5e5   |  $937.55\pm251.59$   |    $999.25\pm236.45$     |
|Env Step 2.5e5 ~ 5e5 |  $1168.47\pm191.36$  |  $2051.02\pm424.40$        |
|Env Step 5e5 ~ 7.5e5 | $1693.07\pm337.33$    |  $2975.78\pm108.25$ |
|Env Step 7.5e5 ~ 1e6 | $2637.24\pm304.30$ |     $3159.55\pm84.38$          |



|   Ant | f-IRL (FKL) | (Ours) PAGAR-f-IRL (FKL) |
|--------               |-------                 | ------------|
| Env Step 0 ~ 5e5      |    $724.91\pm84.05$ | $722.57\pm76.32$ |
| Env Step 5e5~1e6       |$1021.42\pm721.94$ | $1806.51\pm 1340.36$ |
| Env Step 1e6 ~1.5e6   | $3566.84 \pm 684.62$  | $4668.45\pm466.18$|
| Env Step 1.5e6~2e6 | $4484.41\pm297.48$ |  $5008.88\pm371.89$ |



|   HalfCheetah | f-IRL (FKL) | (Ours) PAGAR-f-IRL (FKL) |
|--------               |-------                 | ------------|
| Env Step 0 ~ 5e5      | $66.83\pm   103.92$      | $67.75\pm 97.73$|
| Env Step 5e5~1e6       | $535.342\pm 316.16$     | $515.56 \pm 334.29$ |
| Env Step 1e6 ~1.5e6   | $5710.97\pm 2533.63$    | $6695.71\pm3062.43$ |
| Env Step 1.5e6~2e6    |   $10860.97\pm667.82$  | $11195.02\pm554.76$ |



|   Walker2D | f-IRL (FKL) | (Ours) PAGAR-f-IRL (FKL) |
|--------               |-------                 | ------------|
| Env Step 0 ~ 5e5      |$975.37\pm178.59$ | $974.58\pm194.77$ |
| Env Step 5e5~1e6       | $1442.53\pm448.38$ | $1737.44\pm781.81$ |
| Env Step 1e6 ~1.5e6   | $3501.35\pm827.54$ | $3661.00\pm270.14$|
| Env Step 1.5e6~2e6    | $4490.94\pm553.47$ | $4500.95\pm529.30$|

Our implementation is built on the codebase provided by the authors of f-IRL. The method for combining PAGAR with f-IRL is analogous to our approach with GAIL, using the same adversarial reward learning objectives as outlined in Equations 6, 7, 32, and 34. We also use a fixed $\lambda$ parameter. For the IRL reward learning and RL policy learning process, we have directly utilized the implementation from f-IRL. Detailed explanations will be included in the revised version of the paper.

[1] Ni, Tianwei, et al. "f-irl: Inverse reinforcement learning via state marginal matching." CORL 2021.

---

### Author Response · Authors · 2024-08-14
**Thank You for Your Constructive Feedback and Suggestions**

Dear Reviewers,

We sincerely appreciate your time and effort in reviewing our submission. Your insights are invaluable, and we are grateful for the constructive feedback provided throughout this process. Your comments have significantly helped us refine and clarify our contributions.

> ## 1.	The Paper’s Main Idea

This paper proposes reorienting Inverse Reinforcement Learning (IRL)-based Imitation Learning (IL) from data alignment to task alignment. We have worked to clearly articulate the core idea of our work, as outlined in our response:
 * The **goal** of our method is to learn a policy that fulfills tasks as formulated in Definition 1, which is general and encompasses many real-world examples, as demonstrated in [1] and as proved in our individual response.
 * We introduced the concept of task-aligned reward functions and explained our **insight** that achieving high utility under these task-aligned reward functions is essential for task fulfillment.
 * Our **approach** treats expert demonstrations as weak supervision to learn a set of reward functions that capture task-aligned rewards and then trains a policy to achieve high utility across this set.

> ##  2.	Complexity and Relevance of Benchmarks

We have emphasized that the MiniGrid environment is a well-established RL benchmark with its own publication at NeurIPS 2023 [2]. Despite its seemingly simple appearance, MiniGrid presents significant challenges for current IRL/IL algorithms due to its **massive state space** (partially observable, random layout in every episode, etc.), and other complexities. We will further clarify this in our revised manuscript to address any concerns regarding the complexity of the environments we used.

> ## 3.	Expanded Experimental Evaluation

In response to requests for more diverse and challenging experiments, we have included new experimental results, such as **continuous control tasks** and additional baselines like **f-IRL** and **RECOIL** [3]. These results demonstrate the broader applicability of our method in both online and **offline IL** and the effectiveness of our method across different types of environments, further reinforcing the robustness of our approach.

> ## 4. Moving Forward

We have carefully considered your feedback and will take several steps to further improve our submission. We plan to enhance the clarity of our contributions and streamline the theoretical sections to ensure that the main ideas are communicated more effectively. Additionally, we will expand our experimental results on other complex benchmarks, such as Locomujoco.

Once again, we sincerely thank you for your thoughtful reviews and contributions to improving our submission. We are confident that the revisions made in response to your feedback will result in a stronger and more impactful paper.

Best regards,

The Authors

[1] Abel, David, et al. On the expressivity of Markov reward, NeurIPS 2021

[2] Chevalier-Boisvert, Maxime, et al. “Minigrid & miniworld: Modular & customizable reinforcement learning environments for goal-oriented tasks.” NeurIPS 2023

[3] Sikchi et al.; Dual RL: Unification and new methods for reinforcement and imitation learning, ICLR 2024

---

### Decision · Program_Chairs · 2024-09-25

**Decision:**

Accept (poster)

**Comment:**

This paper develops a novel approach to inverse RL that prioritizes alignment of the learned reward function with the task objectives rather than the demonstration data, as done in typical IRL methods. The authors provide a solid motivation and theoretical foundation for this approach. There were some initial concerns with clarity and the empirical evaluation, focusing mostly on the limited complexity and scope of the chosen domains. The authors provided additional results and clarifications in their rebuttal that partially addressed these concerns, yet reviewers still felt that a more substantial evaluation with more complex environments are needed for a final version of this paper. The authors are strongly encouraged to incorporate this advice from the reviewers, and their suggestions for clarity improvements, in preparing the next draft of this paper.